# Integrative proteogenomic analysis identifies COL6A3-derived endotrophin as a mediator of the effect of obesity on coronary artery disease

Satoshi Yoshiji [1,2,3,4,5] ✉, Tianyuan Lu [2,6,7,8], Guillaume Butler-Laporte [2,9,10], Julia Carrasco-Zanini-Sanchez [11,12], Chen-Yang Su [2,3,13], Yiheng Chen[1,2,14], Kevin Liang[2,13], Julian Daniel Sunday Willett [2,13,15], Shidong Wang[16], Darin Adra[2], Yann Ilboudo[2], Takayoshi Sasako[2], Satoshi Koyama [17,18], Tetsushi Nakao [17,18], Vincenzo Forgetta[14], Yossi Farjoun[2,19], Hugo Zeberg [20,21], Sirui Zhou [1,2,3], Michael Marks-Hultström [2,22,23], Mitchell J. Machiela [24], Rama Kaalia [25,26], Hesam Dashti[25,26,27], Melina Claussnitzer [25,26,27,28], Jason Flannick [5,29,30], Nicholas J. Wareham[11], Vincent Mooser [1,3], Nicholas J. Timpson [31,32], Claudia Langenberg [11,12,33] & J. Brent Richards [1,2,13,34,35] ✉

Obesity strongly increases the risk of cardiometabolic diseases, yet the underlying mediators of this relationship are not fully understood. Given that obesity strongly influences circulating protein levels, we investigated proteins mediating the effects of obesity on coronary artery disease, stroke and type 2 diabetes. By integrating two-step proteome-wide Mendelian randomization, colocalization, epigenomics and single-cell RNA sequencing, we identified five mediators and prioritized collagen type VI α3 (COL6A3). COL6A3 levels were strongly increased by body mass index and increased coronary artery disease risk. Notably, the carboxyl terminus product of COL6A3, endotrophin, drove this effect. *COL6A3* was highly expressed in disease-relevant cell types and tissues. Finally, we found that body fat reduction could reduce plasma levels of COL6A3-derived endotrophin, indicating a tractable way to modify endotrophin levels. In summary, we provide actionable insights into how circulating proteins mediate the effects of obesity on cardiometabolic diseases and prioritize endotrophin as a potential therapeutic target.

More than 1.9 billion people worldwide have obesity, which is strongly linked to cardiometabolic diseases, including coronary artery disease (CAD), stroke and type 2 diabetes[1,2]. However, most of the factors mediating this relationship are not yet fully understood. Therefore, identification of modifiable mediators could yield potential therapeutic targets. Circulating proteins are potential candidates because obesity broadly influences plasma proteins[3,4], and these have critical roles in disease development and progression. Moreover, circulating proteins can be measured and sometimes perturbed[5]. Thus, understanding their role in disease could provide avenues to lessen the impact of obesity on cardiometabolic diseases.

Mendelian randomization (MR) is a genetic epidemiology approach that can be used to estimate the causal effects of exposures on outcomes while minimizing confounding and avoiding reverse causation[5–12]. Although MR relies on several key assumptions[6,7], when these are met, it can be a powerful tool to describe causal relationships.

**Fig. 1 | Study overview and summary.** To identify proteins mediating the effects of obesity on cardiometabolic diseases, we employed a two-step MR approach. In step 1, we assessed the impact of BMI on 4,907 plasma proteins using two-sample MR, through which we identified 1,213 proteins influenced by BMI, termed 'BMI-driven proteins'. In step 2, we evaluated the effects of these BMI-driven proteins on cardiometabolic diseases through additional two-sample MR analyses. Subsequent work included follow-up analyses of COL6A3 and an evaluation of the potential actionability of this protein and other identified mediators. Created using BioRender.com.

Furthermore, advances in large-scale proteomics have facilitated the discovery of genetic variants that influence plasma protein levels on a proteome-wide scale[13–15]. These variants, referred to as protein quantitative trait loci (pQTLs), can be used in MR to estimate the causal effects of circulating protein levels on disease. Such methods have been successfully leveraged to prioritize therapeutic targets[5,16]. Although drug discovery is costly and prone to failure[17], drugs with human genetics evidence are more likely to be successful in clinical trials[18,19]. Therefore, proteogenomics-based MR could play an important part in prioritization of potential targets. In addition, MR can be leveraged to understand mediators of the biological pathways connecting a risk factor with disease outcomes when applied in a two-step MR study design[20–22]. In step 1, the effects of a risk factor, such as body mass index (BMI), on mediators are estimated. In step 2, the effects of the identified mediators on the outcome of interest are assessed. Previously, we used this approach to identify a circulating protein, nephronectin, that mediates the effect of obesity on COVID-19 severity[22].

Here, by integrated two-step proteome-wide MR, colocalization, observational assessments, epigenomics and single-cell RNA sequencing, we identify five mediators of the effects of obesity on cardiometabolic diseases and prioritize COL6A3 as a potentially actionable therapeutic target.

## Results

### Overall study design

The overall study design and a summary of the results are illustrated in Fig. 1. The study consisted of a two-step MR approach, follow-up analyses of COL6A3 and an assessment of actionability, as follows.

(1) MR step 1 evaluated the causal effect of BMI on the levels of 4,907 circulating plasma proteins.
(2) MR step 2 assessed the causal effects of BMI-driven proteins on four cardiometabolic outcomes (CAD, ischemic stroke, cardioembolic stroke and type 2 diabetes).
(3) Follow-up analyses for COL6A3 and its cleavage product, known as endotrophin, assessed its role in CAD through replication MR, observational assessments, epigenomic analyses and single-cell sequencing analysis.
(4) Assessment of actionability for COL6A3-derived endotrophin and other protein mediators was based on multivariable MR and phenome-wide association studies.

The datasets used in each step are described in Supplementary Table 1. Each of the four analysis steps and the corresponding results are described in detail below.

### MR step 1: Causal effects of BMI on plasma protein levels

We performed causal inference using two-sample MR to estimate the effects of BMI on 4,907 circulating proteins using two separate genome-wide association studies (GWAS) that included participants of European genetic ancestry, one for BMI from the UK Biobank and GIANT consortium ($n$ = 681,275 individuals)[23] and the other for circulating proteins from deCODE[14] ($n$ = 35,559 individuals). We used the inverse-variance weighted (IVW) method for the primary analysis with Bonferroni correction ($P < 0.05/4,907 = 1 \times 10^{-5}$) and then filtered the results with multiple sensitivity analyses, including assessment of heterogeneity, pleiotropy, reverse causation and directional concordance with body fat percentage. No evidence of weak instrumental variables was identified (Supplementary Table 2).

Some proteins were targeted by more than one protein-targeting aptamer. This was the case for COL6A3, whose carboxyl-terminal and amino-terminal regions were targeted by separate aptamers. For clarity, we refer to protein-targeting aptamers as 'proteins' unless otherwise specified. Specifically for COL6A3, we refer to the C-terminal COL6A3-targeting aptamer (aptamer sequence ID (seqID): 11196-31)

as 'C-terminal COL6A3' and the N-terminal COL6A3-targeting aptamer (seqID: 10511-10) as 'N-terminal COL6A3'.

We found that BMI influenced 1,213 proteins, with all of these passing tests of significance and the sensitivity analyses (Fig. 2, Extended Data Fig. 1 and Supplementary Tables 3 and 4). Notably, 94.7% of these proteins showed high concordance in direction between BMI and body fat percentage. ($r$ = 0.92; $P < 2.2 \times 10^{-16}$). Hereafter, these 1,213 proteins are referred to as BMI-driven proteins. The MR step 1 captured well-known associations of BMI with plasma proteins including leptin[24].

### MR step 2: effects of BMI-driven proteins on cardiometabolic diseases

Next, we estimated the causal effects of these BMI-driven proteins on CAD, ischemic stroke, cardioembolic stroke and type 2 diabetes, again using two-sample MR (Fig. 3a). The BMI-driven protein levels identified in MR step 1 were used as exposures. The outcomes were CAD[25] (181,522 cases and 1,165,690 controls), ischemic stroke, cardioembolic stroke[26] (62,100 ischemic stroke cases, 10,804 cardioembolic stroke cases and 1,234,808 controls) and type 2 diabetes[27] (80,154 cases and 853,816 controls). We used GWAS for these traits from individuals of European genetic ancestry to match the data used for BMI. To minimize the risk of horizontal pleiotropy, we used *cis*-acting pQTLs (*cis*-pQTLs) from deCODE[14] as instrumental variables. These variants, which are located close to transcription start sites (TSSs; ±1-Mb region), are more likely to influence outcomes through circulating protein levels than through independent pathways. To further reduce pleiotropy, we restricted *cis*-pQTLs to those associated with only one protein (Extended Data Fig. 1b). After the *cis*-pQTL search and harmonization, up to 350 proteins of the 1,213 that passed step 1 of the MR were retained and evaluated for each outcome in step 2. We applied Bonferroni correction for the number of proteins tested in each MR.

Following MR with *cis*-pQTLs and sensitivity analyses, we performed colocalization to evaluate whether the pQTL of the protein of interest and the disease outcome shared a single causal variant within a 500-kb region surrounding the lead *cis*-pQTL. This approach helped to mitigate bias due to different linkage disequilibrium (LD) structures. No evidence of weak instrumental variables was found (Supplementary Table 2).

We identified nine protein–disease associations after MR, sensitivity analyses and colocalization; these included associations of COL6A3 and PCSK9 with CAD, F11 with ischemic and cardioembolic stroke, and SPATA20 with type 2 diabetes. Among these, COL6A3 was associated with the highest odds of the outcome per one standard deviation increase in protein levels (odds ratio (OR) = 1.47, 95% confidence interval (CI): 1.26–1.70, $P = 4.46 \times 10^{-7}$) (Fig. 3b,c). PCSK9 served as a 'positive control' owing to its established role as a drug target for reducing CAD risk[28–30]. Full results are provided in Supplementary Tables 5–8. Potential bias due to sample overlap in MR was estimated to be minimal (Supplementary Note and Supplementary Table 9).

Given that BMI increased the risk of cardiometabolic diseases ($\beta_{\text{BMI-to-cardiometabolic}} > 0$; OR for CAD = 1.47, 95% CI: 1.38–1.57, $P = 4.2 \times 10^{-31}$; OR for ischemic stroke = 1.21, 95% CI: 1.14–1.28, $P = 7.6 \times 10^{-11}$; OR for cardioembolic stroke = 1.19, 95% CI: 1.05–1.36, $P = 6.2 \times 10^{-3}$; OR for type 2 diabetes = 2.58, 95% CI: 2.20–3.03, $P = 1.8 \times 10^{-31}$), we further restricted the analysis to proteins that met the condition $\beta_{\text{BMI-to-protein}} \times \beta_{\text{protein-to-cardiometabolic}} > 0$. Of the nine protein–disease associations, six met this criterion (Fig. 3b and Supplementary Table 10).

### Follow-up analyses of C-terminal COL6A3

Circulating C-terminal COL6A3 levels were strongly increased by BMI—more so than those of N-terminal COL6A3 (Fig. 2b)—and showed the highest odds of the outcome per standard deviation increase among all mediators of the effects of BMI on the outcomes (Fig. 3b). We therefore further tested the hypothesis that C-terminal COL6A3 mediates the relationship between obesity and CAD using analyses from orthogonal resources. In addition, we examined the domain-specific effect of COL6A3.

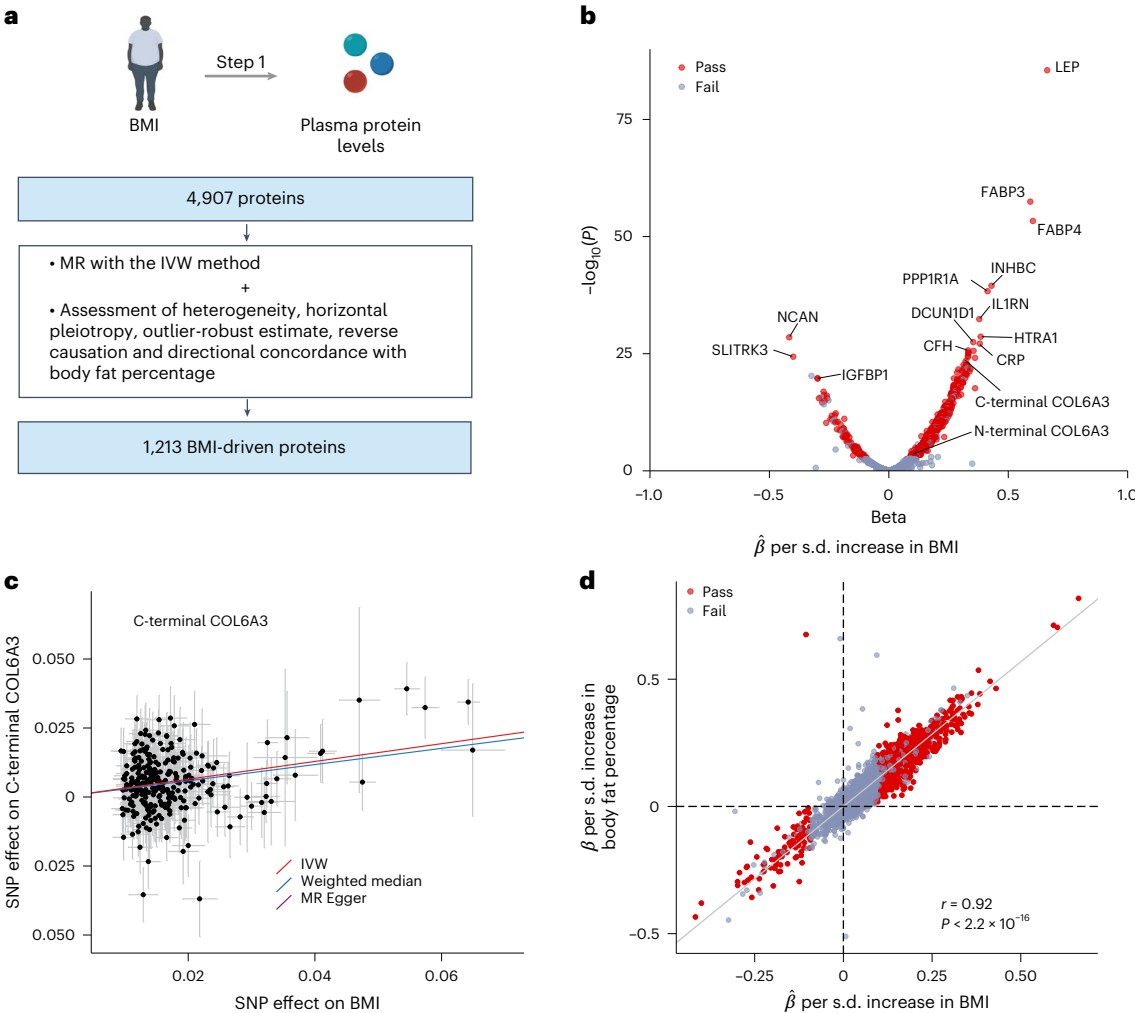

**Fig. 2 | MR step 1: estimating the causal effects of BMI on plasma protein levels. a**, Flow diagram outlining MR step 1. **b**, Volcano plot illustrating the effects of BMI on each plasma protein from MR analyses using the IVW method. The *x* axis represents beta estimates, and the *y* axis represents −log₁₀*P* values from MR results. The *P* values were obtained using the random-effects IVW method (two-sided test). Red dots represent proteins that passed all tests, including significance with Bonferroni correction (*P* < 0.05/4,907), as well as tests for heterogeneity, directional horizontal pleiotropy, reverse causation and directional concordance with body fat percentage. Gray dots represent proteins that failed any of these tests. **c**, MR scatter plot showing the effects of BMI on

plasma levels of COL6A3 using the IVW (primary analysis; red regression line), weighted median (blue regression line) or MR-Egger slope (purple regression line) methods. Note that the MR-Egger slope (purple: *β* = 0.32) overlaps with the IVW slope (red: *β* = 0.32). Error bars represent the 95% CI for each variant's effect estimate. **d**, Directional consistency between MR results for the effects of BMI on plasma proteins and MR results for the effects of body fat percentage on plasma protein levels using the IVW method. The *x* axis denotes beta estimates from MR results, and *r* denotes Pearson's correlations. *P* values were obtained using two-sided Pearson's correlation test.

## Replication MR for step 1 and step 2 for C-terminal COL6A3

For MR step 1, we repeated the MR using the GWAS from Fenland and replicated the finding that a standard deviation increase in BMI increased levels of C-terminal COL6A3 (*β* = 0.31, 95% CI: 0.20–0.42, *P* = 4.8 × 10⁻⁸). For step 2, we repeated the MR using different sources of *cis*-pQTLs from other cohorts (Methods). MR across all cohorts supported the causal effect of C-terminal COL6A3 levels on CAD, with consistent directionality (Supplementary Table 11). Specifically, we found that each standard deviation increase in C-terminal COL6A3 increased the odds of CAD in the UK Biobank[31] (OR = 1.30, 95% CI: 1.17–1.45, *P* = 2.4 × 10⁻⁶), Fenland (OR = 1.23, 95% CI: 1.12–1.35, *P* = 8.8 × 10⁻⁶) and ARIC (OR = 1.09, 95% CI: 1.05–1.13, *P* = 1.6 × 10⁻⁵) data. Notably, UK Biobank used the Olink Explore 3072 assay[31], whereas deCODE[14], Fenland[13] and ARIC[15] used the SomaScan v.4 assay. The Olink Explore 3072 assay targets the same C-terminal domain as the C-terminal COL6A3-targeting aptamer (seqID: 11196-31). The consistent MR results across cohorts and assays strengthen the evidence for a causal role of C-terminal COL6A3 in CAD risk.

## Observational evaluation in the EPIC-Norfolk cohort

To triangulate evidence[32], we performed an observational association analysis with a randomly selected subcohort of the EPIC-Norfolk study (*n* = 872), which included 207 prevalent or incident cases of CAD. BMI was associated with increased plasma levels of C-terminal COL6A3 (*β* = 0.06, 95% CI: 0.04–0.08, *P* = 8.5 × 10⁻¹²), and a standard deviation increase in plasma C-terminal COL6A3 levels increased the odds of CAD (OR = 1.34, 95% CI: 1.12–1.59, *P* = 1.1 × 10⁻³), consistent with the MR results.

## BMI and plasma C-terminal COL6A3 levels in the UK Biobank

We performed multivariable linear regression analysis to evaluate the association between baseline BMI and plasma C-terminal COL6A3 levels in 35,100 individuals from the UK Biobank. After adjustment for age, sex, recruitment center and protein sample processing time, BMI was significantly associated with increased C-terminal COL6A3 level (*β* = 0.073, 95% CI: 0.071–0.075, *P* < 2.2 × 10⁻¹⁶; Supplementary Table 12).

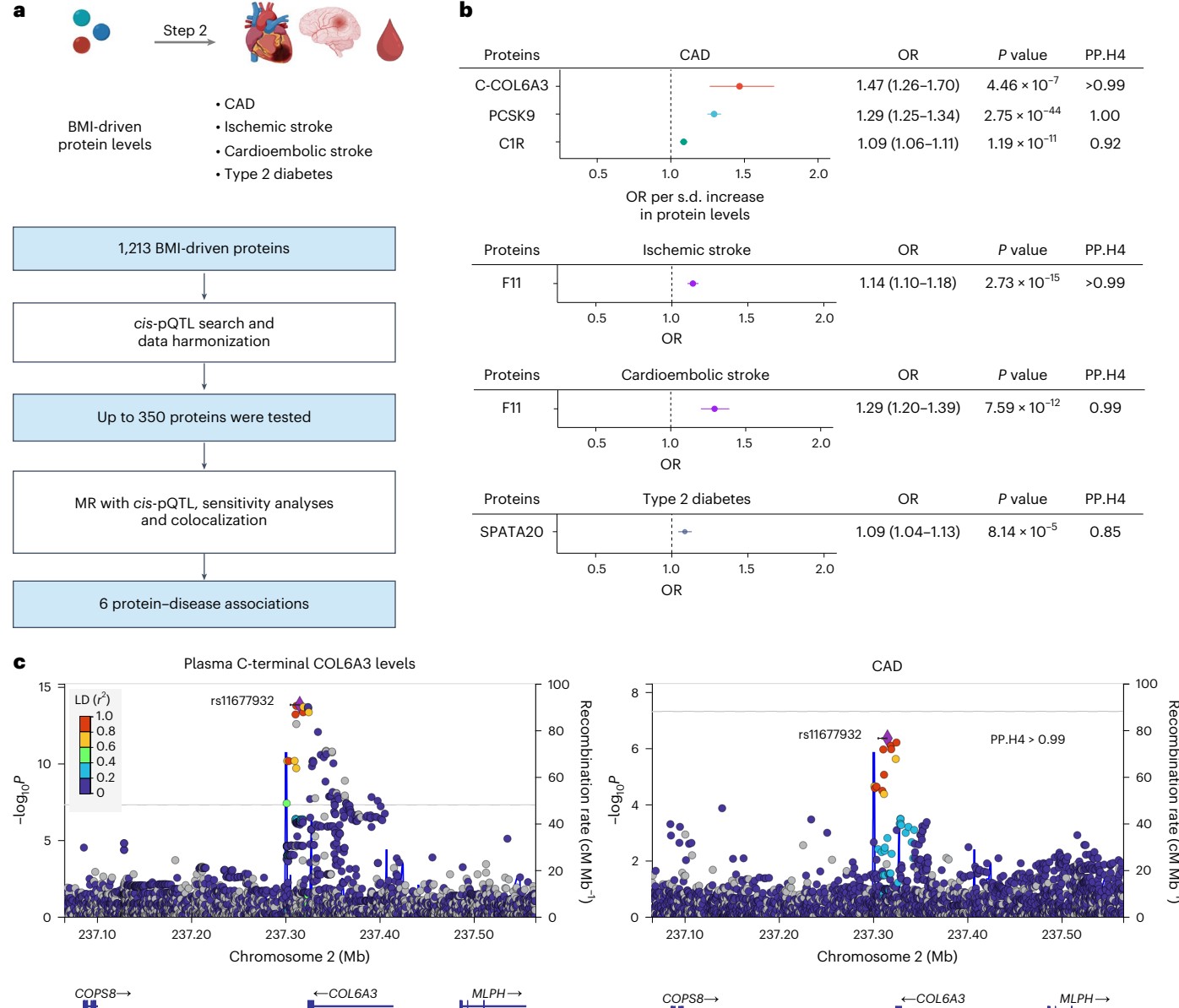

**Fig. 3 | Step 2 MR: estimating the causal effects of BMI-driven proteins on cardiometabolic diseases. a**, Flow diagram of the MR step 2 analyses. **b**, Forest plots showing the effects of BMI-driven proteins on four cardiometabolic diseases (CAD, ischemic stroke, cardioembolic stroke and type 2 diabetes). The MR analyses were conducted using the largest available GWAS of coronary artery disease[25] (181,522 cases and 1,165,690 controls), ischemic stroke (62,100 cases and 1,234,808 controls), cardioembolic stroke[26] (10,804 cases and 1,234,808 controls) and type 2 diabetes[27] (80,154 cases and 853,816 controls). C-COL6A3, C-terminal COL6A3. *P* values were obtained using the random-effects IVW method (two-sided test). Error bars represent the 95% CI for effect estimates. **c**, LocusZoom plots of (left) the pQTL for C-terminal COL6A3 and (right) CAD in the 500-kb region surrounding the lead *cis*-pQTL, rs11677932. PP.H4, posterior probability of having the shared causal variant (hypothesis H4 in colocalization). **a**, created using BioRender.com.

## Cox regression analysis for cumulative incidence of CAD

We also performed multivariable Cox proportional-hazards regression analysis in 38,361 individuals (2,969 cases and 32,131 controls) from the UK Biobank (Supplementary Table 12). After adjusting for the above covariates, a standard deviation increase in plasma C-terminal COL6A3 levels was associated with increased cumulative incidence of CAD (hazard ratio (HR) = 1.40, 95% CI: 1.35–1.45, $P < 2.2 \times 10^{-16}$).

Finally, we plotted Kaplan–Meier estimates for the cumulative incidence of CAD stratified by baseline COL6A3 level quantiles (25th, 50th, 75th and 100th percentiles; Fig. 4). The highest incidence was observed in the 75–100th percentile group, with the lowest in the 0–25th percentile group (log-rank test $P < 2.2 \times 10^{-16}$).

Given the robustness of these findings, we then explored the potential mechanism whereby C-terminal COL6A3 may influence CAD.

## Identification of the causal domain of COL6A3

The *COL6A3* locus has been identified as a putative causal gene in multiple CAD GWAS[25,33,34]; however, its specific mechanism remains unclear. The C-terminal domain of COL6A3, known as the Kunitz domain, is cleaved to form endotrophin, a fragment that has been implicated in fibrosis and inflammation and in obesity-induced metabolic dysfunction[35–40] (Fig. 5a). Thus, we evaluated whether this domain drives the effect of COL6A3 on CAD.

The SomaScan v.4 assay involves two separate aptamers targeting two domains of COL6A3, the N-terminal and C-terminal

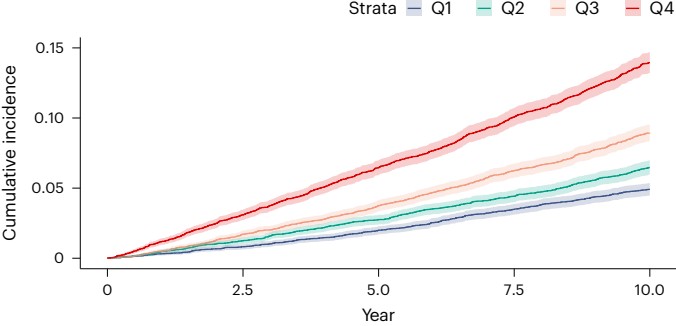

**Fig. 4 | Baseline COL6A3 levels and cumulative incidence of CAD.** Multivariable Cox proportional-hazards regression analysis in 38,361 individuals (2,969 cases and 32,131 controls) from the UK Biobank. Q1 (blue) represents the lowest 25% group, Q2 (green) the 26–50% group, Q3 (orange) the 51–75% group and Q4 (red) the highest quantile group (76–100%, from the 75th percentile to the maximum value) based on baseline plasma COL6A3 levels. The center lines represent effect estimates in each group, and the shaded areas around the lines represent 95% CIs.

(Kunitz) domains, allowing us to disentangle their effects. Notably, the C-terminal-binding aptamer (Fig. 5b) increased the risk of CAD (OR = 1.47 per standard deviation increase in the protein level, 95% CI: 1.26–1.70, $P = 4.5 \times 10^{-8}$), whereas the N-terminal-binding aptamer (that is, the non-cleaved portion of COL6A3) did not (OR = 1.00, 95% CI: 0.87–1.14, $P = 0.98$) in domain-aware MR (Fig. 5c and Supplementary Table 13). This suggests that the C-terminal domain, which forms endotrophin, explains the effect of COL6A3 on CAD, and the aptamer binding to the C-terminal of COL6A3 may capture the plasma levels of endotrophin or endotrophin-containing fragments. Hereafter, we refer to such fragments as endotrophin for clarity.

To further test this hypothesis, we repeated the MR with *cis*-pQTL from the UK Biobank, which used the Olink Explore 3072 assay[31,41]. The Olink Explore 3072 assay uses a polyclonal antibody to target the C-terminal (Kuniz domain) of COL6A3. MR showed that a standard deviation increase in C-terminal COL6A3 levels strongly increased the odds of CAD (OR = 1.30, 95% CI: 1.17–1.45, $P = 2.4 \times 10^{-6}$) (Supplementary Table 13), supported by colocalization (Extended Data Fig. 2). The *cis*-pQTL (rs1050785) from UK Biobank was in high LD ($R^2 = 0.73$) with the *cis*-pQTL (rs11677932) of the C-terminal-targeting aptamer from deCODE, but it was not in LD ($R^2 = 0.0$) with the *cis*-pQTL of the N-terminal-targeting aptamer of COL6A3 (rs2646260). These findings from two proteomic assays suggest that circulating endotrophin is likely to explain the effect of COL6A3 on CAD.

Moreover, domain-aware MR for step 1 showed that BMI more strongly elevated C-terminal COL6A3 (cleaved portion) levels ($\beta = 0.32$, 95% CI: 0.26–0.38, $P = 3.7 \times 10^{-24}$) than those of N-terminal COL6A3 (uncleaved portion) ($\beta = 0.10$, 95% CI: 0.04–0.16, $P = 2.1 \times 10^{-3}$), as shown by the nonoverlapping CIs (Supplementary Table 13). Similarly, body fat percentage had a stronger effect on C-terminal COL6A3 ($\beta = 0.38$, 95% CI: 0.30–0.46, $P = 8.3 \times 10^{-20}$) than on N-terminal COL6A3 ($\beta = 0.14$, 95% CI: 0.07–0.22, $P = 1.2 \times 10^{-4}$) (Fig. 5d). These results indicate that obesity, as measured by BMI and body fat percentage, preferentially increases C-terminal COL6A3 and endotrophin.

### Body fat compartments and C- and N-terminal COL6A3 levels

We performed MR analyses to determine which body fat compartments specifically influence plasma C- and N-terminal COL6A3 levels, using GWAS of magnetic resonance imaging (MRI)-derived fat compartment volumes for two-sample MR (Fig. 5e). Among the three fat compartments—abdominal subcutaneous adipose tissue, visceral adipose tissue and gluteofemoral adipose tissue—abdominal subcutaneous adipose tissue significantly increased plasma levels of both C- and N-terminal COL6A3, with a more pronounced increase for C-terminal COL6A3

(Fig. 5e and Supplementary Table 14). Notably, this finding aligns with those of previous studies showing that the collagen matrix surrounding subcutaneous fat is rich in C-terminal COL6A3 (ref. 39), which in turn releases endotrophin; and that increased expression of *COL6A3* in subcutaneous adipose tissues and increased plasma abundance of endotrophin, the cleaved product of the C-terminal of COL6A3, are associated with adipose tissue fibrosis, insulin resistance and metabolic dysfunction[40,42–45].

### Fine-mapping, variant-to-gene mapping and epigenomics

The lead *cis*-pQTL for C-terminal COL6A3 in the deCODE cohort, rs11677932, was identified as a *cis*-pQTL specifically for C-terminal COL6A3—not for N-terminal COL6A3 or any other protein—underscoring its specificity. We further investigated the profile of rs11677932 through fine-mapping, variant-to-gene (V2G) mapping and epigenetic data analysis.

### Fine-mapping using SuSiE

Fine-mapping with SuSiE in the 500-kb region surrounding rs11677932 revealed that this variant was within the 95% credible set and had the highest posterior inclusion probability (PIP) for both the C-terminal COL6A3 pQTL and CAD ($\text{PIP}_{\text{COL6A3}} = 35.2\%$, $\text{PIP}_{\text{CAD}} = 29.3\%$; Supplementary Table 15).

### V2G mapping

By querying the Open Targets Genetics database[46], we mapped rs11677932 to *COL6A3* with the highest V2G score. This was supported by its identification as a pQTL in an independent study (ref. 47), enhancer–TSS interactions in the FANTOM5 CAGE enhancer atlas[48], and its proximity to the gene's TSS, reinforcing its role as a valid instrumental variable (Supplementary Table 16).

### Regulatory role of the lead *cis*-pQTL of C-terminal COL6A3

We assessed the regulatory potential of rs11677932 using ENCODE[49] and RegulomeDB[50]. RegulomeDB assigns a heuristic ranking score, representing the potential to be functional in regulatory elements[51]. The variant received a strong RegulomeDB score of 1b based on expression QTL (eQTL), transcription factor (TF) binding, TF motif, DNase footprint and DNase peak. The variant was located in an open chromatin region and an active enhancer domain in multiple tissues, including adipose tissues, coronary arteries and aorta (Fig. 6). Notably, rs11677932 was the *cis*-eQTL of COL6A3 in the aorta in GTEx v.8, with directionally concordant effects. Moreover, the variant disrupted the conserved nucleotide in the MEF2B TF binding motif (Fig. 6), consistent with the reduced effects of this variant on *COL6A3* expression and plasma protein levels. In addition, in adipose-derived mesenchymal cells from 44 bariatric surgery patients, rs11677932 was significantly associated with morphological changes in adipocytes (Supplementary Note 3).

### Sex-stratified analyses of C-terminal COL6A3

To assess sex differences in associations, we performed step 1 MR (BMI to C-terminal COL6A3) and step 2 MR (C-terminal COL6A3 to CAD risk) in a sex-stratified manner. We generated sex-stratified pQTLs for C-terminal COL6A3 in 19,747 females and 16,876 males from the UK Biobank and identified *cis*-pQTL for both sexes (Extended Data Fig. 3a).

In Step 1 MR, BMI increased C-terminal COL6A3 levels in females ($\beta = 0.16$, 95% CI: 0.11–0.20, $P = 1.1 \times 10^{-12}$) and males ($\beta = 0.10$, 95% CI: 0.05–0.14, $P = 1.4 \times 10^{-5}$) (Extended Data Fig. 3b and Supplementary Table 17), replicating the result of the sex-combined analysis. In step 2 MR, a standard deviation increase in plasma C-terminal COL6A3 levels increased CAD risk in males (OR = 1.63, 95% CI: 1.34–1.98, $P = 1.26 \times 10^{-6}$), whereas females showed a positive trend (OR = 1.18, 95% CI: 0.95–1.46, $P = 0.13$) (Extended Data Fig. 3c and Supplementary Table 17). Colocalization showed a shared causal signal for pQTLs between sexes ($\text{PP}_{\text{shared}} > 0.99$), with rs105785 as the lead pQTL (Extended Data Fig. 3d); it was also the lead pQTL in the sex-combined pQTL.

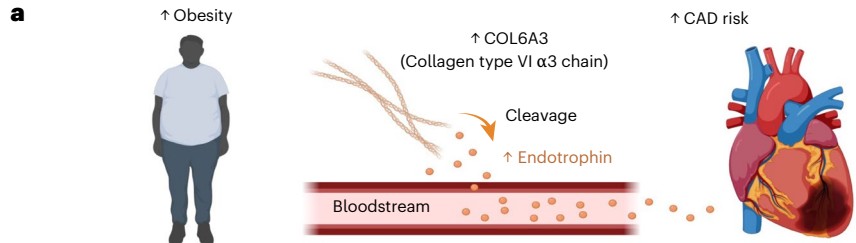

**b** COL6A3 (UniProt ID: P12111)

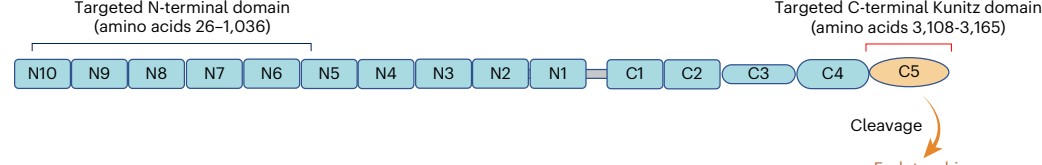

**c** MR for the effects of C- and N-terminal COL6A3 on the risk of CAD stratified by C- and N-terminal COL6A3

| Exposure | Outcome | | OR (95% CI) | P value |
|---|---|---|---|---|
| C-terminal | CAD | | 1.47 (1.26–1.70) | 4.5 × 10⁻⁷ |
| N-terminal | CAD | | 1.00 (0.87–1.14) | 0.98 |

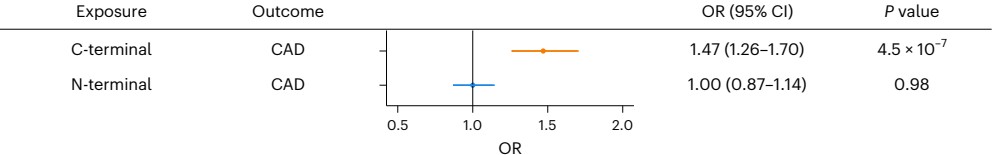

**d** MR for the effects of BMI and body fat percentage on COL6A3 stratified by C- and N-terminal COL6A3

| Exposure | Outcome | | Beta (95% CI) | P value |
|---|---|---|---|---|
| BMI | C-terminal | | 0.32 (0.26–0.38) | 3.65 × 10⁻²⁴ |
| | N-terminal | | 0.10 (0.04–0.16) | 2.07 × 10⁻³ |
| Body fat percentage | C-terminal | | 0.38 (0.30–0.46) | 8.33 × 10⁻²⁰ |
| | N-terminal | | 0.14 (0.07–0.22) | 1.24 × 10⁻⁴ |

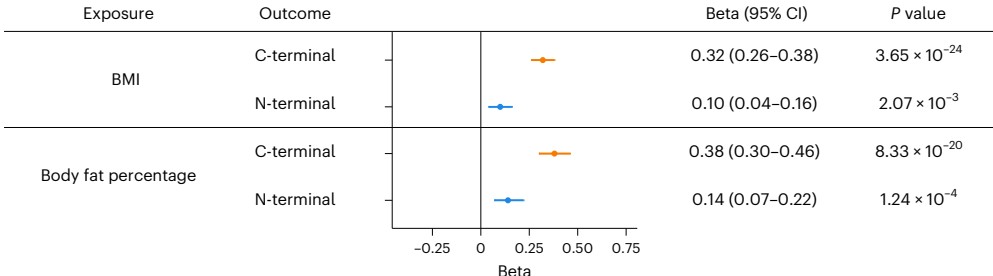

**e** MR for the effects of body fat compartments on COL6A3 stratified by C- and N-terminal COL6A3

| Exposure | Outcome | | Beta (95% CI) | P value |
|---|---|---|---|---|
| Abdominal subcutaneous adipose tissue | C-terminal | | 0.52 (0.35–0.69) | 9.23 × 10⁻¹⁰ |
| | N-terminal | | 0.24 (0.07–0.42) | 7.26 × 10⁻³ |
| Visceral adipose tissue | C-terminal | | 0.16 (–0.28, 0.60) | 0.47 |
| | N-terminal | | 0.08 (–0.19, 0.35) | 0.56 |
| Gluteofemoral adipose tissue | C-terminal | | 0.09 (–0.13, 0.30) | 0.43 |
| | N-terminal | | 0.12 (0.01–0.22) | 0.03 |

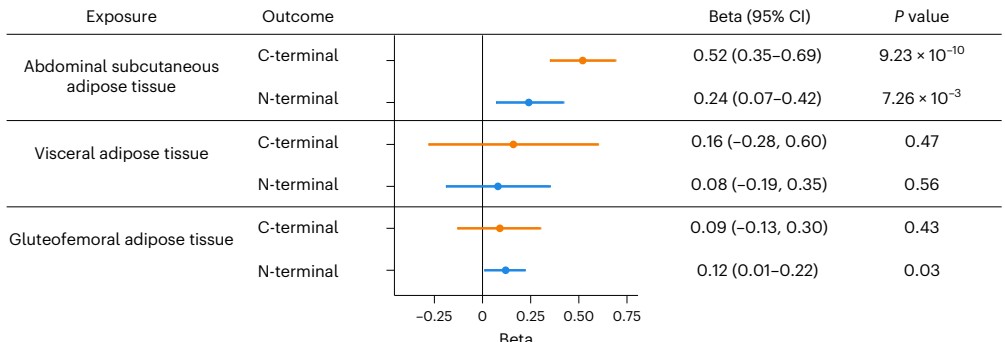

**Fig. 5 | Follow-up analyses for COL6A3. a**, Schematic illustration of proposed relationship between obesity, COL6A3, endotrophin and CAD. Obesity leads to increased production of COL6A3, whose C-terminal is cleaved into an active form termed endotrophin, which increases the risk of CAD. **b**, Schematic diagram of COL6A3 (UniProt ID: P12111). COL6A3 comprises a short collagenous region (gray line between N1 and C1) flanked by multiple von Willebrand factor type A modules, specifically N1–N10 in the N-terminal region and C1–C2 in the C-terminal region. In addition, COL6A3 contains three unique C-terminal domains (C3–C5) that are absent from other collagen type VI families. The most C-terminal domain, C5, is cleaved into soluble endotrophin. The two amino acid sequences of COL6A3 targeted by the aptamers to measure COL6A3 levels are as follows: the N-terminal-binding aptamer targets the amino acid sequence

26–1036 (uncleaved section), whereas the C-terminal aptamer targets the amino acid sequence 3108–3165 (cleaved section). The figure has been modified from ref. 67,68. **c**, MR analysis of the effects of C-terminal and N-terminal COL6A3 on the risk of CAD. **d**, MR for the effects of BMI and body fat percentage on COL6A3 stratified by C- and N-terminal COL6A3. **e**, MR for the effects of body fat compartments on COL6A3 stratified by C- and N-terminal COL6A3. We used MRI-derived GWAS on abdominal subcutaneous adipose tissue, visceral adipose tissue and gluteofemoral adipose tissue from 40,032 individuals in the UK Biobank, reported by Agrawal et al.[69]. The two-sample MR method was as described in the step 1 MR analysis. In **c–e**, P values were obtained using the random-effects IVW method (two-sided test). Error bars represent the 95% CI for effect estimates. **a**, created using BioRender.

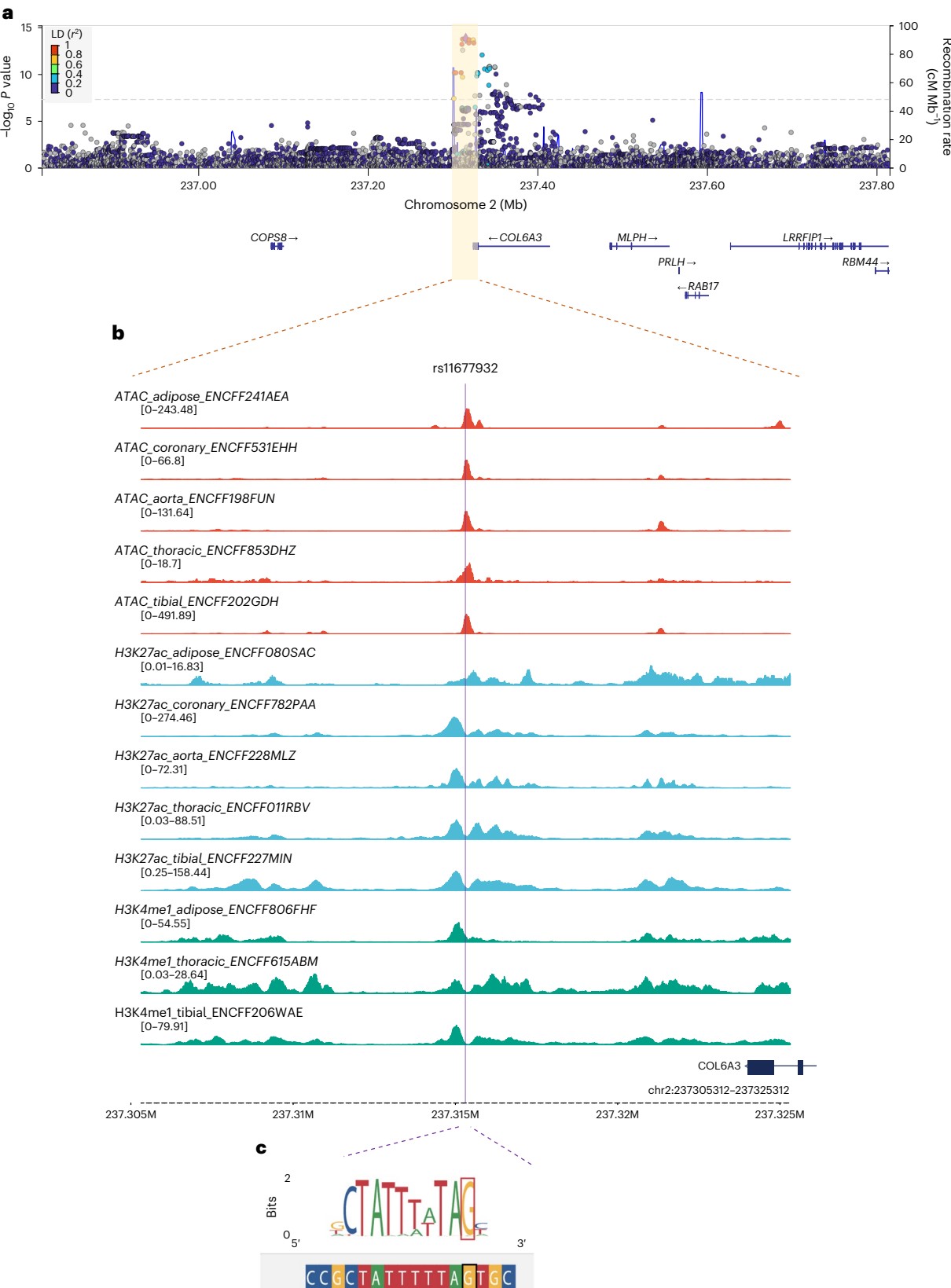

**Fig. 6 | Epigenetic profile of the lead *cis*-pQTL for C-terminal COL6A3.**
**a**, LocusZoom plot of the pQTL for C-terminal COL6A3 in the 1-Mb region surrounding the lead *cis*-pQTL from deCODE, rs11677932. The *y* axis on the left represents the −log$_{10}$ *P* value from the two-sided *Z* test. The yellow shaded region (chr2:237305312–237325312; GRCh38) is enlarged in **b**. **b**, ATAC-seq (red),

H3K4me3 ChIP–seq (blue) and H3K27ac ChIP–seq (green) data for adipose tissue, coronary artery, aorta, thoracic artery and tibial artery. These data are publicly available through ENCODE and RegulomeDB. **c**, rs11677932 is predicted to affect the binding of TF MEF2B. ENCODE accession ID: ENCSR782UOT; target: BORCS8-MEF2B, MEF2B.

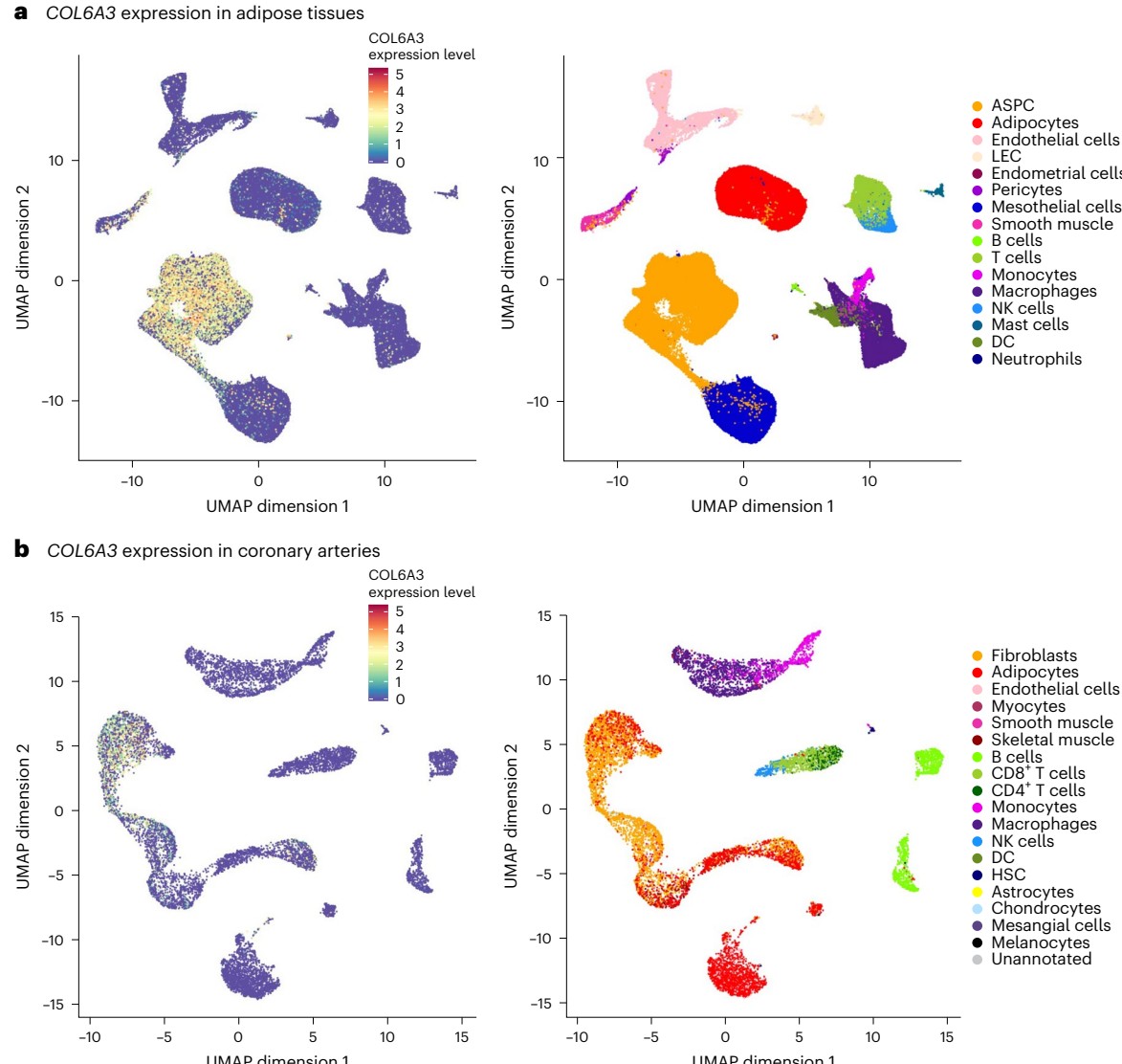

**Fig. 7 | Single-cell sequencing analyses of *COL6A3*. a,b,** *COL6A3* expression patterns in the adipose tissues (**a**) and coronary arteries (**b**). UMAP plots are colored by *COL6A3* expression (left) and cell type annotation (right). We obtained single-cell transcriptomic data for human adipose tissue from ref. 54 (SCP1376 at https://singlecell.broadinstitute.org/) and data for coronary arteries from ref. 53 (GSE131780 at the Gene Expression Omnibus database https://www.ncbi.nlm.nih.gov/geo/). ASPC, adipose stem and progenitor cells; LEC, lymphatic endothelial cells; NK, natural killer cells; DC, dendritic cells; HSC, hematopoietic stem cells.

In addition, sex-stratified Cox regression analysis for cumulative incidents of CAD in the UK Biobank showed that a standard deviation increase in plasma C-terminal COL6A3 levels was associated with an increased cumulative incidence of CAD in both females (HR = 1.15, 95% CI: 1.07–1.23, $P = 6.5 \times 10^{-5}$) and males (HR = 1.24, 95% CI: 1.18–1.30, $P < 2.2 \times 10^{-16}$).

### *COL6A3* expression analyses

We explored *COL6A3* expression using GTEx v.8, which includes data from 49 tissues across 838 individuals[52]. *COL6A3* was significantly expressed in several tissues, including adipose tissue and coronary arteries, compared with whole blood ($P < 0.001$) (Extended Data Fig. 4). This suggests that these tissues may locally produce COL6A3 and its cleavage product, endotrophin. Given that adipose tissue is a primary source of COL6A3, and coronary arteries are key sites in CAD[53], we further analyzed single-cell *COL6A3* expression in human white adipose tissues[54] and coronary arteries from patients with CAD[53].

In single-cell RNA sequencing, *COL6A3* expression was significantly enriched in adipose progenitor and stem cells of adipose tissues compared with other cell types (permutation $P < 0.001$) (Fig. 7a). Given that these cell populations have critical roles in maintenance of adipose tissue and metabolic function[55,56], this suggests that metabolic dysfunction may be a mechanism through which COL6A3 influences CAD. In addition, *COL6A3* was significantly expressed in fibroblasts, which are key players in coronary artery atherosclerosis[57], compared with other cell types in the coronary artery (permutation $P < 0.001$) (Fig. 7b). These findings suggest that these cell types may be responsible for local production of COL6A3 in these tissues.

### Assessment of actionability

The clinical relevance of the identified mediators depends on whether modification of these targets through weight loss or other methods can influence disease outcomes. We explored whether reducing fat mass or increasing lean mass could improve plasma levels of COL6A3-derived endotrophin and other proteins, thereby reducing the risk of cardiometabolic diseases. Using multivariable MR, we evaluated the independent effects of body fat and lean mass (that is, body fat-free mass) on the protein mediators and cardiometabolic disease outcomes.

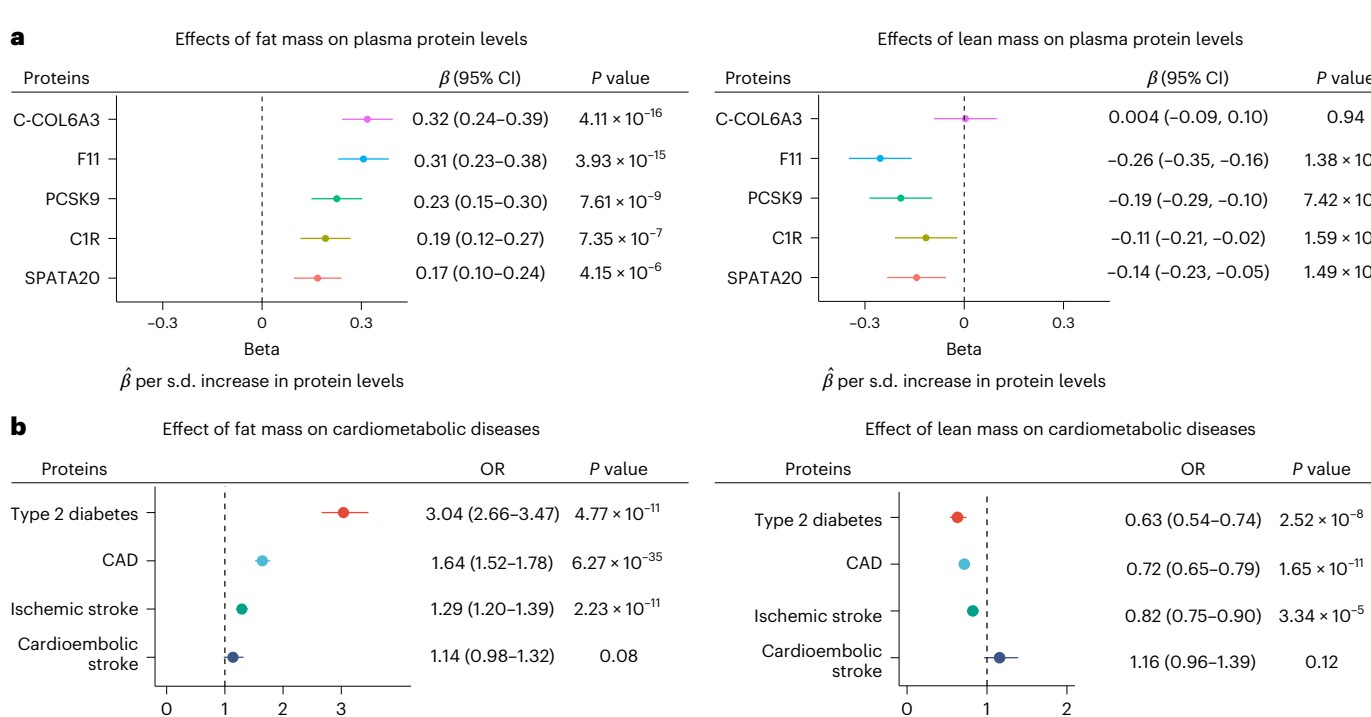

**Fig. 8 | Effects of fat mass and lean mass on proteins and cardiometabolic diseases. a,b**, We performed multivariable MR using fat mass (left) and lean mass (right) as exposures and plasma protein levels of the seven protein mediators (**a**) and cardiometabolic diseases (**b**) as outcomes. *P* values were obtained using the random-effects IVW method (two-sided test). Error bars represent the 95% CI for effect estimates.

We found that fat mass independently increased plasma levels of all protein mediators (COL6A3-derived endotrophin, F11, PCSK9, C1R and SPATA20) (Fig. 8a and Supplementary Table 18) and increased the odds of type 2 diabetes, CAD and ischemic stroke (Supplementary Table 19). Conversely, lean mass independently decreased plasma levels of some protein mediators including F11 and PCSK9 (Fig. 8b and Supplementary Table 18).

This has important clinical implications for actionability because interventions such as exercise, appropriate diet or weight-loss drugs such as GLP-1 receptor agonist semaglutide and GLP-1/GIP coagonist tirzepatide, which reduces body fat mass more than lean mass[58,59], could improve levels of these proteins and subsequently decrease the risk of cardiometabolic diseases. However, future clinical trials are needed to confirm this hypothesis.

Last, we evaluated the implications of reducing COL6A3-derived endotrophin through a phenome-wide association analysis. Given that some drug trials fail owing to unexpected adverse events[17,60], understanding the potential effects of perturbing the target on a phenome-wide level may help to anticipate possible adverse events. We assessed traits associated with the lead *cis*-pQTL of COL6A3 (rs11677932) using data from the UK Biobank, FinnGen and the GWAS catalog via the Open Target Genetics database[46] at $P < 1.0 \times 10^{-5}$. Lower plasma levels of COL6A3-derived endotrophin (A-allele of rs11677932; $\beta = -0.07$, $P = 1.5 \times 10^{-14}$) were associated with increased heel-bone mineral density ($\beta = 0.02$, $P = 2.9 \times 10^{-19}$) and increased lung function (FEV1/FVC) ($\beta = 0.02$, $P = 5.2 \times 10^{-13}$), in addition to a reduced risk of CAD ($\beta = -0.03$, $P = 2.7 \times 10^{-12}$) (Supplementary Table 20). This suggests that decreasing COL6A3-derived endotrophin levels may decrease the risk of multiple morbidities without apparent adverse events, making it an attractive therapeutic target.

## Discussion

Obesity is a major risk factor for various diseases, and therapies are required that reduce its clinical consequences. Here, we identified five protein mediators that partially mediate the effects of obesity on cardiometabolic diseases. Levels of these proteins, including COL6A3, could potentially be decreased through reduction in body fat; this indicates their potential clinical actionability. Furthermore, follow-up analyses suggested that endotrophin, the cleaved product of C-terminal COL6A3, drives the effect of obesity on CAD.

The main finding of this study is the mediating role of endotrophin in the effect of obesity on CAD. Previous studies have reported endotrophin as an important hormone that induces metabolic dysfunction, fibrosis and inflammation in rodent models[35,36,39,45], and increased circulating endotrophin levels have been observationally associated with cardiovascular events and all-cause mortality in cross-sectional studies in humans[37,38,40,61]. However, observational studies cannot distinguish cause and consequence. Our study provides evidence that endotrophin acts as a causal mediator in the relationship between obesity and CAD in humans. Given our finding that reducing levels of COL6A3 and its cleaved product endotrophin can reduce the risk of CAD without apparent adverse health outcomes, directly targeting endotrophin could be an attractive therapeutic approach that may be particularly effective in individuals with obesity.

Several rodent studies have shown that bone morphogenetic protein 1 (BMP1)[39], matrix metallopeptidase 14 (MMP14)[62] and other MMPs[43] can release the C terminus of COL6A3 as endotrophin after proteolytic cleavage. Inhibition of BMP1 reduces scar formation and supports the survival of cardiomyocytes[63], possibly owing in part to lower levels of endotrophin. Nevertheless, BMP1 also cleaves other procollagens into mature collagens, which introduces pleiotropy. Therefore, more research is necessary to determine how to selectively inhibit the cleavage of the C terminus of COL6A3 to reduce endotrophin levels. Our study also replicated findings regarding other proteins, including PCSK9 and F11; these are discussed further in Supplementary Note 4.

This study had several limitations. First, our analyses focused on individuals of European genetic ancestry. Currently, the largest

*cis*-pQTL dataset for individuals of African genetic ancestry[15] is much smaller than that for those of European genetic ancestry ($n = 1,871$ versus $n = 35,559$), and the same applies to CAD GWAS (17,247 cases[64] versus 181,522 cases[33]), limiting statistical power. Second, we refrained from emphasizing the proportion by which C-terminal COL6A3 mediates the causal effect of BMI on CAD risk, given the challenges associated with using *cis*-MR mediation analysis for this estimation[65] (see Supplementary Note 5). Third, some *cis*-pQTL may be affected by epitope effects rather than true protein level changes, especially when the *cis*-pQTLs are protein-altering variants (PAVs) or in high LD with PAVs and lack associated eQTL[66]. However, this was not the case for rs11677932. The variant was not a PAV nor in high LD with any PAVs, and the variant was also a *cis*-eQTL. Finally, although we triangulated multiple lines of evidence, future clinical trials are required to explore the effect of pharmacologically influencing the identified protein levels.

In conclusion, by integrating two-step proteome-wide MR, colocalization, observational assessments, epigenomics, and single-cell RNA sequencing, we identified five actionable mediators of obesity's effect on cardiometabolic diseases and prioritized COL6A3-derived endotrophin as a therapeutic target to reduce CAD risk. This two-step framework can be generalized to identify some of the molecular mechanisms whereby risk factors cause disease in humans.

## Online content

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

[1]Department of Human Genetics, McGill University, Montréal, Québec, Canada. [2]Lady Davis Institute, Jewish General Hospital, McGill University, Montréal, Québec, Canada. [3]Canada Excellence Research Chair in Genomic Medicine, Victor Phillip Dahdaleh Institute of Genomic Medicine, McGill University, Montréal, Québec, Canada. [4]Kyoto-McGill International Collaborative Program in Genomic Medicine, Graduate School of Medicine, Kyoto University, Kyoto, Japan. [5]Programs in Metabolism and Medical & Population Genetics, The Broad Institute of MIT and Harvard, Cambridge, MA, USA. [6]Department of Statistical Sciences, University of Toronto, Toronto, ON, Canada. [7]Department of Population Health Sciences, School of Medicine and Public Health, University of Wisconsin-Madison, Madison, WI, USA. [8]Department of Biostatistics and Medical Informatics, School of Medicine and Public Health, University of Wisconsin-Madison, Madison, WI, USA. [9]Division of Infectious Diseases, McGill University Health Centre, Montréal, Québec, Canada. [10]Centre for Human Genetics, University of Oxford, Oxford, UK. [11]MRC Epidemiology Unit, Institute of Metabolic Science, University of Cambridge, Cambridge, UK. [12]Precision Healthcare University Research Institute, Queen Mary University of London, London, UK. [13]Quantitative Life Sciences Program, McGill University, Montréal, Québec, Canada. [14]5 Prime Sciences, Montréal, Québec, Canada. [15]Department of Anatomic Pathology and Laboratory Medicine, New York Presbyterian - Weill Cornell Medical Center, New York, NY, USA. [16]SomaLogic, Boulder, CO, USA. [17]Program in Medical and Population Genetics, Broad Institute of MIT and Harvard, Cambridge, MA, USA. [18]Cardiovascular Research Center and Center for Genomic Medicine, Massachusetts General Hospital, Boston, MA, USA. [19]Fulcrum Genomics, Somerville, MA, USA. [20]Department of Physiology and Pharmacology, Karolinska Institutet, Stockholm, Sweden. [21]Max Planck Institute for Evolutionary Anthropology, Leipzig, Germany. [22]Anaesthesiology and Intensive Care Medicine, Department of Surgical Sciences, Uppsala University, Uppsala, Sweden. [23]Integrative Physiology, Department of Medical Cell Biology, Uppsala University, Uppsala, Sweden. [24]Division of Cancer Epidemiology and Genetics, National Cancer Institute, Rockville, MD, USA. [25]Type 2 Diabetes Systems Genomics Initiative, The Broad Institute of MIT and Harvard, Cambridge, MA, USA. [26]Novo Nordisk Foundation Center for Genomic Mechanisms of Disease, The Broad Institute of MIT and Harvard, Cambridge, MA, USA. [27]Center for Genomic Medicine and Endocrine Division, Massachusetts General Hospital, Boston, MA, USA. [28]Department of Medicine, Harvard Medical School, Boston, MA, USA. [29]Division of Genetics and Genomics, Boston Children's Hospital, Boston, MA, USA. [30]Department of Pediatrics, Harvard Medical School, Boston, MA, USA. [31]Integrative Epidemiology Unit, University of Bristol, Bristol, UK. [32]Population Health Sciences, Bristol Medical School, University of Bristol, Bristol, UK. [33]Computational Medicine, Berlin Institute of Health (BIH) at Charité – Universitätsmedizin Berlin, Berlin, Germany. [34]Department of Epidemiology, Biostatistics and Occupational Health, McGill University, Montréal, Québec, Canada. [35]Department of Twin Research, King's College London, London, UK. ✉e-mail: satoshi.yoshiji@mcgill.ca; brent.richards@mcgill.ca

## Methods

### Ethics

All contributing cohorts obtained ethical approval from their intuitional ethics review boards. The contributing cohorts included the UK Biobank, GIANT consortium, deCODE study, Fenland study, ARIC study, INTERVAL study, CARDIoGRAMplusC4D, GIGASTROKE and MAGIC consortium. For individual-level data, the study was approved by the UK Biobank (application number: 27449) and the Norfolk Research Ethics Committee (no. 05/Q0101/191), and all participants gave their informed written consent. For human adipose-derived mesenchymal stem cells used in LipocyteProfiler, each participant gave written informed consent before inclusion, and the study protocol was approved by the ethics committee of the Technical University of Munich (study no. 5716/13) and the Broad Institute of MIT and Harvard (IRB number: ORSP-1613).

### MR step 1

**MR to evaluate the effects of BMI on plasma protein levels.** We performed two-sample MR using BMI as the exposure and circulating protein levels as outcomes. The BMI GWAS exposure data came from a meta-analysis of the UK Biobank and GIANT consortium involving 693,529 individuals of European ancestry[23] (Supplementary Table 1). For the outcomes, we used GWAS of protein levels from deCODE[14], measuring 4,907 proteins in 35,559 individuals of European ancestry using SomaScan assay v.4 from SomaLogic. The aptamer seqIDs can be queried at https://menu.somalogic.com/.

Instrumental variables were genome-wide significant and independent single-nucleotide polymorphisms (SNPs) with $P < 5 \times 10^{-8}$ and $r^2 < 0.001$. We excluded SNPs in the human major histocompatibility complex region because of their complex LD structures. Clumping was performed using PLINK v.1.9 (https://www.cog-genomics.org/plink/) with a 10-Mb window. When the instrumental variable SNPs were not present in the outcome GWAS, we identified proxy SNPs with $r^2 \geq 0.8$ using snappy v.1.0 (https://gitlab.com/richards-lab/vince.forgetta/snappy/). Two-sample MR was conducted using the random-effects IVW method for the primary analysis, implemented using TwoSampleMR v.0.5.6. We set $P < 1 \times 10^{-5}$ (0.05/4,907; Bonferroni correction) as a stringent threshold for significance.

Importantly, MR relies on three key assumptions[6,7]: that there exist genetic variants that (1) are associated with the risk factor of interest (relevance); (2) are not correlated with confounders of the instrument–outcome relationship (independence); and (3) affect the outcome only through the exposure (exclusion restriction, also known as lack of horizontal pleiotropy). To reduce the risk of weak instrument bias, which violates the relevance assumption, we calculated F-statistics and evaluated whether they were greater than 10 (refs. [70,71]) (Supplementary Table 2). Further, to safeguard against the violation of these assumptions, we conducted multiple sensitivity analyses.

**Sensitivity analyses.** We tested for heterogeneity, directional horizontal pleiotropy (MR-Egger intercept test), outlier-robust estimate (MR-PRESSO outlier-corrected estimate, weighted median estimate and MR-Egger slope estimate), reverse causation and directional concordance with body fat percentage. Heterogeneity was assessed using the $I^2$ statistic, with $I^2 > 50\%$ indicating substantial heterogeneity. The MR-Egger intercept was used to test for directional horizontal pleiotropy, with $P < 0.05$ indicating its presence.

For outlier-robust estimates, we used MR-weighted median, MR-Egger slope and MR-PRESSO outlier-corrected estimates as supplementary analyses to evaluate the directional concordance of the effects. Proteins were required to show a directionally consistent effect across all methods, including IVW; otherwise, they were removed. MR-PRESSO outlier-corrected estimates were used only if outliers were detected.

For reverse MR, wherein we examined the effects of plasma protein levels on BMI, we used *cis*-pQTL from deCODE and BMI GWAS from UK

Biobank, as in previous work[22]. We used the IVW method or the Wald ratio method when only one SNP was available. We used Bonferroni correction as a threshold for significance ($P < 1.39 \times 10^{-4}$).

To assess directional concordance with body fat percentage, we performed two-sample MR using body fat percentage as exposure and plasma protein levels as outcomes. We used GWAS of body fat percentage in 454,633 individuals of European ancestry from the UK Biobank (accession ID: ukb-b-8909 at IEU OpenGWAS project) and protein levels for GWAS from deCODE. Proteins were required to achieve $P < 0.05$ in the MR and show consistent directional effects across BMI and body fat percentage to pass the test; otherwise, they were removed.

### MR step 2

**MR for the effects of BMI-driven proteins on disease outcomes.** We performed two-sample MR using circulating protein levels as exposures and cardiometabolic diseases as outcomes. We used *cis*-pQTL from deCODE[14] as the instrumental variables. The *cis*-pQTL was defined as a pQTL located within 1 Mb of the TSS of the corresponding protein-coding gene. For the outcome, we used GWAS of CAD[25] (181,522 CAD cases and 1,165,690 controls), ischemic stroke (62,100 cases; 1,234,808 controls), cardioembolic stroke[26] (10,804 cases; 1,234,808 controls) and type 2 diabetes[27] (80,154 cases and 853,816 controls). After data harmonization and proxy search, we retained 348, 319, 303 and 326 proteins for each outcome, respectively. To estimate the causal effect, we used random-effects IVW or Wald ratio when only one SNP was available as an instrumental variable. Bonferroni correction was applied for the number of proteins tested in each MR analysis. To minimize the risk of horizontal pleiotropy, we removed variants associated with more than one protein in a *cis*-acting manner; therefore, we only retained the variants that were *cis*-pQTL for one protein (7,008 of 7,572 variants). To further test the absence of directional horizontal pleiotropy, we used the MR-Egger intercept test when applicable (that is, if there were at least three instrumental variables). In addition, we used the MR-Steiger test from TwoSampleMR v.0.5.6 to assess reverse causation, whereby cardiometabolic diseases influence plasma levels of proteins.

**Assessment of bias due to sample overlap in two-sample MR.** Relative bias, which quantifies the extent to which causal estimates in MR are biased owing to sample overlap relative to the observational estimate[72], was calculated as:

$$\text{relative bias} = \phi \times \frac{1}{F},$$

where $\phi$ is the proportion of the sample overlap (ranging from 0 to 1), and $F$ is the $F$-statistic of the exposure. We calculated the relative bias for MR steps 1 and 2. Further details are available in Supplementary Notes 1 and 2. The STROBE-MR checklist, which is a framework for reporting MR studies, can be found in Supplementary Note 6.

**Colocalization.** To ensure that the proteins and cardiometabolic diseases shared a causal genetic signal and to avoid false-positive findings, we also performed colocalization using the coloc R package (v.5.1.0)[73]. We evaluated whether *cis*-pQTL of the protein had the same causal variant as cardiometabolic diseases within a 500-kb region, using default priors of $p_1 = 10^{-4}$, $p_2 = 10^{-4}$ and $p_{12} = 10^{-5}$ for coloc, where $p_1$ is the prior probability of trait 1 having a genetic association in the region, $p_2$ is the prior probability of trait 2 having a genetic association in the region and $p_{12}$ is the prior probability of the two traits having a shared genetic association. We considered the posterior probability of a shared causal variant ($\text{PP}_{\text{shared}}$) > 0.8 as evidence of colocalization.

### Follow-up analyses

**Replication MR using *cis*-pQTL from different cohorts.** To replicate the causal estimates for the effects of COL6A3 on CAD, we conducted

two-sample MR using published *cis*-pQTLs from different cohorts—the UK Biobank[31] (*n* = 34,557 individuals), Fenland[13] (*n* = 10,708) and ARIC (*n* = 7,213)—using the method described for MR step 2. For the CAD outcome, we used the same CAD GWAS[33] as in the primary analyses.

**Observational analysis in the EPIC-Norfolk cohort.** EPIC-Norfolk study is a cohort in Eastern Englnad[74], and proteomic profiling was performed using the SomaScan v.4 assay for *n* = 872 individuals (Supplementary Note 7). Participants were identified as CAD cases if the corresponding ICD-codes (ICD-9: 410–414, ICD-10: I20–I25) were registered on the death certificate or as the cause of hospitalization. The case definition included all individuals identified as prevalent (at the baseline study assessment) or incident CAD cases over the follow-up period of more than 20 years. The plasma protein levels were normalized with rank-based inverse normal transformation using R package RNOmni (v.1.01). We used linear regression, adjusting for age and sex, to estimate the association between BMI and plasma COL6A3 levels. We used logistic regression, adjusting for age and sex, to evaluate the association between BMI and CAD risk as well as that between COL6A3 levels and CAD risk. Results with *P* < 0.025 (0.05/2; Bonferroni correction) were considered to be statistically significant.

**Observational analysis in the UK Biobank.** *BMI and C-terminal COL6A3.* We performed multivariable linear regression analysis to evaluate the associations between baseline BMI and plasma C-terminal COL6A3 levels in 35,100 individuals from the UK Biobank. We included participants from the UK Biobank for whom we had protein measurements obtained with the Olink Explore 3072 assay (UK Biobank data field: 30900) and ICD-10-based diagnosis (data field: 41270). For covariates, we included age at recruitment (data field: 21022), sex (data field: 31), recruitment center (data field: 54), Olink measurement batch (resource: 1016) and Olink processing time (resource: 1016). The outcome was plasma COL6A3 levels, which were rank-based normal transformed using the RankNorm() function from the RNOmni v1.0.1.2 R package. A result was considered significant if BMI achieved *P* < 0.05.

*C-terminal COL6A3 and CAD risk.* Using multivariable Cox proportional-hazards regression analysis, we tested whether baseline plasma C-terminal COL6A3 levels were associated with the cumulative incidence of CAD, adjusting for the same covariates (age, sex, recruitment center, Olink measurement batch and Olink processing time). CAD was defined by (1) a record of I20–I25 (ischemic heart disease) in ICD-10, (2) an operation record of percutaneous transluminal coronary angioplasty or coronary artery bypass grafting, or (3) a record of death due to I20–I25. The time to event was calculated by subtracting the date of event registry from the date of enrollment (data field: 53), focusing on events that occurred within 10 years from enrollment. We excluded prevalent CAD cases who met the above criteria before enrollment and those whose date of event was not recorded. Controls included those without a record of CAD based on doctor diagnosis (data field: 6150), self-reported heart attack (data field: 20002) or ICD-10 record of I20–I25.

Cox proportional-hazards regression analysis was performed using the coxph() function from the survival v.3.5.8 R package. A result was considered significant if the C-terminal COL6A3 level achieved *P* < 0.05. For plotting Kaplan–Meier curves, participants were stratified into four groups based on baseline plasma C-terminal COL6A3 levels (25%, 50%, 75% and 100% quantiles). We used the log-rank test to evaluate statistically significant differences in survival curves. A result with *P* < 0.05 was considered statistically significant.

**Identification of the causal domain of COL6A3.** *Target region of the SomaScan and Olink Explore assays.* We used SomaScan Menu (https://menu.somalogic.com/) to determine the target amino acid sequence of two aptamers for COL6A3 from a SomaScan v.4 assay with additional support from SomaLogic. We also obtained data on the target region

of Olink Explore 3072 assay from Olink Proteomics. In the SomaScan v.4 assay, two aptamers targeted COL6A3: one targeted its C-terminal, also known as the Kunitz domain (UniProt ID: P12111; target amino acid sequence: 3108–3165), and the other targeted the N-terminal (UniProt ID: P12111; target amino acid sequence: 26–1036). The seqIDs for the C-terminal- and N-terminal-targeting aptamers were 11196-31 and 10511-10, respectively. The Olink Explore 3072 assay targeted the C-terminal Kunitz domain of COL6A3 with a polyclonal antibody (OID20292:v1).

**LD evaluation of *cis*-pQTL of COL6A3.** We used the LDmatrix tool available at LDlink (https://ldlink.nci.nih.gov) with European samples from the 1000 Genomes Project as the reference panel[75] to calculate $R^2$ values between three SNPs: the *cis*-pQTL for COL6A3 from UK Biobank (rs1050785), the *cis*-pQTL of the C-terminal-targeting aptamer (rs11677932) from deCODE, and the *cis*-pQTL of the N-terminal-targeting aptamer of COL6A3 (rs2646260) from deCODE.

**MR for body fat compartments and of C- and N-terminal COL6A3.** We performed two-sample MR using MRI-derived body fat compartment volumes as exposures and plasma levels of C- and N-terminal COL6A3 from deCODE as outcomes. For the body fat compartment measurements, we used MRI-derived GWAS for abdominal subcutaneous adipose tissue, visceral adipose tissue and gluteofemoral adipose tissue of 40,032 individuals in the UK Biobank that had been reported by Agrawal et al.[69]. We used pQTL from deCODE[14] for the outcomes. The two-sample MR method was as described in the MR step 1 analysis. We set a significance threshold of $P < 8.3 \times 10^{-3}$ (0.05/6; Bonferroni correction).

**Fine-mapping using SuSiE.** We performed fine-mapping with susieR v.0.11.92 using European samples from the 1000 Genomes Project as the reference panel[75]. We set the maximum number of causal variants to 5 and evaluated whether the lead *cis*-pQTL for COL6A3 (rs11677932; the variant with the smallest *P* value) was in the 95% credible set for the *cis*-pQTL of C-terminal COL6A3 and for CAD GWAS within the 500-kb region surrounding rs11677932 (±250 kb).

**V2G mapping of the *cis*-pQTL of C-terminal COL6A3.** We downloaded the Open Target Genetics V2G database (v.22.02.01) from the EMBL-EBI FTP site (https://ftp.ebi.ac.uk/pub/databases/opentargets/genetics/). We then queried the database for rs11677932, the lead *cis*-pQTL of C-terminal COL6A3, and assessed which gene the variant was mapped to with the highest V2G score.

**Regulatory role of the *cis*-pQTL of C-terminal COL6A3.** RegulomeDB is a database that facilitates the interpretation of noncoding variants by integrating functional genomic assays and computational approaches[50,51]. We queried the database for rs11677932 (the lead *cis*-pQTL from deCODE) to evaluate whether there was supporting evidence for a role of the variant as a regulatory variant. We used assay for transposase-accessible chromatin using sequencing (ATAC-seq), histone H3 K27 acetylation (H3K27ac) chromatin immunoprecipitation with sequencing (ChIP–seq) and H3K4 monomethylation (H3K4me1) ChIP–seq datasets from ENCODE and Regulome (Supplementary Table 21). For visualization, we downloaded BigWig files for the corresponding data and plotted a ±50-kb region surrounding rs11677932 (chr2:237315312) using the trackplot R v.1.0 package (https://github.com/PoisonAlien/trackplot). The TF-binding motif was visualized with RegulomeDB (ENCODE accession ID for the corresponding data: ENCSR782UOT; target: BORCS8-MEF2B, MEF2B). We also queried GTEx v.8 (ref. 52) to evaluate whether rs11677932 was an eQTL in any tissues (https://gtexportal.org/home/snp/rs11677932).

**Sex-stratified analyses of C-terminal COL6A3.** *Sex-stratified pQTL for C-terminal COL6A3.* To generate sex-stratified pQTL for C-terminal COL6A3, we used data from 19,747 females and 16,876 males from the

UK Biobank (data field 30900; category 1839). We performed GWAS following the methods described in the UK Biobank Pharma Proteomics Project (UKB-PPP) flagship paper[76] with minor modifications: proteomics data were preprocessed and underwent quality control by the UKB-PPP; and GWAS was performed using REGENIE v.3.2.9 (ref. 77), as described in Supplementary Note 8.

### Sex-stratified MR step 1
We performed two-sample MR to evaluate the causal effects of BMI on plasma levels of C-terminal COL6A3 in a sex-stratified manner. For BMI GWAS, we used sex-stratified GWAS from the GIANT consortium in 73,137 females and 60,586 males[78]. We performed two-sample MR as described previously in the MR step 1 section in males and in females.

### Sex-stratified MR step 2
To identify instrumental variables of C-terminal COL6A3, we performed clumping for the pQTL generated in the above step. We used PLINK v.1.9 with a 10-Mb window in the cis-region of COL6A3 with the same criteria as in other steps ($P < 5 \times 10^{-8}$ and $r^2 < 0.001$). Then, we performed cis-MR using C-terminal COL6A3 as the exposure and CAD risk as the outcome. We used the Wald ratio method to estimate the effect. Furthermore, we tested whether the causal variant was shared between males and females using colocalization. A posterior probability of having the shared causal variant (hypothesis H4 in colocalization) greater than 0.8 was considered to be strong evidence of colocalization (see 'Colocalization' for further details of the methods used).

### Sex-stratified Cox regression analysis for CAD risk.
We repeated the Cox regression described above ('Cox regression analysis for cumulative incidence of CAD'), stratifying participants based on sex (data field: 31). We considered $P < 0.025$ (0.05/2) to indicate statistical significance.

### COL6A3 expression analyses.
We downloaded bulk gene expression data in human tissues (GTEx_Analysis_2017-06-05_v8_RNASeQCv1.1.9_gene_tpm.gct.gz) from the GTEx portal (https://gtexportal.org/). We generated violin plots for COL6A3 expression levels in each tissue using R v.4.1.2 and used a two-sided Wilcoxon rank sum test to compare COL6A3 expression in each tissue with its expression in whole blood.

### Single-cell RNA sequencing analysis.
To investigate COL6A3 expression at single-cell resolution in adipose tissues and coronary arteries, we reanalyzed the published expression matrix data from ref. 54 (SCP1376 at https://singlecell.broadinstitute.org/) and ref. 53 (GSE131780 at the Gene Expression Omnibus database; https://www.ncbi.nlm.nih.gov/geo/), focusing on COL6A3 expression. The quality control procedure used for the dataset is described in Supplementary Note 9. We used SingleR v.2.0.0 to annotate the cell clusters, with the Blueprint/ENCODE dataset as the reference, using default settings. To assess whether certain cell types expressed COL6A3 more significantly than others, we performed 1,000 permutations of the cell type labels and calculated the frequency (permutation $P$ value) of the same cell type containing the same or a larger proportion of cells expressing COL6A3 compared with all cells.

### Follow-up analyses for the identified proteins
#### Assessment of actionability.
To estimate the independent effects of fat mass and lean mass on plasma protein levels, we performed multivariable MR using fat mass and lean mass as exposures and protein levels as outcomes.

### GWAS of fat mass and lean mass.
We retrieved GWAS data for fat mass and lean mass (that is, fat-free mass) from the UK Biobank through the OpenGWAS portal (https://gwas.mrcieu.ac.uk/), including data for 454,137 individuals of European ancestry for fat mass and 454,850 individuals for lean mass. The accession codes for the datasets were ukb-b-19393 for fat mass and ukb-b-13354 for lean mass. The fat mass and fat-free mass of the UK Biobank participants (second release, 2017) were evaluated by the UK Biobank with bioelectrical impedance analysis using a Tanita BC418MA body composition analyzer.

### Multivariable MR with fat mass and lean mass.
To obtain instrumental variables, we applied the same selection criteria as in steps 1 and 2 of MR ($P < 5 \times 10^{-8}$ and $r^2 < 0.001$), excluding those in the major histocompatibility complex region (GRCh37; chr6:28477797–33448354). We performed data harmonization in TwoSampleMR v.0.56 and multivariable MR with the IVW method and a random-effect model in MVMR v.0.3 (ref. 71). We calculated conditional F-statistics using MVMR v.0.3 (ref. 71) and evaluated whether they were greater than 10 (refs. 70,71). The phenotypic correlation matrix was calculated using metaCCA v.1.22.0 (ref. 79). For additional sensitivity analyses, we performed multivariable MR-Egger analysis using MendelianRandomization v.0.6.085 (ref. 80).

### Phenome-wide association study for rs11677932.
We queried traits associated with the lead cis-pQTL of COL6A3 (rs11677932) from deCODE in the UK Biobank, FinnGen and GWAS catalog using Open Target Genetics (https://genetics.opentargets.org/).

### Statistics and reproducibility
No statistical method was used to predetermine the sample size. No randomization or blinding was conducted. No data were excluded from the analyses unless stated otherwise.

### Reporting summary
Further information on research design is available in the Nature Portfolio Reporting Summary linked to this article.

## Data availability
The GWAS of plasma COL6A3 levels in males and females are available at the GWAS Catalog (GCP ID: GCP001023). We used publicly available GWAS summary statistics from the following sources: BMI GWAS from GIANT and UK Biobank (https://portals.broadinstitute.org/collaboration/giant/; https://doi.org/10.1093/hmg/ddy271); and plasma proteome GWAS from deCODE (https://www.deCODE.com/summarydata/; https://doi.org/10.1038/s41588-021-00978-w), UK Biobank (https://www.ukbiobank.ac.uk/; https://doi.org/10.1101/2022.06.17.496443), Fenland (https://omicscience.org/apps/pgwas/; https://doi.org/10.1126/science.abj1541) and ARIC (http://nilanjanchatterjeelab.org/pwas/; https://doi.org/10.1038/s41588-022-01051-w). We also used the CAD GWAS from CARDIoGRAMplusC4D (http://www.cardiogramplusc4d.org/; https://doi.org/10.1038/s41588-022-01233-6), stroke GWAS from GIGASTROKE (GCST90104534 and GCST90104535, at https://www.ebi.ac.uk/gwas/studies/) and type 2 diabetes GWAS from Mahajan et al. (https://doi.org/10.1038/s41588-022-01058-3)[27]. For gene expression data, we used data from Nathan et al. (SCP498 at the Single Cell Portal; https://singlecell.broadinstitute.org/) and Wirka et al. (GSE131780 at the Gene Expression Omnibus database; https://www.ncbi.nlm.nih.gov/geo/). Epigenomic data are available at RegulomeDB (https://regulomedb.org/)[51] and ENCODE (https://www.encodeproject.org/)[49]. V2G scores are available at Open Target Genetics (https://genetics-docs.opentargets.org/data-access/data-download). Individual-level data of the UK Biobank, EPIC-Norfolk and CellGenBankCohort[81] are available through the respective parties upon agreement. Source data are provided with this paper.

## Code availability
We used R v.4.1.2 (https://www.r-project.org/), TwoSampleMR v.0.5.6 (https://mrcieu.github.io/TwoSampleMR/), snappy v.1.0 (https://gitlab.com/richards-lab/vince.forgetta/snappy), coloc v.5.1.0 (https://chr1swallace.github.io/coloc/), PLINK v.1.9 (https://www.cog-genomics.org/plink/), susieR v.0.11.92 (https://github.com/stephenslab/susieR/),

REGENIE v.3.2.9 (https://github.com/rgcgithub/regenie/), SingleR v.2.0.0 (https://github.com/dviraran/SingleR) and Seurat v.4.0.6 (https://satijalab.org/seurat/). Custom code is available via GitHub (https://github.com/satoshi-yoshiji/cardiometab_proteogenomics/)[82]. Further details about the code is available from the corresponding authors upon request.

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

## Acknowledgements

The Richards research group is supported by the Canadian Institutes of Health Research (CIHR: 365825, 409511, 100558, 169303), the McGill Interdisciplinary Initiative in Infection and Immunity (MI4), the Lady Davis Institute of the Jewish General Hospital, the Jewish General Hospital Foundation, the Canadian Foundation for Innovation, the NIH Foundation, Cancer Research UK, Genome Québec, the Public Health Agency of Canada, McGill University, Cancer Research UK (grant number C18281/A29019) and the Fonds de Recherche Québec Santé (FRQS). J.B.R. is supported by an FRQS Mérite Clinical Research Scholarship. Support from Calcul Québec and Compute Canada is acknowledged. TwinsUK is funded by the Welcome Trust, Medical Research Council, European Union, the National Institute for Health Research (NIHR)-funded BioResource, Clinical Research Facility and Biomedical Research Centre based at Guy's and St Thomas' NHS Foundation Trust in partnership with King's College London. M.J.M. is supported by the Intramural Research Program of the National Cancer Institute. R.K., M.C., H.D. and M.C. are supported in part by the Novo Nordisk Foundation (NNF21SA0072102). N.J.T. is a Wellcome Trust Investigator (202802/Z/16/Z) and the principal investigator of the Avon Longitudinal Study of Parents and Children (MRC & WT 217065/Z/19/Z), is supported by the University of Bristol NIHR Biomedical Research Centre (BRC-1215-2001) and the MRC Integrative Epidemiology Unit (MC_UU_00011/1) and works within the CRUK Integrative Cancer Epidemiology Programme (C18281/A29019). The Genotype-Tissue Expression (GTEx) Project was supported by the Common Fund of the Office of the Director of the National Institutes of Health (NIH) and by the NCI, NHGRI, NHLBI, NIDA, NIMH and NINDS. The data used for the analyses described in this manuscript were obtained from the GTEx Portal on 26 March 2023. S.Y. is supported by the Japan Society for the Promotion of Science and McGill University. T.L. is supported by a Schmidt AI in Science Postdoctoral Fellowship, a Vanier Canada Graduate Scholarship, an FRQS doctoral training fellowship and a McGill University Faculty of Medicine Studentship. G.B.L. is supported by scholarships from the FRQS, the CIHR and Québec's ministry of health and social services. Y.C. is supported by an FRQS doctoral training fellowship and the Lady Davis Institute/ TD Bank Studentship Award. C.-Y.S. is supported by a CIHR Canada Graduate Scholarship Doctoral Award, an FRQS doctoral training fellowship and a Lady Davis Institute/TD Bank Studentship Award. K.L. is supported by a CIHR Canada Graduate Scholarship Doctoral Award. T.S. is supported by the JSPS Promotion of Joint International Research (23KK0301). S.K. is supported by the Japan Society for the Promotion of Science (202160643), Uehara Memorial Foundation and NIH, National Heart Lung and Blood Institute (NHLBI, K99HL169733). V.M. is holder of a Canada Excellence Research Chair (CERC) jointly funded by the federal TIPS agency and McGill University. The funders had no role in study design, data collection and analysis, decision to publish or preparation of the manuscript.

## Author contributions

S.Y. and J.B.R. designed the study. S.Y., T.Y., J.C.Z.S., R.K. and H.D. collected the data and conducted the analyses. S.Y. and J.B.R. supervised the study. G.B.L., J.C.Z.S., C.-Y.S., Y.C., K.L., J.D.S.W., S.W., D.A., Y.I., T.S., S.K., T.N., V.F., Y.F., H.Z., S.Z., M.H., M.M., R.K., H.D., M.C., J.F., N.J.W., V.M., N.J.T. and C.L. provided critical input and contributed to discussions and revisions of the manuscript.

## Competing interests

J.B.R. has served as an advisor to GlaxoSmithKline and Deerfield Capital. J.B.R.'s institution has received investigator-initiated grant funding from Eli Lilly, GlaxoSmithKline, and Biogen for projects unrelated to this research. J.B.R. is the CEO of 5 Prime Sciences (www.5primesciences.com), which provides research services for biotech, pharma and venture capital companies for projects unrelated to this research. T.L., Y.C. and V.F. are employees of 5 Prime Sciences. The remaining authors declare no competing interests.

## Additional information

**Extended data** is available for this paper at https://doi.org/10.1038/s41588-024-02052-7.

**Correspondence and requests for materials** should be addressed to Satoshi Yoshiji or J. Brent Richards.

**(a) Step 1 MR filtering**

| **4,907 plasma proteins** |
| :---: |

↓

| •Significance test: MR with the inverse variance weighted (IVW) method to assess the causal effect of BMI on 4,907 proteins<br>•Proteins must achieve $P < 1.0 \times 10^{-5}$ (Bonferroni correction: 0.05/4,907) |
| :---: |

↓

| **1,308 proteins passed the significance test** |
| :---: |

↓

| •Test of heterogeneity in causal estimates across instrumental variables with $I^2$ statistics<br>•Proteins must have $I^2 < 50\%$ |
| :---: |

↓

| **1,308 proteins passed the heterogeneity test** |
| :---: |

↓

| •Directional horizontal pleiotropy test with the MR-Egger intercept test<br>•Proteins must have $P > 0.05$ |
| :---: |

↓

| **1,234 proteins passed directional horizontal pleiotropy test** |
| :---: |

↓

| •Directional concordance test: MR estimates using outlier-robust methods include the MR weighted median, MR-Egger slope, and MR-PRESSO outlier-corrected estimates.<br>•Proteins must show a directionally consistent effect across all methods, including the IVW |
| :---: |

↓

| **1,234 proteins passed directionally consistent estimates across all methods** |
| :---: |

↓

| •Reverse causation test: MR for the effect of the plasma protein levels on BMI<br>•Proteins must not significantly influence BMI (as described previously in Yoshiji et al. *Nat. Metab.* 2023) |
| :---: |

↓

| **1,221 proteins passed the reverse causation test** |
| :---: |

↓

| •Directional concordance effect across BMI and body fat percentage: MR to assess the effect of body fat percentage and BMI on plasma protein levels<br>•Proteins must achieve $P < 0.05$ in the MR for the effect of body fat percentage and show consistent directional effects (i.e., same beta sign as in MR with BMI) |
| :---: |

↓

| **1,213 proteins showed directionally consistent estimates across BMI and body fat percentage** |
| :---: |

**(b) Step 2 MR filtering:**

The pQTL variants that are associated with more than one protein in *cis* were removed.

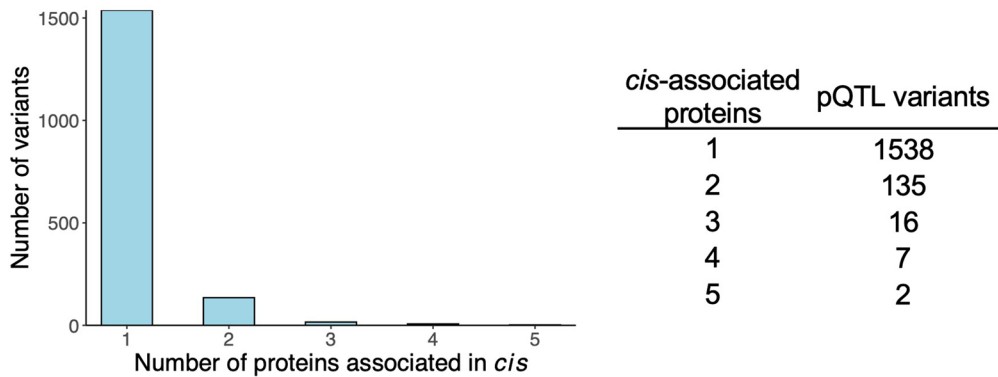

| *cis*-associated proteins | pQTL variants |
| :---: | :---: |
| 1 | 1538 |
| 2 | 135 |
| 3 | 16 |
| 4 | 7 |
| 5 | 2 |

**Extended Data Fig. 1 | Filtering flowchart for step 1 MR. (a)** Flowchart for filtering proteins in step 1 MR. **(b)** A histogram showing the number of pQTL variants that were associated with more than one protein in *cis* and thus removed before Step 2 MR. Further details can be found in Methods.

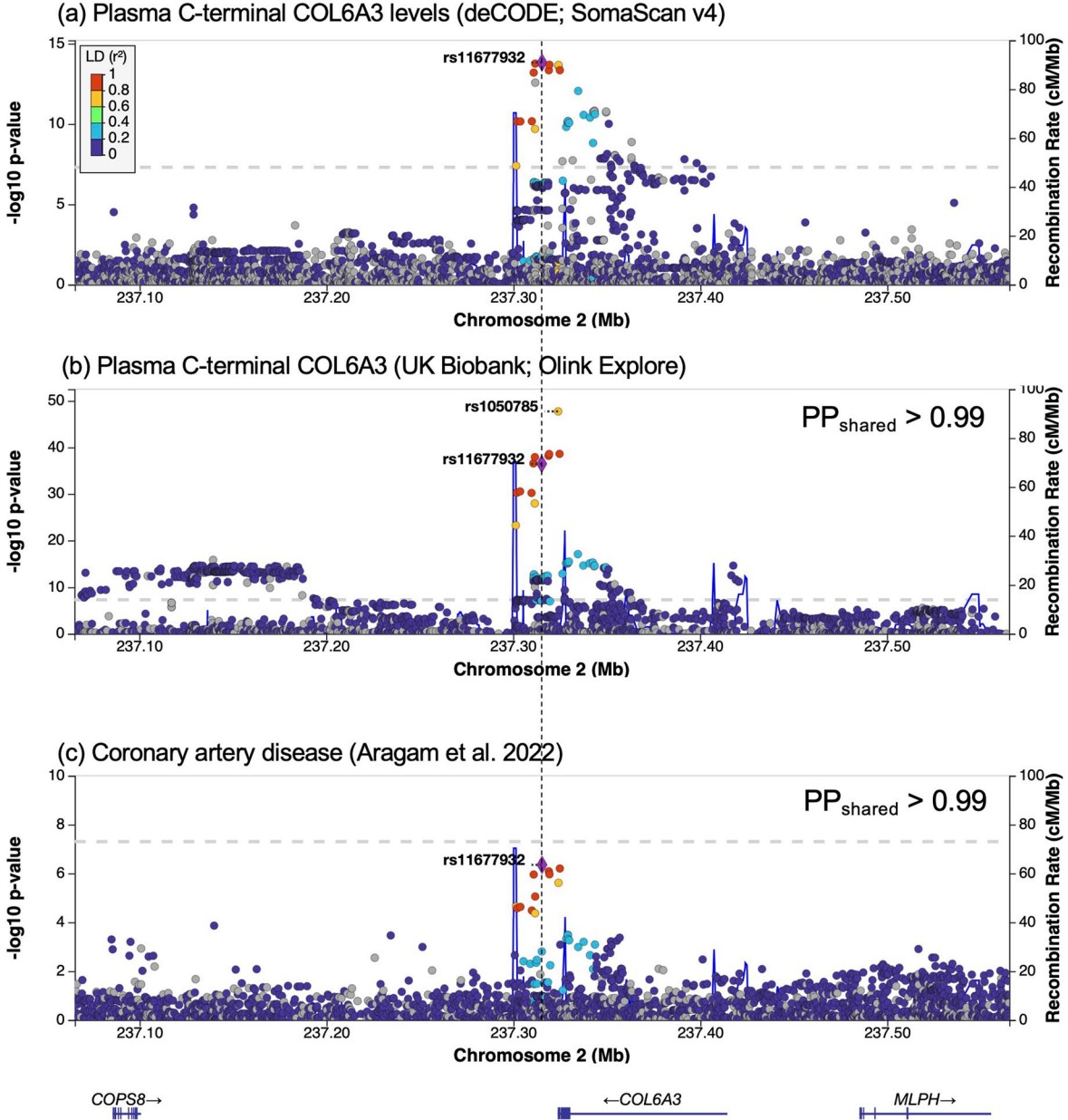

**Extended Data Fig. 2 | LocusZoom plot for cis-pQTL of COL6A3 and coronary artery disease.** (**a**) LocusZoom plot of the pQTL for C-terminal COL6A3 in the 500 kb-region surrounding the lead *cis*-pQTL from deCODE (rs11677932), which used SomaScan v4 assay. Y-axis on the left represents -log10(*P*-value) from the two-tailed *Z* test. (**b**) pQTL for C-terminal COL6A3 in the same region from the UK Biobank, which used Olink Explore 3072 assay[76]. rs1050785 was the lead *cis*-pQTL in this pQTL, which was in LD ($R^2$ = 0.73) with rs11677932. (**c**) Coronary artery disease GWAS from Aragam et al. in the same region[33].

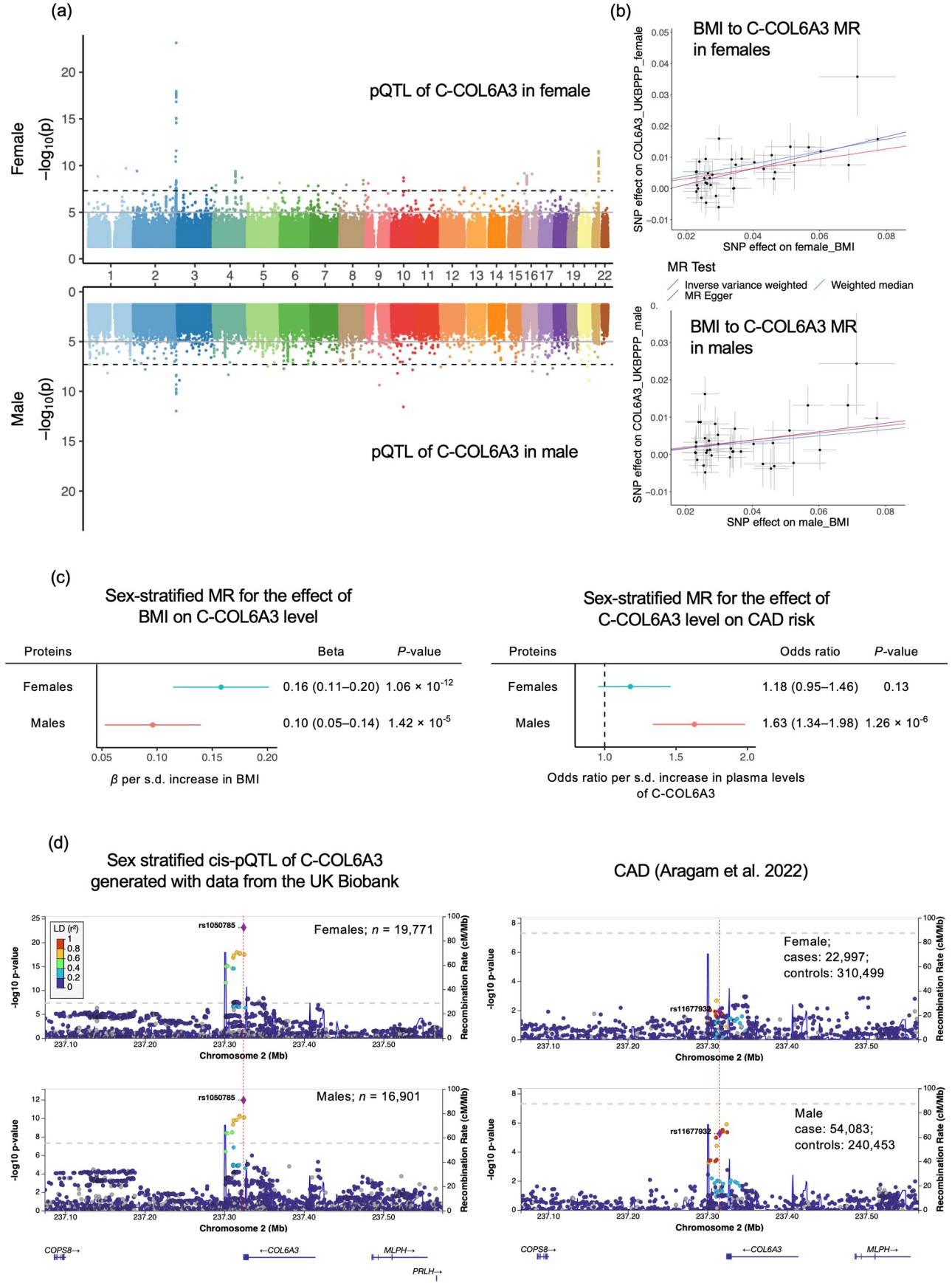

**Extended Data Fig. 3 | See next page for caption.**

**Extended Data Fig. 3 | Sex-stratified analyses of C-terminal COL6A3 (C-COL6A3).** (**a**) Sex-stratified GWAS of plasma levels of C-terminal COL6A3 (sex-stratified pQTL). Only variants with $P < 0.05$ (two-sided Z-test) are presented. The horizontal dashed and gray lines represent $-\log_{10}(P)$ corresponding to $P < 5 \times 10^{-8}$ (genome-wide significance) and $P < 1 \times 10^{-5}$ (suggestive significance) respectively. (**b**) Scatter plots showing the sex-stratified two-sample MR results for the effect of BMI on plasma levels of COL6A3 using the inverse-variance weighted method (primary analysis; red regression line), weighted median (blue regression line), or MR-Egger slope methods (purple regression line). The $P$-values were obtained using the random-effects inverse variance weighted method (two-tailed test). The error bars represent the 95% CI for effect estimates. (**c**) Sex-stratified Step 1 MR results for the effect of BMI on C-terminal COL6A3 levels (left panel) and Step 2 MR results for the effect of C-terminal COL6A3 on CAD risk (right panel). The error bars represent the 95% CI for effect estimates. (**d**) Sex-stratified LocusZoom plots for *cis*-pQTL of C-terminal COL6A3 from the UK Biobank (left panel) and coronary artery disease GWAS from Aragam et al.[33] (right panel). Y-axis on the left represents $-\log10(P$-value) from the two-tailed $Z$ test. C-COL6A3 = C-terminal COL6A3.

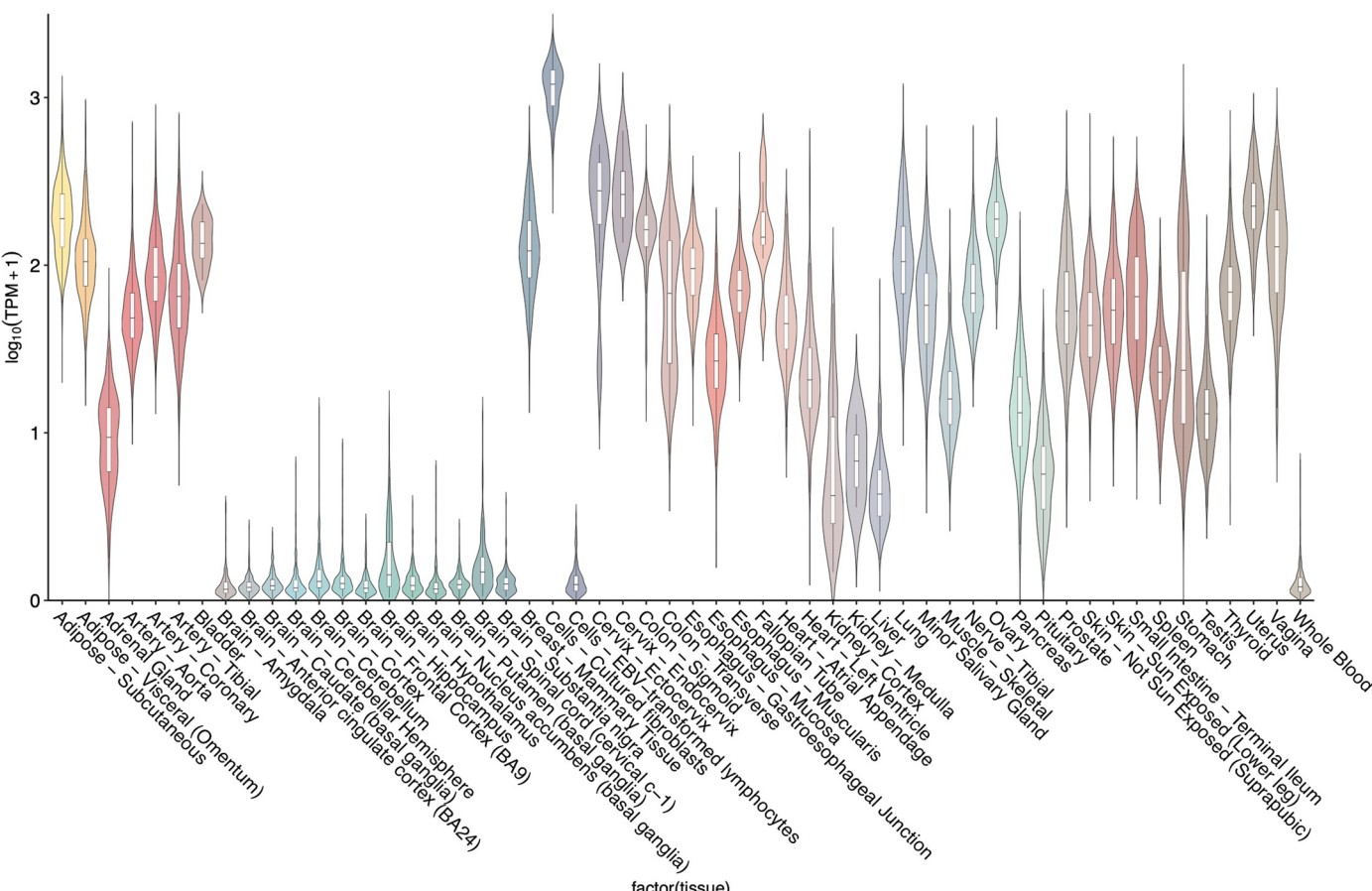

**Extended Data Fig. 4 | COL6A3 expression profile in human tissues in GTEx v.8.** *COL6A3* expression levels in 49 human tissues from GTEx v.8[52] were represented on a log transcript per 10 thousand plus one (TPM + 1) scale. Violin plots illustrate the distribution of expression levels, with boxes showing the interquartile range and horizontal lines indicating the median expression level. Whiskers represent the maximum and minimum values within 1.5 times the IQR from the first and third quartiles.

# Reporting Summary

## Statistics

For all statistical analyses, confirm that the following items are present in the figure legend, table legend, main text, or Methods section.

| n/a | Confirmed | |
|---|---|---|
| ☐ | ☒ | The exact sample size (*n*) for each experimental group/condition, given as a discrete number and unit of measurement |
| ☐ | ☒ | A statement on whether measurements were taken from distinct samples or whether the same sample was measured repeatedly |
| ☐ | ☒ | The statistical test(s) used AND whether they are one- or two-sided<br>*Only common tests should be described solely by name; describe more complex techniques in the Methods section.* |
| ☐ | ☒ | A description of all covariates tested |
| ☐ | ☒ | A description of any assumptions or corrections, such as tests of normality and adjustment for multiple comparisons |
| ☐ | ☒ | A full description of the statistical parameters including central tendency (e.g. means) or other basic estimates (e.g. regression coefficient) AND variation (e.g. standard deviation) or associated estimates of uncertainty (e.g. confidence intervals) |
| ☐ | ☒ | For null hypothesis testing, the test statistic (e.g. *F*, *t*, *r*) with confidence intervals, effect sizes, degrees of freedom and *P* value noted<br>*Give P values as exact values whenever suitable.* |
| ☒ | ☐ | For Bayesian analysis, information on the choice of priors and Markov chain Monte Carlo settings |
| ☒ | ☐ | For hierarchical and complex designs, identification of the appropriate level for tests and full reporting of outcomes |
| ☐ | ☒ | Estimates of effect sizes (e.g. Cohen's *d*, Pearson's *r*), indicating how they were calculated |

*Our web collection on statistics for biologists contains articles on many of the points above.*

## Software and code

Policy information about availability of computer code

| | |
|---|---|
| Data collection | No software has been used for data acquisition. |
| Data analysis | We used R v4.1.2 (https://www.r-project.org/), TwoSampleMR v.0.5.6 (https://mrcieu.github.io/TwoSampleMR/), survival v3.5.8 (https://github.com/therneau/survival), trackplot R v1.0 (https://github.com/PoisonAlien/trackplot), RNOmni v1.01 (https://github.com/zrmacc/RNOmni), snappy v1.0 (https://gitlab.com/richardslab/vince.forgetta/snappy), coloc v5.1.0 (https://chr1swallace.github.io/coloc/), PLINK v1.9 (http://pngu.mgh.harvard.edu/purcell/plink/), susieR v 0.11.92 (https://github.com/stephenslab/susieR/), REGENIE v3.2.9 (https://github.com/rgcgithub/regenie/), and Seurat v4.0.6 (https://satijalab.org/seurat/). Custom codes are available on GitHub (https://github.com/satoshiyoshiji/cm_proteogenomics/). |

For manuscripts utilizing custom algorithms or software that are central to the research but not yet described in published literature, software must be made available to editors and reviewers. We strongly encourage code deposition in a community repository (e.g. GitHub). See the Nature Portfolio guidelines for submitting code & software for further information.

# Data

Policy information about availability of data

All manuscripts must include a data availability statement. This statement should provide the following information, where applicable:
- Accession codes, unique identifiers, or web links for publicly available datasets
- A description of any restrictions on data availability
- For clinical datasets or third party data, please ensure that the statement adheres to our policy

The GWAS of plasma COL6A3 level in males and females are available at GWAS Catalog (GCP ID: GCP001023). We used publicly available GWAS summary statistics from the following source:

BMI GWAS from GIANT and UK Biobank (https://portals.broadinstitute.org/collaboration/giant/; doi: 10.1093/hmg/ddy271).

Plasma proteome GWAS from deCODE (https://www.deCODE.com/summarydata/; doi: 10.1038/s41588-021-00978-w), UK Biobank (https://www.ukbiobank.ac.uk/; doi: 10.1101/2022.06.17.496443), Fenland (https://omicscience.org/apps/pgwas/; doi: 10.1126/science.abj1541), and ARIC (http://nilanjanchatterjeelab.org/pwas/; doi: 10.1038/s41588-022-01051-w).

We also used the CAD GWAS from CARDIoGRAMplusC4D (http://www.cardiogramplusc4d.org/ doi: 10.1038/s41588-022-01233-6), stroke GWAS from GIGASTROKE (GCST90104534 and GCST90104535, at https://www.ebi.ac.uk/gwas/studies/), and type 2 diabetes GWAS from Mahajan et al. (https://doi.org/10.1038/s41588-022-01058-3).

For gene expression data, we used data from Nathan et al. (SCP498 at Single Cell Portal https://singlecell.broadinstitute.org/) and Wirka et al (GSE131780 at Gene Expression Omnibus database https://www.ncbi.nlm.nih.gov/geo/).

Epigenomic data are available at RegulomeDB (https://regulomedb.org/) and ENCODE (https://www.encodeproject.org/).

Variant-to-gene (V2G) scores are available at the Open Target Genetics (https://genetics-docs.opentargets.org/data-access/data-download).

Individual-level data of the UK Biobank, EPIC-Norfolk, and CellGenBankCohort are available through respective party upon agreement.

# Research involving human participants, their data, or biological material

Policy information about studies with human participants or human data. See also policy information about sex, gender (identity/presentation), and sexual orientation and race, ethnicity and racism.

| | |
|---|---|
| Reporting on sex and gender | We performed sex-stratified analyses based on genetically determined sex (UK Biobank data-field 22001). |
| Reporting on race, ethnicity, or other socially relevant groupings | We focused on analyzing data solely from European-ancestry individuals to prevent confounding by population stratification. While the ARIC cohort reported cis-pQTL for individuals of African ancestry[16], the sample size (n = 1,871) is still limited when compared to data for those of European ancestry (deCODE study; n = 35,559). The same applies to CAD GWAS, with 181,522 CAD cases in European ancestry individuals compared to only 17,247 cases in African ancestry individuals. This limited sample size in African ancestry individuals reduces the statistical power of MR analysis. Therefore, further efforts are needed to increase the sample size of non-European-ancestry data. |
| Population characteristics | •BMI GWAS: We used the BMI GWAS meta-analysis with the largest sample size, comprising 693,529 European ancestry individuals from the GIANT consortium and UK Biobank.<br>•Body fat percentage GWAS: We used body fat percentage GWAS in 454,633 individuals of European ancestry from the UK Biobank, obtained from the IEU OpenGWAS project (https://gwas.mrcieu.ac.uk/). The accession ID was ukb-b-8909.<br>•Proteomic GWAS: For the primary analysis, we used the largest proteomic GWAS available, which measured 4,907 proteins in 35,559 individuals of European ancestry from the deCODE study (Ferkingstad et al.). We also used the GWAS of plasma COL6A3 level from UK Biobank in 35,571 individuals of European ancestry, that from the Fenland in 12,084 individuals of European ancestry, and that from ARIC in individuals of European ancestry in 7,213 individuals of Icelandic ancestry.<br><br>•Coronary artery disease GWAS: Meta-analysis of GWASs in individuals of European ancestry (181,522 cases and 984,168 controls) from UK Biobank + CARDIoGRAMplusC4D (Aragam et al. Nat Genet 2022).<br>•Ischemic stroke GWAS: Meta-analysis of GWASs in individuals of European ancestry (73,652 cases and 1,234,808 controls) from GIGASTROKE (Mishra et al. Nature 2022).<br>•Cardioembolic stroke GWAS: Meta-analysis of GWASs in individuals of European ancestry (122,616 cases and 2,475,240 controls) from GIGASTROKE (Mishra et al. Nature 2022).<br>•Type 2 diabetes GWAS: Meta-analysis of GWASs in individuals of European ancestry (80,154 cases and 853,816 controls) from DIAMANTE (Mahajan et al. Nat Genet 2022).<br>•EPIC-Norfolk: The EPIC-Norfolk study is a component of the pan-European EPIC Study, a population-based cohort in Norfolk, a county in Eastern England. We performed observational association analysis with a randomly selected sub-cohort of the EPIC-Norfolk study (n = 872), which included 207 prevalent or incident cases of CAD. Mean age = 59.05 (SD = 9.54). N females = 502 (57.6%).<br>•UK Biobank: The UK Biobank is a large-scale population-based prospective study. We performed Cox regression analysis using individual-level data from 38,361 people, which included baseline clinical variables, C-terminal COL6A3 levels, and cumulative CAD events over up to 10 years. |
| Recruitment | · EPIC-Norfolk: The EPIC-Norfolk study recruited middle-aged individuals from the general population of Norfolk, a county in Eastern England, who attended the baseline assessment between 1993–1998.<br>· Other studies: All other studies described their recruitment methods in their respective studies, which are cited in the References section |
| Ethics oversight | All contributing cohorts obtained ethical approval from their intuitional ethics review boards. The contributing cohorts include UK Biobank, GIANT consortium, deCODE study, Fenland study, AGES Reykjavik study, INTERVAL study, CARDIoGRAMplusC4D, GIGASTROKE, and MAGIC consortium. For individual-level data, the study was approved by the UK Biobank (application number: 27449) and the Norfolk Research Ethics Committee (no. 05/ Q0101/191), and all participants |

gave their informed written consent. For human adipose-derived mesenchymal stem cells used in LipocyteProfiler, each participant gave written informed consent before inclusion and the study protocol was approved by the ethics committee of the Technical University of Munich (Study No 5716/13) and the Broad Institute of MIT and Harvard (IRB number: ORSP-1613).

Note that full information on the approval of the study protocol must also be provided in the manuscript.

# Field-specific reporting

Please select the one below that is the best fit for your research. If you are not sure, read the appropriate sections before making your selection.

☒ Life sciences ☐ Behavioural & social sciences ☐ Ecological, evolutionary & environmental sciences

For a reference copy of the document with all sections, see nature.com/documents/nr-reporting-summary-flat.pdf

# Life sciences study design

All studies must disclose on these points even when the disclosure is negative.

| | |
|---|---|
| Sample size | The sample size of each dataset is provided in Supplementary Table 1. |
| Data exclusions | Plasma proteins whose cis-pQTLs were not available in the largest proteomic GWAS study were excluded from Mendelian randomization (MR) analyses evaluating the effect of obesity on cardiometabolic outcomes. |
| Replication | Apart from the primary discovery MR analysis with data from deCODE, we performed another set of MR using independent cohorts: the UK Biobank, the Fenland study, and the ARIC study. |
| Randomization | Not applicable since this is not an interventional study. |
| Blinding | Not applicable since this is not an interventional study. |

# Reporting for specific materials, systems and methods

We require information from authors about some types of materials, experimental systems and methods used in many studies. Here, indicate whether each material, system or method listed is relevant to your study. If you are not sure if a list item applies to your research, read the appropriate section before selecting a response.

## Materials & experimental systems

| n/a | Involved in the study |
|---|---|
| ☒ | Antibodies |
| ☐ | ☒ Eukaryotic cell lines |
| ☒ | Palaeontology and archaeology |
| ☒ | Animals and other organisms |
| ☒ | Clinical data |
| ☒ | Dual use research of concern |
| ☒ | Plants |

## Methods

| n/a | Involved in the study |
|---|---|
| ☒ | ChIP-seq |
| ☒ | Flow cytometry |
| ☒ | MRI-based neuroimaging |

## Eukaryotic cell lines

Policy information about cell lines and Sex and Gender in Research

| | |
|---|---|
| Cell line source(s) | Human adipose-derived mesenchymal stem cells were obtained from the Munich Obesity BioBank (MOBB). |
| Authentication | The cells were not authenticated. |
| Mycoplasma contamination | The cells were negative for mycoplasma. |
| Commonly misidentified lines (See ICLAC register) | Not used in the study. |

