## [Peer Review File · Nature Genetics]

Integrative proteogenomic analysis identifies COL6A3-derived endotrophin as a mediator of the effect of obesity on coronary artery disease

Corresponding Author: Professor J. Brent Richards

Version 0:

Decision Letter:

19th August 2023

Dear Brent,

Your Article "COL6A3-derived endotrophin mediates the effect of obesity on coronary artery disease: an integrative proteogenomics analysis" has been seen by three referees. You will see from their comments below that, while they find your work of potential interest, they have raised substantial concerns that must be addressed. In light of these comments, we cannot accept the manuscript for publication at this time, but we would be interested in considering a suitably revised version that addresses the referees' concerns.

We hope you will find the referees' comments useful as you decide how to proceed. If you wish to submit a substantially revised manuscript, please bear in mind that we will be reluctant to approach the referees again in the absence of major revisions.

To guide the scope of the revisions, the editors discuss the referee reports in detail within the team, including with the chief editor, with a view to identifying key priorities that should be addressed in revision, and sometimes overruling referee requests that are deemed beyond the scope of the current study. In this case, we ask that you address all technical queries related to the genetic association and Mendelian randomization analyses, extending these analyses as requested by the referees and revising the presentation accordingly, and address concerns regarding the specificity of BMI's effects on COL6A3 vs. other plasma proteins and the strength of the evidence implicating COL6A3 as a causal mediator of the effects of BMI on CAD risk, revising the interpretations where appropriate. We hope you will find this prioritized set of referee points to be useful when revising your study. Please do not hesitate to get in touch if you would like to discuss these issues further.

If you choose to revise your manuscript taking into account all reviewer and editor comments, please highlight all changes in the manuscript text file. At this stage, we will need you to upload a copy of the manuscript in MS Word .docx or similar editable format.

*2) If you have not done so already please begin to revise your manuscript so that it conforms to our Article format instructions, available here. Refer also to any guidelines provided in this letter.

*3) Include a revised version of any required Reporting Summary: <https://www.nature.com/documents/nr-reporting-summary.pdf>

Link Redacted

If you wish to submit a suitably revised manuscript, we would hope to receive it within 3-6 months. If you cannot send it within this time, please let us know. We will be happy to consider your revision so long as nothing similar has been accepted for publication at Nature Genetics or published elsewhere. Should your manuscript be substantially delayed without notifying us in advance and your article is eventually published, the received date would be that of the revised, not the original, version.

Thank you for the opportunity to review your work.

Sincerely,
Kyle

Kyle Vogan, PhD
Senior Editor
Nature Genetics
<https://orcid.org/0000-0001-9565-9665>

Referee expertise:

Referee #1: Genetics, cardiometabolic diseases, Mendelian randomization

Referee #2: Genetics, complex traits, molecular QTLs

Referee #3: Genetics, cardiometabolic diseases, molecular QTLs

Reviewers' Comments:

Reviewer #1:
Remarks to the Author:

Yoshiji et al. perform an integrative genomics study on circulating proteins that mediate the effects of obesity on cardio metabolic disease. They identified seven circulating proteins, including COL6A3 which was increased by body mass index. COL6A3 importantly increased the risk of CAD. Performed a series of follow up analyses to assess the clinical actionability of this finding.

Overall, I thought this was a well-done study to highlight endotrophin as a potential therapeutic target. The authors should be commended on this carefully thought out integrative analysis. My comments are below:

1. Two sample MR should be performed where there is no sample overlap between the two GWAS; otherwise, can cause bias in the causal estimates.

For the BMI MR analysis, please confirm that the GWAS for BMI from the GIANT/UK Biobank does not have sample overlap with the GWAS for circulating protein levels from the deCODE, e.g. does GIANT GWAS contain deCODE / UK Biobank samples?

Also, same question for the GWAS for cardio metabolic disease outcomes for coronary artery disease, ischemic stroke, cardioembolic stroke, type 2 diabetes.

2. "Directional horizontal pleiotropy was tested using the MR-Egger intercept test." The MR-Egger regression is used to estimate the average pleiotropic effect across a set of instruments via an intercept parameter. Other tests of horizontal pleiotropy should be considered including Cochran's Q (modified) and MR-PRESSO global test

3. Include statement about MR analysis following MR-STROBE guidelines?

4. IVW, weighted median and MR-Egger were performed. Typically, multiple MR tests are performed for sensitivity analysis. Although the ones tested are amongst the first developed and more popular MR tests, other MR tests can be considered that test a fuller range of % pleiotropy SNVs / IVs. For example, ones that test for outlier detection / removal in MR (only a few SNVs/IVs have strong pleiotropy).

5. pQTLs (and eQTLs) are notoriously difficult in terms of identifying the causal gene for disease because they are driven by non-coding variants.

Although colocalization was one of the steps performed, to ensure the same causal genetic signal between pQTL and GWAS with cardio metabolic diseases, other more sophisticated fine-mapping approaches exist. How can we be assured that the circulating protein indicated in the pQTL is the underlying causal gene contributing to disease? Can the authors run other fine-mapping methods to ensure consistency with the coloc results?

6. The focus of the study has been on using pQTL for MR for BMI effect on plasma protein levels. Identified and honed in on COL6A3. As a sensitivity analysis, has the authors considered performing MR analysis of BMI on COL6A3 gene expression levels, to show that BMI is influencing both mRNA and protein levels of COL6A3.

7. I understand the focus on the paper is on circulating proteins, but drug discovery and validation is typically best supported not by just one genetic association signal but a series of genetic association signals via the allelic series model. Is it possible to test if there are low frequency and/or rare coding variants in COL6A3 are associated with CAD?

8. The domain-aware MR is quite intriguing and interesting way to determine that the C-terminal end of COL6A3 explains the effect of COL6A3 levels on CAD.

9. The GTEx results don't seem that convincing. COL6A3 is expressed in adipose, and coronary artery tissue which are disease relevant. However, it's also expressed in several other tissues that aren't notably relevant to cardio metabolic diseases such as colon tissue. In fact, the gene is expressed in all tissues except brain tissues. Therefore, it's difficult to say that the gene is tissue-specific in the disease relevant tissues. Can consider moving this figure to the supplement.

Minor comments:

10. Figure 2B: Would be nice to highlight positive controls for MR analysis of BMI on plasma protein - Have any of these been confirmed in previous studies?

11.- The PCSK9 positive control example is a nice validation of the approach - perhaps label this on Figure 2B

Reviewer #2:

Remarks to the Author:

Summary: In the current manuscript "COL6A3-derived endotrophin mediates the effect of obesity on coronary artery disease: an integrative proteogenomics analysis", Yoshiji et al. attempt to integrate published results from genomics and proteomics studies in order to identify plasma proteins affecting cardio- metabolic diseases through their effect on body mass index. Following mendelian randomization analysis, the authors assess and claim a causal role of obesity on the level of a certain number of proteins, including COL6A3 plasma levels. They further tested through MR the effect of the levels of these proteins on cardiometabolic outcomes including CAD and claim a causal effect of COL6A3 levels on CAD. Currently, the results are not fully convincing. Whereas the topic could have been of interest, the way the analysis is performed and the results are presented do not provide clarity and conviction.

1) It appears that the attempt of finding specific proteins that have their level influenced by BMI did not bring a successful narrow selection. In the abstract and through parts of the text, the authors omit to mention and interpret that a large majority of tested proteins or 2,714 of 4,907 have their level influenced by BMI in a causal manner and that they consider "BMI driven proteins" (including COL6A3).

2) The whole step 1 of the study is then becoming difficult to use. The authors need to realize that majority and protein have their level affected causally by BMI. Whereas I understand the focus on certain locus where variant have been associated with CAD, they have to realize that a vast majority of proteins have their level influenced by BMI so it is not pointing to a specific set, or not that meaningful.

I think that readers in general would have difficulty to accept that step 1 of the study is giving a meaningful result.

- 3) The fraction of protein presented in the step 1 represents 55.3% of proteins tested with aptamer, even if the authors use a stringent FDR of 0.5%. It would be interesting to see what is the number of proteins at an FDR of 5%.
- 4) Also, since assay on such platform are not always succeeding to measure the attempted levels, in large part due to the low concentration in plasma of these proteins, thus this fraction can even be underestimated.
- 5) It can be noted that based on the aptamer-based study that the authors used, an even greater fraction of proteins associates with BMI directly (87%). It should also be pointed that there is a number of other proteins that have an even more significant direct association with BMI than COL6A3. Whereas many proteins are displayed on figure 2b, COL6A3 is not. This suggests that the rank of COL6A3 is lower.
- 6) The protein with levels associates the most strongly with BMI were LEP, FABP3 and FABP4. The levels of these proteins associating most strongly to BMI when directly associated, where also directly associated strongly to coronary artery disease in the report data from Ferkingstad.
- 7) Similar to figure 2c, they should think of looking at association of BMI variants and LEP, FABP3 and FABP4 levels and would notice that they are not different in interpretation.
- 8) Thus, for majority of tested proteins (at least 55%), BMI has a causal role and talking about BMI driven proteins appears misleading when it is such a massive effect all over the proteome.
- 9) The association of variants at COL6A3 with CAD has been present in summary statistics for 6 years (Van der Harst P 2017) and also more recently by Koyama (2020) at genome-wide significance according to Open Target Genetics ([https://genetics.opentargets.org/Gene/ENSG00000163359/associations?colocTraitFilter=Coronary artery disease&traitFilter=Coronary artery disease](https://genetics.opentargets.org/Gene/ENSG00000163359/associations?colocTraitFilter=Coronary%20artery%20disease&traitFilter=Coronary%20artery%20disease))
- 10) When presenting COL6A3 association to CAD and protein level, more detailed and convincing reasoning and analysis are required. Since the step 1 of the study is not convincing, I would start with trying to perform MR or colocalization of all reported CAD variants with proteomics data.
- 11) In the chapter starting at line 218, the authors should present a locus plot of the cis association of COL6A3 plasma levels.
- 12) They should also present a locus plot of the association of variants to CAD at COL6A3.
- 13) They should present and discuss trans pQTL COL6A3 association and discuss them (from any study in the literature).
- 14) The authors should show an effect/effect plot for all variants correlating with COL6A3 in cis and trans and perform and MR not limiting to cis pQTL variants.
- 15) The authors should show a colocalization plot between COL6A3 levels and CAD using variants at COL6A3 locus.
- 16) The authors need through the text line 117-121 of introduction to clearly indicate that they perform mendelian randomization on existing pQTL data. Then they need to clearly indicate if it is aptamer (and what normalization of the data has been performed; SMP or no SMP) or antibody-based methods.
- 17) The authors need to indicate that they are mainly using existing CAD reported data and they should clearly show what are the best signal at COL6A3 locus with CAD.
- 18) The authors restricted instrumental variables to genetic variants that were cis-pQTLs to only one protein. Can they please in main text underline what fraction is removed? I assume it is very limited.
- 19) In figure 3b and in abstract and line 237, the authors show a predicted effect on CAD of on s.d. in COL6A3 genetically determined level by cis pQTL. I think this can easily be misleading, since they have a genetic instrument, they should show for the best cis pQTL variant what is the effect on CAD and what its colocalization with CAD signal is.
- 20) Concerning the example that the authors use for PCSK9, the authors have to be very careful in at least two ways that would not support their claim. First, whereas PCSK9 levels associate with variants in cis, the strongest signal and few other at the locus are represented by coding variants. Coding variants that are cis pQTL are sometimes influenced by epitope binding effects. Thus, it is not clear that the Arg46Leu variant associating with CAD associates with expression level based on the Somascan experiment. Second, whereas obesity influence PCSK9 plasma level, it is not clear to me that there is a large relative contribution of BMI to the CAD risk mediated by PCSK9.
- 21) When discussing the two Somascan probes from Somascan, the authors should clearly name them in the main text.
- 22) Surprisingly, the authors insist in many words on the value of the mediation analysis, but they restrict the numerical result to the supplementary tables. We note that the COL6A3 mediation would have survived a less than 2000 tests. We also note that the authors have 2,714 "BMI proteins" and 4 diseases that they test.

23) The pQTL and CAD results are published at genome-wide significance at COL6A3 - again I would emphasize that as in the result the author should make it more clear.

24) Also, do the authors believe that the sentinel cis pQTL or any linked marker in the credible set can have a functional role?

25) Have the authors performed any single-track assay to confirm the Somascan or Olink measures?

26) The authors should in depth assess if any RNA expression (eQTL as well as splice QTL) can be linked or can explain the cis pQTL of COL6A3 in plasma.

27) Overall, I believe that the BMI angle is very unclear and not specific at all. I would recommend shortening the story and make it to the point of COL6A3 and CAD.

28) Please remove the first part and all mention to BMI as a prior as well as mediation analysis. Try to present convincing evidence for cis pQTL for COL6A3 in different methods with locus plot for Somascan (2 probes) and Olink as well as CAD or related diseases.

29) Since the pQTL and CAD use published data, it would have been interesting if the authors could determine in vitro how the variants affect the protein and its level (transcription, post transcription)

30) Since COL6A3 mutations are known in OMIM to cause Bethlem myopathy 1|Dystonia 27|Ullrich congenital muscular dystrophy 1, it would be appreciated to determine the mechanism of action that is described there. Does it involve the involvement of cardiac disease in the phenotypic spectrum? What do the known mutations do? How could they be put in the context of pQTL?

Reviewer #3:

Remarks to the Author:

In this paper, the authors searched for proteome mediators of the relationship between BMI and CAD applying Mendelian Randomization approaches. They detected COL6A3 as an interesting candidate and performed follow-up analyses regarding this protein using different sources of evidence such as epidemiologic data and single-cell analysis. The topic and approach are interesting, but I have a few suggestions for improvement of the analyses and interpretations.

Major:

1. My biggest concern is with respect to the overall error control of the analyses. Since the authors rely on a sequential testing approach (two Mendelian randomization tests plus mediation test for each of the proteins), the overall error rate is not obvious. In this regard the used FDR cut-offs are not convincingly justified. This could be improved for example by performing permutation testing of the study design or running a small simulation study given the covariance structure of data.

2. BMI is not a convincing marker of obesity and there are contradicting results regarding its causal relationship to CAD. I would propose also considering WHR for which large GWAS are also available.

3. I could not find the estimated total causal effect of BMI on CAD. Please add. Likewise, the proportions of causal effect mediated by the proteins should be provided as well as an estimate of the direct BMI effect.

4. Since multiple proteins are considered as potential mediators, a multi-trait MR analysis could be appropriate to estimate the total amount of mediated causal BMI effect.

5. It remains unclear how the MR assumptions regarding BMI->Proteome are checked. How did you exclude reverse causality? When testing the causal relationship between BMI and a specific gene, I would strongly suggest removing BMI instruments located at the same chromosome as the protein coding gene.

6. Regarding the causal relationships Protein->CAD, the authors should comment on the differences in statistical power for these analyses, which strongly depend on the strength of the cis-effects found for the proteins. Maybe a table comparing detectable causal effects of the MR-analyses for the different proteins could be helpful.

7. I am a bit disappointed that the authors did not consider sex as a major parameter since both obesity and CAD show strong sex-dimorphisms and there are examples of sex-specific causal effects for both traits. I would suggest adding this analysis because there are sex-stratified GWAS results available for obesity-related traits. This might not be available for CAD, but still, sex-specific causal effects could be approximately estimated by considering the causal relationship Protein (sex-specific) -> CAD (overall). This approach was proposed / applied by others.

8. Page 10: The authors mention a replication MR of the COL6A3 -> CAD relationship in other studies. It remains unclear how this was performed. Did you perform single study MR analyses? Or did you only use study-specific pQTLs as

instruments? Please clarify. Please present the meta-effect across studies.

9. Searching the supplemental tables is cumbersome. Please consider adding a main table with the primary results.

10. The possible mediating role of endotrophin could be discussed in more detail. There are for example findings of this hormone in relation to heart failure. Moreover, since sex hormones were described as causally related to obesity and CAD, it could be interesting to analyse possible relationships of endotrophin with sex hormones.

Minor:

1. Page 3: It is a bit odd to cite COVID papers here because these are not related to the topic of this research. There are numerous other papers considering causal effects on obesity / CAD and possible interrelationships which would be more appropriate to cite here to my opinion.

Version 1:

Decision Letter:

Our ref: NG-A62490R

20th Jul 2024

Dear Dr. Richards,

Thank you for submitting your revised manuscript "COL6A3-derived endotrophin mediates the effect of obesity on coronary artery disease: an integrative proteogenomics analysis" (NG-A62490R). It has now been seen by the original referees and their comments are below. The reviewers find that the paper has improved in revision, and therefore we'll be happy in principle to publish it in Nature Genetics, pending minor revisions to satisfy the referees' final requests and to comply with our editorial and formatting guidelines.

As the current version of your manuscript is in a PDF format, please email us a copy of the file in an editable format (Microsoft Word or LaTeX)-- we can not proceed with PDFs at this stage.

Sincerely,

Michael Fletcher, PhD
Senior Editor, Nature Genetics
ORCID: 0000-0003-1589-7087

Reviewer #1 (Remarks to the Author):

The authors have reasonably addressed all my comments.

The fine-mapping, variant-to-gene mapping and epigenetic data analysis, and the causal effect of MRI-derived body fat compartments are nice additions.

I also appreciate the attempt to test rare coding variants in the C-terminal of COL6A4 with CAD (although negative due to lack of power).

Reviewer #2 (Remarks to the Author):

The revision is very extensive. The relationship between COL6A3 and CAD is well described.

However it is not clear to me that this fully mediates the effect of obesity on CAD. Could the author soften their title and abstract.

COL6A3 is clearly causative for CAD but the point that it mediates the obesity effect is more modest.

Thank you again for the extensive effort.

I suggest you add one supp figure :

Can the authors plot the correlation of protein levels to BMI on one axis versus the correlation of BMI to CAD on the other axis: Is COL6A3 outlying there?

Reviewer #3 (Remarks to the Author):

None.

May 7, 2024

Dear Editor and Reviewers

Re: Revision of **“COL6A3-derived endotrophin mediates the effect of obesity on coronary artery disease: an integrative proteogenomics analysis”**

We appreciate the opportunity to resubmit our revised manuscript (NG-A62490) for consideration in *Nature Genetics*. We have carefully addressed the comments from the reviewers and made comprehensive revisions to enhance our manuscript. Below, we provide our responses in a detailed, point-by-point format. Changes in the revised manuscript are highlighted in red and accompanied by line numbers. Additionally, we have summarized the key modifications in the following pages.

We are also pleased to share that our study was selected for an oral platform presentation at the American Society of Human Genetics (ASHG) Annual Meeting 2023, an honor typically reserved for the top 8% of submissions. This distinction highlights the significance and potential impact of our research. The feedback received during this session was both positive and constructive, and we have incorporated these insights into our revised manuscript. Additionally, our study received the Featured Research Award from the Canadian Cardiovascular Society, further recognizing its contribution to the field.

We eagerly await your feedback on our submission and are ready to address any further questions or comments you may have.

Best regards,

Sincerely,

J. Brent Richards

Professor of Medicine, McGill University, Senior Lecturer, King's College London (Honorary), Pavilion H-413, Jewish General Hospital, 3755 Côte-Ste-Catherine Montréal, Québec, H3T 1E2, Canada.

Tel: +1-514-340-8222 Fax: +1-514-340-7529

Response to Editor Dr. Kyle Vogan

In this case, we ask that you address all technical queries related to the genetic association and Mendelian randomization analyses, extending these analyses as requested by the referees and revising the presentation accordingly, and address concerns regarding the specificity of BMI's effects on COL6A3 vs. other plasma proteins and the strength of the evidence implicating COL6A3 as a causal mediator of the effects of BMI on CAD risk, revising the interpretations where appropriate. We hope you will find this prioritized set of referee points to be useful when revising your study. Please do not hesitate to get in touch if you would like to discuss these issues further.

Thank you very much for handling our manuscript . We have carefully considered the reviewers' feedback and have made substantial revisions to address their concerns, particularly regarding the technical aspects, the specificity of COL6A3's role, and the evidence supporting COL6A3 as a mediator in the relationship between obesity and coronary artery disease (CAD).

Key changes include:

- **Change from the FDR correction to Bonferroni correction**

In the revised analyses for both steps 1 and steps 2 MR, we used a Bonferroni correction instead of a False Discovery Rate (FDR) correction. We opted for this approach to minimize the potential for false positive findings. Notably, COL6A3 survived the Bonferroni correction in both step 1 and step 2, and therefore, this change did not affect the downstream analyses focused on COL6A3 and endotrophin.

- **Additional filtering for step 1 MR**

In step 1 MR, where we assessed the causal effect of BMI on plasma protein levels, we included additional filtering as outlined in **Extended Fig. 1**. This included the MR-PRESSO test of directional concordance with body fat percentage, aiming to strengthen the evidence for the effect of obesity on COL6A3. Notably, MR found that the C-terminal of COL6A3 was strongly increased by both BMI ($P = 3.65 \times 10^{-24}$) and body fat percentage ($P = 8.33 \times 10^{-20}$), and therefore downstream analyses focusing on COL6A3 was not affected.

Extended Fig. 1. Filtering flowchart for step 1 MR.

Extended Fig. 1. Filtering flowchart for step 1 MR.

• Evaluating the specific body fat compartments that increase plasma levels of C-terminal COL6A3

Regarding the specificity of BMI's effect on COL6A3, we conducted additional MR analyses to evaluate which body fat components specifically influence plasma C- and N-terminal COL6A3 levels, using GWAS of MRI-derived fat compartment volumes as the exposure for two-sample MR. Among the three fat compartments analyzed—abdominal subcutaneous adipose tissue (ASAT), visceral adipose tissue (VAT), and gluteofemoral adipose tissue (GFAT)—ASAT were found to significantly increase plasma levels of C-terminal COL6A3 (**Fig. 5**). Notably, this finding aligns with previous studies that:

(i) the collagen matrix surrounding subcutaneous fat is rich in C-terminal COL6A3, which in turn releases endotrophin¹,

(ii) increased expression of COL6A3 in subcutaneous adipose tissues is associated with adipose tissue fibrosis, insulin resistance, and metabolic dysfunction²⁻⁶.

(b) COL6A3 (UniProt ID: P12111)

(c) MR for the effect of C- and N-terminal COL6A3 on the risk of CAD stratified by C- and N-terminal COL6A3

(d) MR for the effect of BMI and body fat percentage on COL6A3 stratified by C- and N-terminal COL6A3

(e) MR for the effect of body fat compartments on COL6A3 stratified by C- and N-terminal COL6A3

- **Regulatory role of the lead *cis*-pQTL of C-terminal COL6A3**

We also incorporated ATAC-seq data, along with H3K4me3 and H3K27ac ChIP-seq data from ENCODE⁷ and RegulomeDB⁸, demonstrating that the lead *cis*-pQTL for C-terminal COL6A3 (rs11677932) is located in an open chromatin region and an active enhancer domain across multiple tissues, including adipose tissues and arteries (**Fig. 6**). RegulomeDB assigns a heuristic ranking score, indicating the variant's potential functionality within regulatory elements⁹. The variant received a RegulomeDB score of 1b, suggesting strong evidence of its regulatory role. This assessment is based on data from eQTL studies, transcription factor (TF) binding, TF motifs, DNase footprinting, and DNase peaks. Additionally, this variant is identified as a *cis*-eQTL in the aorta with a directionally consistent effect, where the rs11677932-A allele is associated with reduced COL6A3 gene expression in the aorta, as reported in GTEx. **Notably, this variant disrupts the conserved nucleotide in the MEF2B transcription factor binding motif, which is consistent with the reduced effect of this variant on the expression of COL6A3.** Collectively, these findings offer a comprehensive narrative: the variant's location in a regulatory region and its impact on TF binding contribute to decreased COL6A3 expression and subsequently lower plasma COL6A3 levels.

Fig. 6. Epigenetic profile of the lead cis-pQTL for C-terminal COL6A3.

- **Cox-regression analysis to evaluate the association between baseline C-terminal COL6A3 and the cumulative incidence of CAD**

As an additional orthogonal evaluation, we performed multivariable Cox proportional-hazards regression analysis to evaluate the association between baseline plasma C-terminal COL6A3 levels and up to 10 years of cumulative incidence of CAD in 35,100 individuals (2,969 cases and 32,131 controls) from UK Biobank. After adjustment for age, sex, recruitment center, and protein sample processing time, we found that an s.d. increase in plasma C-terminal COL6A3 level was associated with an increased cumulative incidence of CAD (HR = 1.40, 95% CI: 1.35–1.45, $P < 2.2 \times 10^{-16}$).

Moreover, Kaplan-Meier estimates for the cumulative incidence of CAD by COL6A3 level quantiles (25%, 50%, 75%, and 100%) (**Fig. 4**) showed statistically significant differences, with the top 100% group having the highest incidence and the 25% the lowest (log-rank test $P < 2.2 \times 10^{-16}$).

Fig. 4. Kaplan Meier estimates for cumulative incidents of coronary artery disease by baseline COL6A3 level quantiles in the UK Biobank

- **Sex-stratified analysis of C-terminal COL6A3 level**

Using data from the UK Biobank, we generated sex-stratified pQTLs for C-terminal COL6A3 for both males and females. We then conducted sex-stratified analyses in two steps: Step 1 MR assessed the effect of BMI on C-terminal COL6A3 levels, and Step 2 MR evaluated the impact of C-terminal COL6A3 on CAD risk (**Extended Fig. 3**). For Step 1 MR, we replicated causal effect of BMI on C-terminal COL6A3 level in both sexes. For Step 2 MR, we replicated the causal effect of C-terminal COL6A3 on CAD risk in both sexes.

level in both sexes. For Step 2 MR, we replicated the causal effect of C-terminal COL6A3 level on CAD risk in males, and found a suggestive, directionally consistent effect in females. Additionally, we conducted sex-stratified Cox regression analysis for the 10-year cumulative incidence of CAD in the UK Biobank. We found that an increase in plasma C-terminal COL6A3 level was associated with an increased cumulative incidence of CAD in both females (HR = 1.15, 95% CI: 1.07–1.23, $P = 6.5 \times 10^{-5}$) and males (HR = 1.24, 95% CI: 1.18–1.30, $P = 2.2 \times 10^{-16}$). Collectively, these findings indicate the shared effect of C-terminal COL6A3 on CAD risk in both sexes.

Extended Fig. 3. Sex-stratified analyses of C-terminal COL6A3 (C-COL6A3).

These changes, alongside other modifications made in response to reviewers' constructive comments, have greatly improved the manuscript. Our study uses multiple lines of evidence, study designs, and data types to highlight COL6A3-derived endotrophin as a potential therapeutic target.

Response to Reviewer 1

Comments from Reviewer 1

Overall, I thought this was a well-done study to highlight endotrophin as a potential therapeutic target. The authors should be commended on this carefully thought out integrative analysis. My comments are below

We thank the reviewer for the valuable comments. We have revised our manuscript according to your insightful suggestions. Please find our point-by-point responses below.

Comments from Reviewer 1

1. Two sample MR should be performed where there is no sample overlap between the two GWAS; otherwise, can cause bias in the causal estimates.
--

For the BMI MR analysis, please confirm that the GWAS for BMI from the GIANT/UK Biobank does not have sample overlap with the GWAS for circulating protein levels from the deCODE, e.g. does GIANT GWAS contain deCODE / UK Biobank samples?
--

Also, same question for the GWAS for cardio metabolic disease outcomes for coronary artery disease, ischemic stroke, cardioemtabolic stroke, type 2 diabetes.

We appreciate your valuable comments. In the revised article, we have discussed the potential bias arising from sample overlap between the exposure and outcome GWAS in our two-sample MR, based on the framework established by Burgess et al¹⁰. As Burgess et al. elucidated, sample overlap in two-sample MR can bias the causal estimate either towards the null or in the direction of the observed association between the risk factor and the outcome. This bias is a linear function of the overlap proportion between the samples and is also influenced by the strength of the genetic variant, as indicated by the F-statistic. When the F-statistic is low, the potential for bias increases. For instance, with an F-statistic of 10 and a 50% sample overlap, the relative bias—defined as the degree to which MR's causal estimate is biased due to sample overlap compared to the observational estimate—is estimated to be $0.5 \times 1/10$, which equates to 5%.

We consulted Dr. Burgess and calculated the maximum possible relative bias as follows: The sample overlap in our study is minimal: 3.9% in Step 1 MR and up to 2.85% in Step 2 MR. Furthermore, there was no indication of weak instrumental variables: the F-statistic for BMI in Step 1 MR was high at 94.16, and the median F-statistics for proteins in Step 2 MR was 272.42, with the minimum F-statistic being 27.48, well above the threshold of 10. Consequently, the estimated relative bias was a mere 0.0418% for Step 1 MR and 0.1% or less for Step 2 MR. This suggests that the bias due to sample overlap is negligible in our study, owing to the small sample overlap and the robust instrumental variables, as reflected by high F-statistics. Dr. Burgess agreed with this conclusion

(personal communication). We have included this in the Manuscript and Supplementary Note, as detailed below.

Manuscript page 12

Potential bias due to sample overlap in two-sample MR

Additionally, we evaluated the potential bias due to sample overlap in two-sample MR in both Step 1 and Step 2. Sample overlap between the exposure and outcome GWAS can bias the causal estimate either towards the null or in the direction of the observational association between the risk factor and outcome. Relative bias quantifies the extent to which MR's causal estimate is biased due to sample overlap relative to the observational estimate¹⁰. The maximum relative bias was calculated to be 0.041% for Step 1 MR (see **Methods** and **Supplementary Table 10**). In Step 2 MR, we estimated the causal effects of protein levels on the risk of CAD, ischemic stroke, cardioembolic stroke, and type 2 diabetes. The maximum relative bias was calculated to be 0.096% for CAD, 0.100% for ischemic stroke, 0.104% for cardioembolic stroke, and 0.139% for type 2 diabetes (**Supplementary Table 10**).

Supplementary Note

Potential bias due to sample overlap in two-sample MR

Additionally, we evaluated the potential bias due to sample overlap in two-sample MR in both Step 1 and Step 2. Sample overlap between the exposure and outcome GWAS can bias the causal estimate either towards the null or in the direction of the observational association between the risk factor and outcome. Relative bias, which quantifies the extent to which MR's causal estimate is biased due to sample overlap relative to the observational estimate¹⁰, is calculated as:

$$Relative\ bias = \phi \times \frac{1}{F}$$

where ϕ is the proportion of the sample overlap (ranging from 0 to 1) and F is the F-statistic of the exposure.

In Step 1, we estimated the causal effect of BMI on protein levels. The F-statistic for BMI was 94.16, and the sample overlap was at most 3.93%, given that the deCODE

cohort contributed data for 26,799 individuals to the GIANT consortium, and the GWAS of BMI included 693,529 individuals. Thus, the maximum relative bias for Step 1 MR was calculated to be 0.0418% (see **Methods** and **Supplementary Table XX**).

In Step 2 MR, we estimated the causal effects of protein levels from deCODE on coronary artery disease, ischemic stroke, cardioembolic stroke, and type 2 diabetes. The median F-statistic for protein levels was 272.42 (25th percentile: 103.24, 75th percentile: 660.48, minimum: 27.48) (**Supplementary Table 2**). The maximum potential sample overlap between the outcome GWAS and pQTL from deCODE was 35,559, reflecting the number of individuals with available proteomics data¹¹. Consequently, the maximum percentage of potential sample overlap was calculated to be 2.64% for CAD (35,559 out of 1,347,212 individuals in the CAD GWAS), 2.74% for ischemic stroke (35,559 out of 1,296,908), 2.85% for cardioembolic stroke (35,559 out of 1,245,612), and 2.65% for type 2 diabetes (35,559 out of 1,339,889). Using the minimal F-statistics and the maximal possible sample overlap, the estimated maximum relative bias was estimated to be 0.096% for CAD, 0.100% for ischemic stroke, 0.103% for cardioembolic stroke, and 0.096% for type 2 diabetes (**Supplementary Table 10**).

Overall, due to the robust genetic instruments and minimal sample overlap, the estimated relative bias was found to be low.

Comments from Reviewer 1

2. "Directional horizontal pleiotropy was tested using the MR-Egger intercept test." The MR-Egger regression is used to estimate the average pleiotropic effect across a set of instruments via an intercept parameter. Other tests of horizontal pleiotropy should be considered including Cochran's Q (modified) and MR-PRESSO global test.

4. IVW, weighted median and MR-Egger were performed. Typically, multiple MR tests are performed for sensitivity analysis. Although the ones tested are amongst the first developed and more popular MR tests, other MR tests can be considered that test a fuller range of % pleiotropy SNVs / IVs. For example, ones that test for outlier detection / removal in MR (only a few SNVs/IVs have strong pleiotropy).

We appreciate the valuable feedback. We have incorporated the Cochran's I^2 heterogeneity test ($I^2 < 50\%$) and the directional concordance test utilizing MR-PRESSO outlier-corrected estimates, alongside with other outlier-robust methods (weighted median and MR-Egger slope methods) in Step 1 MR.

The caveat of the MR-PRESSO global test is that its statistical power is sensitive to the number of instrumental variables (IVs), as discussed in MR-PRESSO's GitHub page (Are MR-PRESSO sensitive to numbers of instrumental SNPs? #21: <https://github.com/rondolab/MR-PRESSO/issues/21>). This may indicate that MR-PRESSO global test can be overly sensitive when there is a large number of IVs for an exposure such as BMI, which has 304 IVs.

Indeed, we observed that MR-PRESSO global test will be highly significant for well-known BMI-protein associations, such as BMI-LEP ($P = 1.33 \times 10^{-4}$) and BMI-CRP ($P = 3.33 \times 10^{-4}$), although the IVW-derived estimates and outlier-corrected estimates by MR-PRESSO were directionally concordant and both significant ($P < 1 \times 10^{-5}$)

This is partially because MR-PRESSO global test compares the observed sum of residual squares with the expected residual sum of squares (RSS) from the effect size. Thus, the divergence between observed and expected RSS cumulatively increases as the number of IVs increases, even in cases of balanced horizontal pleiotropy.

Note that β_{ZX} and β_{ZY} refer to the SNP-exposure and SNP-outcome association estimate. β_i refers to the wald-type estimator for SNP i , given by $\beta_i = \beta_{ZY}(i)/\beta_{ZX}(i)$.

Therefore, we decided not to filter proteins based on the MR-PRESSO global test. However, we did incorporate MR-PRESSO outlier-corrected estimates into the new

directional concordance test in Step 1 MR in the revised manuscript. Specifically, we required proteins to show directionally concordant estimates across all methods (IVW, weighted median, MR-Egger slope, and MR-PRESSO outlier-corrected estimates).

Additionally, in response to a comment from another reviewer, we transitioned from FDR correction to Bonferroni correction. We acknowledge that this approach is overly conservative, especially since some proteins are highly correlated with each other, but we adopted it to safeguard against false positives.

The revised filtering process is illustrated in the flowchart below and is also included as a new Extended Figure (**Extended Fig. 1**). We did not perform MR-PRESSO in Step 2, where we conducted *cis*-MR, because many proteins have only one or a few instrumental variables, for which MR-PRESSO cannot be run. The manuscript has been updated accordingly.

Manuscript page 8

1) Step 1 MR: Estimation of the causal effect of BMI on plasma protein levels

...

We performed two-sample MR, using the inverse variance weighted (IVW) method as the primary analysis and then filtered these results dependent upon sensitivity analyses. To control for multiple testing, we applied Bonferroni correction ($P < 1 \times 10^{-5}$; $0.05/4,907$) as a stringent significance threshold for the IVW-derived estimates. We acknowledge that this approach may be overly conservative due to the correlations among protein levels, but we did so to minimize the risk of false positives. Sensitivity analyses included tests for heterogeneity, directional horizontal pleiotropy (MR-Egger intercept test), outlier-robust estimates (MR-PRESSO outlier-corrected estimate, weighted median estimate, and MR-Egger slope estimate), reverse causation, and directional concordance with body fat percentage (see **Methods**). No evidence of weak instrumental variables (suspected when F-statistics < 10) were found (**Supplementary Table 2**).

We found that BMI influenced 1,213 proteins, passing tests of significance, and the above sensitivity analyses (**Supplementary Table 3 and 4**). Hereafter, these 1,213

proteins are referred to as BMI-driven proteins (**Fig. 2a, 2b, and 2c and Extended Figure 1**).

For assessing directional concordance with body fat percentage, we performed MR to evaluate the effect of body fat percentage on the same 4,907 plasma proteins (**Methods**). We did this because body fat percentage is considered to be a more direct proxy of obesity, whereas BMI is an easy-to-measure, clinically relevant proxy.¹² However, the sample size available to assess the genetic determinants of BMI is larger than that of body fat percentage, providing more precise estimates. We found that body fat percentage influenced 94.7% of all BMI-driven proteins with the same direction of effect as BMI (**Fig. 2d**), illustrating a high concordance of results between the two different measures of obesity ($r = 0.92$; $P < 2.2 \times 10^{-16}$). Only proteins influenced by both BMI and body fat percentage in the same direction were retained for downstream analyses.

Corresponding Methods: page 28

Sensitivity analyses

...

For outlier-robust estimates, we used MR-weighted median, weighted mode, MR-Egger slope, and MR-PRESSO outlier-corrected estimates as supplementary analyses to evaluate the directional concordance of the effect. Proteins must have shown a directionally consistent effect across all methods, including the IVW, or otherwise were removed. Note that MR-PRESSO outlier-corrected estimates are generated only when MR-PRESSO detects outliers; otherwise it generates “NA” and thus not used in the analyses.

Comments from Reviewer 1

3. Include statement about MR analysis following MR-STROBE guidelines?
--

Thank you for pointing this out. We have included the statement about MR-STROBE and added the MR-STROBE checklist as **Supplementary Note**.

STROBE-MR statement

This study follows the STROBE-MR guidelines, and the STROBE-MR checklist can be found in the **Supplementary Note**.

Comments from Reviewer 1

5. pQTLs (and eQTLs) are notoriously difficult in terms of identifying the causal gene for disease because they are driven by non-coding variants. Although colocalization was one of the steps performed, to ensure the same causal genetic signal between pQTL and GWAS with cardio metabolic diseases, other more sophisticated fine-mapping approaches exist. How can we be assured that the circulating protein indicated in the pQTL is the underlying causal gene contributing to disease? Can the authors run other fine-mapping methods to ensure consistency with the coloc results?

With all due respect, pQTLs, including eQTL and pQTL, have been considered a valuable resource for mapping variants, including non-coding variants, to disease-causing genes^{11,13,14}. This is because the transcriptome and proteome are the central layers of information transfer from the genome to phenome¹³, which effectively enables linking variants to diseases¹³. In this case, the lead *cis*-pQTL for C-terminal COL6A3 in the deCODE cohort, rs11677932, was identified as a *cis*-pQTL specifically for C-terminal COL6A3, and not for any other protein, underscoring its specificity and validating its use as an instrumental variable for C-terminal COL6A3. We further investigated the profile of rs11677932 through fine-mapping, variant-to-gene mapping, and epigenetic data analysis.

Fine-mapping with SuSiE provides supporting evidence that rs11677932 is the shared causal variant for plasma C-terminal COL6A3 levels and CAD

To increase the robustness of the evidence, we conducted fine-mapping using SuSiE in the 500 kb region surrounding the index variant of the C-terminal COL6A3 pQTL (rs11677932). SuSiE results revealed that rs11677932 is within the 95% credible set and has the highest posterior inclusion probability for both C-terminal COL6A3 pQTL and CAD.

As additional evidence, we performed variant-to-gene (V2G) mapping with the Open Target Genetics and ENCODE databases.

The lead cis-pQTL of C-terminal COL6A3 (rs11677932) was mapped to COL6A3 with the highest V2G score

The Open Target Genetics database, which uses multiple lines of datasets to map variants to genes, has included eQTL and pQTL as key data sources (<https://genetics-docs.opentargets.org/our-approach/data-pipeline>). For example, the lead *cis*-pQTL for COL6A3, rs11677932, has the strongest evidence to be mapped to COL6A3 (https://genetics.opentargets.org/Variant/2_237315312_G_A/associations). This finding is supported by its designation as a pQTL in an independent study (not used in the current analysis; Sun et al., 2018), and the enhancer-TSS interactions as evidenced by FANTOM5 CAGE expression atlas, as well as its proximity to the transcription start site of the gene.

Epigenetic data supports rs11677932's regulatory role

Additionally, data from RegulomeDB provide evidence that rs11677932 is likely a regulatory variant, with multiple supporting data points yielding a RegulomeDB score of 1b (eQTL + transcription factor (TF) binding + TF motif + DNase Footprint + DNase peak). Notably, this variant disrupts the conserved nucleotide in the MEF2B transcription factor binding motif (**Fig. 6**), which is consistent with the reduced effect of this variant on the expression of COL6A3.

Regarding eQTLs, in the GTEx project, rs11677932 was identified as an eQTL for COL6A3 in the aorta, with effects that are directionally concordant with those observed in the pQTL analysis (rs11677932-A decreases COL6A3 gene expression and reduces COL6A3 protein levels).

Taken together, these findings suggest that rs11677932 likely reduces COL6A3 gene expression and thus protein levels, thereby providing strong support for its role as an instrumental variable for COL6A3 protein levels.

We have modified the manuscript accordingly.

Manuscript pages 17–20

The lead *cis*-pQTL for C-terminal COL6A3 in the deCODE cohort, rs11677932, was identified as a *cis*-pQTL specifically for C-terminal COL6A3, and not for any other measured protein, underscoring its specificity and validating its use as an instrumental variable for C-terminal COL6A3. We further investigated the profile of rs11677932 through fine-mapping, variant-to-gene mapping, and epigenetic data analysis.

Fine-mapping using SuSiE

SuSiE is a Bayesian fine-mapping method that relaxes the assumption of a single causal variant and allows for multiple causal variants. We conducted fine-mapping using SuSiE in the 500 kb region surrounding the index variant of the C-terminal COL6A3 pQTL (rs11677932) (see **Methods**). SuSiE results revealed that rs11677932 is within the 95% credible set and possesses the highest posterior inclusion probability for both the C-terminal COL6A3 pQTL and CAD (**Supplementary Table 17**).

Variant-to-Gene (V2G) mapping using the Open Targets Genetics

The Open Targets Genetics database facilitates variant-to-gene mapping based on multiple lines of evidence, including QTL analyses, chromatin interaction experiments, and proximity to the transcription start site (TSS). We queried the database for the variant and found that rs11677932 is mapped to COL6A3 with the highest V2G score. This finding is corroborated by its identification as a pQTL in an independent study (not included in the current analysis; Sun et al¹⁵), and by enhancer-TSS interactions in the FANTOM5 CAGE enhancer atlas¹⁶, as well as its closeness to the TSS of the gene. This provides supporting evidence that the variant is a valid instrumental variable for plasma COL6A3 levels (**Supplementary Table 18**).

Regulatory role of the lead cis-pQTL of C-terminal COL6A3

We also evaluated whether rs11677932 is a regulatory variant using ENCODE⁷ and RegulomeDB⁸. RegulomeDB assign a heuristic ranking score, representing its potential to be functional in regulatory elements⁹. rs11677932 had a RegulomeDB score of 1b—which is considered to be strong evidence for a regulatory variant—based on data from eQTL, transcription factor (TF) binding, TF motif, DNase Footprint, and DNase peak. The variant is located in an open chromatin region and an active enhancer domain in multiple tissues, including adipose tissues and arteries (**Fig. 6**).

Fig. 6. Epigenetic profile of the lead *cis*-pQTL for C-terminal COL6A3.

(a) LocusZoom plots of the pQTL for C-terminal COL6A3 in the 1Mb-region surrounding the lead *cis*-pQTL from the deCODE study, rs11677932.

(b) ATAC-seq, H3K4me3, and H3K27ac ChIP-seq data for adipose tissue, coronary artery, aorta, thoracic artery, and tibial artery. Data are publicly available at ENCODE and RegulomeDB.

(c) rs11677932 is predicted to affect the binding of a transcription factor MEF2B. ENCODE Accession ID: ENCSR782UOT; Target: BORCS8-MEF2B, MEF2B.

Comments from Reviewer 1

6. The focus of the study has been on using pQTL for MR for BMI effect on plasma protein levels. Identified and honed in on COL6A3. As a sensitivity analysis, has the authors considered performing MR analysis of BMI on COL6A3 gene expression levels, to show that BMI is influencing both mRNA and protein levels of COL6A3.

Thank you for your comment. We agree that it would be interesting to test whether BMI influences the gene expression of COL6A3. However, unfortunately, the GTEx database only includes *cis*-eQTLs, and therefore cannot be used for the outcome in two-sample MR. This limitation arises because BMI's genetic instrumental variables span across the entire genome, and the effect of BMI on the outcome is driven through a combination of trans-effects (as shown in **Fig. 2c** and also below).

Consequently, an outcome GWAS (in this case, eQTL) requires both *cis* and *trans* regions. *Trans*-eQTLs present multiple challenges, including a substantial increase in the burden of multiple testing, for which a large sample size is required. Thus, GTEx decided not to publish *trans*-eQTL and currently there is no large scale *trans*-eQTL datasets in multiple human tissues other than eQTLGen's whole blood eQTL, which is

less relevant in the current study given the limited expression of COL6A3 in the whole blood.

However, it should be noted that in COL6A3's step 1 MR, the causal effect of BMI on COL6A3 is through a combination of *trans* effects (as shown in **Fig 2c**), and the lead *cis*-pQTL of COL6A3 is not significant in the BMI GWAS. This finding is reassuring, suggesting that the direction of effect is from BMI to COL6A3, and not vice versa. Specifically, if COL6A3's pQTL were very significant in the BMI GWAS, it would be more challenging to determine the direction of effect, as we cannot rule out the possibility of reverse causation, i.e., the effect of COL6A3 on BMI. We have included this consideration in the manuscript.

T

Comments from Reviewer 1

7. I understanding the focus on the paper is on circulating proteins, but drug discovery and validation is typically best supported not by just one genetic association signal but a series of genetic association signals via the allelic series model. Is it possible to test if there are low frequency and/or rare coding variants in COL6A3 are associated with CAD?

Thank you for the valuable comment. This is indeed an interesting point. It should be noted that the effect of COL6A3 is driven by its C-terminal—the endotrophin-containing fragment. Therefore, in response to your feedback, we conducted an exploratory REGENIE analysis to test the effect of deleterious variant burden of the C-terminal of COL6A3 (also known as the Kunitz domain, which is cleaved into endotrophin) on CAD using the UK Biobank whole-exome sequencing data, adjusted for age, sex, genotyping array, recruitment center, and the first 20 genetic principal components. Unfortunately, we found only a limited number of deleterious variants in the corresponding coding region (2 loss-of-function [LoF] variants and 5 likely deleterious missense variants, all of which have a MAF < 0.00001; PMID: 34662886). The combination of a relatively small number of rare variants at very low frequencies and the relatively small number of CAD cases (18,995 cases and 362,056 controls that underwent whole-exome sequencing, following definitions in Inouye et al.; PMID: 30309464) resulted in poor statistical power, as only 12 individuals were carriers of any LoF or likely deleterious missense variants. Unsurprisingly, no association was detected between this burden of the C-terminal of COL6A3 and CAD (p-value = 0.51). Overall, domain-aware burden testing in this context requires a substantial increase in sample size to achieve decent power, particularly for a binary outcome like CAD. This remains an area for future research.

However, we note that using different *cis*-pQTL from different studies (deCODE, Fenland, ARIC, UK Biobank) and different platforms (SomaScan in deCODE, Fenland, and ARIC

as well as Olink in the UK Biobank) allowed us to triangulate the findings, and all analyses robustly supported the causal effect of C-terminal COL6A3 on CAD.

Comments from Reviewer 1

8. The domain-aware MR is quite intriguing and interesting way to determine that the C-terminal end of COL6A3 explains the effect of COL6A3 levels on CAD.
--

Thank you very much. To the best of our knowledge, this is the first example of utilizing domain-aware MR to disentangle the domain-specific biology. Reassuringly, we also confirmed with Olink that their assay targets the C-terminal of COL6A3.

In the revised manuscript, we further investigated the causal effect of MRI-derived body fat compartments (subcutaneous adipose tissue; SAT, visceral adipose tissue; VAT, and gluteofemoral fat; GFAT) on plasma levels of C-terminal and N-terminal COL6A3. Our additional MR analyses identified SAT as the key fat compartment influencing plasma C-terminal COL6A3 levels. This finding aligns with previous research indicating that the collagen matrix in subcutaneous fat, rich in C-terminal COL6A3, is involved in adipose tissue fibrosis in the event of obesity, leading to insulin resistance, and metabolic dysfunction through endotrophin release. Therefore, our results, alongside existing studies, provide evidence that obesity leads to increased subcutaneous fat, elevating C-terminal COL6A3 levels and endotrophin, adipose fibrosis, insulin resistance, and coronary artery disease risk.

We believe these findings help to demonstrate the power of proteogenomics and represent one of the key findings of our paper. We have included a new figure of forest plots to visually illustrate the domain-specific causal effect of COL6A3.

Manuscript pages 15–17

Body fat compartments influencing plasma levels of C- and N-terminal COL6A3 levels

We conducted additional MR analyses to evaluate which body fat components specifically influence plasma C- and N-terminal COL6A3 levels, using GWAS of MRI-derived fat compartment volumes for two-sample MR (**Fig. 5e**). Among the three fat compartments analyzed—abdominal subcutaneous adipose tissue (ASAT), visceral adipose tissue (VAT), and gluteofemoral adipose tissue (GFAT)—ASAT was found to significantly increase plasma levels of both C- and N-terminal COL6A3, with a more

pronounced increase in C-terminal COL6A3 (**Fig. 5e; Supplementary Table 14**). Notably, this finding aligns with previous studies showing that (i) the collagen matrix surrounding subcutaneous fat is rich in C-terminal COL6A3¹, which in turn releases endotrophin and (ii) increased expression of *COL6A3* in the subcutaneous adipose tissue and increased plasma abundance of endotrophin, the cleaved product of the C-terminal of COL6A3, are associated with adipose tissue fibrosis, insulin resistance, and metabolic dysfunction²⁻⁶.

Figure 5. Follow-up analyses for collagen type VI $\alpha 3$ (COL6A3).

(a) Schematic illustration of proposed relationship between obesity, COL6A3 (Collagen type VI $\alpha 3$ chain), endotrophin, and coronary artery disease. Obesity leads to increased

production of COL6A3, whose C-terminal is cleaved into an active form, termed endotrophin, which increases the risk of coronary artery disease.

(b) Schematic diagram of COL6A3 (UniProt ID: P12111). COL6A3 consists of a short collagenous region flanked by multiple von Willebrand factor type A (vWF-A) modules (N1–N10 in the N-terminal and C1,2 in the C-terminal). There are three additional C-terminal domains unique to COL6A3 (C3–C5), which are not present in other collagen type VI families. The most C-terminal domain (C5) is cleaved into a soluble protein termed endotrophin.

The two amino acid sequences targeted by the aptamers to measure COL6A3 levels are as follows: the N-terminal-binding aptamer targets the amino acid sequence 26–1036 (uncleaved section), while the C-terminal aptamer targets the amino acid sequence 3108–3165 (cleaved section). The figure has been modified from ref^{17,18}.

(c) MR for the effect of C- and N-terminal COL6A3 on the risk of CAD stratified by C- and N-terminal COL6A3.

(d) MR for the effect of BMI and body fat percentage on COL6A3 stratified by C- and N-terminal COL6A3.

(e) MR for the effect of body fat compartments on COL6A3 stratified by C- and N-terminal COL6A3.

We used MRI-derived GWAS on abdominal subcutaneous adipose tissue (ASAT), visceral adipose tissue (VAT), and gluteofemoral adipose tissue (GFAT) from 40,032 individuals in the UK Biobank, which were reported by Agrawal et al.¹⁹. The two-sample MR method was as described in step 1 MR analysis.

Comments from Reviewer 1

9. The GTEx results don't seem that convincing. COL6A3 is expressed in adipose, and coronary artery tissue which are disease relevant. However, it's also expressed in several other tissues that aren't notably relevant to cardio metabolic diseases such as colon tissue. In fact, the gene is expressed in all tissues except brain tissues. Therefore, it's difficult to say that the gene is tissue-specific in the disease relevant tissues. Can consider moving this figure to the supplement.
--

Thank you for the feedback. We moved the figure to the supplementary material.

Comments from Reviewer 1

Minor comments:

10. Figure 2B: Would be nice to highlight positive controls for MR analysis of BMI on plasma protein - Have any of these been confirmed in previous studies?
--

Thank you for the important comment and suggestion. Indeed, many well-known associations between BMI and plasma proteins have been captured by step 1 MR. For example, leptin (LEP) has been found to be positively associated with BMI as early as 1995²⁰, and it has been found to decrease with weight loss in both humans, aligning with our MR findings. Fatty acid-binding protein 4 (FABP4), also known as adipocyte FABP (A-FABP), has been found to be reduced by body weight reduction in individuals with obesity²¹. Another notable example is CRP, an established marker of inflammation. Obesity is known to be associated with chronic low-grade inflammation and increased plasma CRP levels in humans^{22 23}, which also aligns with our findings. We have modified the manuscript accordingly.

Manuscript pages 8–9

Step 1 MR results captured multiple well-known associations between BMI and plasma proteins. For example, leptin (LEP in **Fig. 2b**) has been found to be positively associated with BMI as early as 1995²⁰, and its levels have been observed to decrease with weight loss in humans, aligning with our MR findings. Fatty acid-binding protein 4 (FABP-4) has been shown to be reduced by body weight reduction in individuals with obesity²¹. Another notable example is C-reactive protein (CRP), an established marker of inflammation. Obesity is known to be associated with chronic low-grade inflammation and increased plasma CRP levels in humans^{22 23}, which also aligns with our findings.

Comments from Reviewer 1
Minor comments: 11.- The PCSK9 positive control example is a nice validation of the approach - perhaps label this on Figure 2B

Thank you for the valuable feedback. We have labelled PCSK9 in the Figure 2B.

Response to Reviewer 2

Comments from Reviewer 2

In the current manuscript "COL6A3-derived endotrophin mediates the effect of obesity on coronary artery disease: an integrative proteogenomics analysis", Yoshiji et al. attempt to integrate published results from genomics and proteomics studies in order to identify plasma proteins affecting cardio- metabolic diseases through their effect on body mass index. Following mendelian randomization analysis, the authors assess and claim a causal role of obesity on the level of a certain number of proteins, including COL6A3 plasma levels. They further tested through MR the effect of the levels of these proteins on cardiometabolic outcomes including CAD and claim a causal effect of COL6A3 levels on CAD. Currently, the results are not fully convincing. Whereas the topic could have been of interest, the way the analysis is performed and the results are presented do not provide clarity and conviction.

We thank Reviewer 2 for the insightful comments. We appreciate your thorough feedback, which greatly helped us design new analyses and increase the robustness and coherence of the study. We have modified the manuscript substantially based on your feedback. Please find our point-by-point response below.

Comments from Reviewer 2

1) It appears that the attempt of finding specific proteins that have their level influenced by BMI did not bring a successful narrow selection. In the abstract and through parts of the text, the authors omit to mention and interpret that a large majority of tested proteins or 2,714 of 4,907 have their level influenced by BMI in a causal manner and that they consider "BMI driven proteins" (including COL6A3).

2) The whole step 1 of the study is then becoming difficult to use. The authors need to realize that majority and protein have their level affected causally by BMI. Whereas I understand the focus on certain locus where variant have been associated with CAD, they have to realize that a vast majority of proteins have their level influenced by BMI so it is not pointing to a specific set, or not that meaningful.

Thank you for your important comment. We believe Step 1 MR is crucial for highlighting the role of obesity in the plasma proteome. Aligning with our findings, multiple independent studies have reported substantial changes in plasma protein levels associated with obesity, using BMI as a reference²⁴⁻²⁷. By including Step 1 MR in our integrative analyses, we were able to highlight the significance of C-terminal COL6A3 and endotrophin in a hypothesis-free manner in the context of influence of obesity. However, we respect your comment that the selection step could be more rigorous. Therefore, we have made the following changes:

(i) Bonferroni correction:

We adopted more stringent filtering by transitioning from FDR to Bonferroni correction in Steps 1 and 2 MR. We acknowledge that this approach is very conservative, given that many protein levels are not independent. However, we opted for this to safeguard against false-positive findings.

(ii) Directional concordance with body fat percentage:

We assessed for directional concordance effects between BMI and body fat percentage using MR to assess the effects of body fat percentage and BMI on plasma protein levels. Here, we required proteins to show consistent directional effects (i.e., the same beta sign as in MR with BMI) with at least nominal significance. Please refer to **Extended Fig. 1**, "Filtering Flowchart for Step 1 MR" (attached below).

(iii) Identifying the specific body fat compartments that increase plasma levels of C-terminal COL6A3, which is cleaved into endotrophin

To further understand the relationship between obesity and C-terminal COL6A3 levels, we evaluated the specific body fat compartments that increase C-terminal COL6A3 levels, which provided new insights. Specifically, we investigated the causal effects of MRI-derived body fat compartments (subcutaneous adipose tissue [SAT], visceral adipose tissue [VAT], and gluteofemoral fat [GFAT]) on plasma levels of C-terminal and N-terminal COL6A3. Our additional MR analyses identified SAT as the key fat compartment influencing plasma C-terminal COL6A3 levels. In fact, this finding aligns with previous research indicating that the collagen matrix of subcutaneous fat is rich in C-terminal COL6A3, and COL6A3-derived endotrophin triggers adipose tissue fibrosis and metabolic dysfunction.

In combination with these existing studies, our additional MR results suggest an intriguing narrative: obesity leads to increased subcutaneous fat, which is rich in C-terminal COL6A3, leading to increased endotrophin release, eventually contributing to increased risk of coronary artery disease.

While we acknowledge that BMI appears to have a large effect upon the circulating proteome, this does not mean that single, specific proteins cannot have biologically relevant effects upon disease. As Reviewer 2 is aware, BMI is associated with hundreds of diseases, and it is likely that one of the ways that obesity exerts its toll upon the human body is through the dysregulation of circulating protein levels. While BMI does influence a large proportion of the proteome, this fact does not mean that some of these proteins do not cause disease. The reason we have undertaken this study, of course, is to identify such proteins that increase the risk of CAD.

We once again greatly appreciate your comments, which were crucial for initiating this additional analysis, which allowed us to intertwine our findings with previous studies and tell a coherent and intriguing story. We have modified the manuscript accordingly.

Comments from Reviewer 2

3) The fraction of protein presented in the step 1 represents 55.3% of proteins tested with aptamer, even if the authors use a stringent FDR of 0.5%. It would be interesting to see what is the number of proteins at an FDR of 5%.
--

I think that readers in general would have difficulty to accept that step 1 of the study is giving a meaningful result.

We would like to note that previous studies from independent groups also found substantial influence of BMI on plasma proteome, such as the one from the UKB-PPP consortium²⁶. However, to further safeguard against false positives, we applied a more stringent multiple testing correction using Bonferroni instead of FDR, and further filtered the data for concordance with body fat percentage. Please refer to our response to Comments 1 and 2.

Comments from Reviewer 2

Try to present convincing evidence for cis pQTL for COL6A3 in different methods with locus plot for Somascan (2 probes) and Olink as well as CAD or related diseases.

4) Also, since assay on such platform are not always succeeding to measure the attempted levels, in large part due to the low concentration in plasma of these proteins, thus this fraction can even be underestimated.

Thank you for the feedback. We have included the LocusZoom plots to illustrate the shared causal signal between *cis*-pQTL of the C-terminal COL6A3 from deCODE and the largest CAD GWAS from Aragam et al (**Fig. 3c**). This was supported by colocalization using coloc and SuSiE (both showed posterior probability of the shared signal > 0.99), which we performed as an additional analysis (**Supplementary Table 15**).

Additionally, to have multiple sources of evidence, we also included the LocusZoom plots for the same regions using multiple sources: *cis*-pQTL for COL6A3 from the UK Biobank, which used Olink Explore assay, and the coronary artery disease from the UK Biobank.

Notably, deCODE (Icelandic cohort) and Fenland (British cohort), both of which used aptamer-based SomaScan v4 assay, identified the same variant (rs11677932) as the lead variant for C-terminal COL6A3. Additionally, the UK Biobank, which used an antibody-based Olink Explore assay, identified *cis*-pQTL variant (rs1050785) that was in moderate LD with rs11677932 ($R^2 = 0.73$). MR using *cis*-pQTL from the UK Biobank with the Olink assay consistently showed that an s.d. increase in C-terminal COL6A3 level was associated with increased odds of CAD (OR = 1.30, 95% CI: 1.17–1.45, $P = 2.4 \times 10^{-6}$). Thus, two separate ways of measuring COL6A3 generated similar results.

Furthermore, rs11677932 is a *cis*-eQTL for COL6A3 in the aorta with directionally consistent effects in GTEx v8 (i.e., rs11677932 leads to decreased COL6A3 expression and decreased plasma protein abundance). These findings also support our main findings.

These results provide robust evidence for the causal effect of C-terminal COL6A3 on CAD. We have modified the manuscript accordingly.

Manuscript pages 11–12

Figure 3. MR analyses for the effect of BMI-driven proteins on cardiometabolic diseases.

(a) Flow diagram of the Step 2 Mendelian randomization (MR) analyses.

(b) Forest plots for the effect of body mass index (BMI)-driven proteins on four cardiometabolic diseases (coronary artery disease, ischemic stroke, cardioembolic stroke, type 2 diabetes). The MR analyses were conducted using the largest available GWAS of coronary artery disease²⁸ (181,522 cases and 1,165,690 controls), ischemic stroke (34,217 cases and 2,703,029 controls), cardioembolic stroke²⁹ (7,193 cases and 2,703,029 controls), and type 2 diabetes³⁰ (80,154 cases and 853,816 controls).

PP.H4 = Posterior probability of having the shared causal variant (hypothesis H4 in colocalization).

(c) LocusZoom plots of the pQTL for C-terminal COL6A3 in the 500kb-region surrounding the lead *cis*-pQTL, rs11677932.

Comments from Reviewer 2

5) It can be noted that based on the aptamer-based study that the authors used, an even greater fraction of proteins associates with BMI directly (87%). It should also be pointed that there is a number of other proteins that have an even more significant direct association with BMI than COL6A3. Whereas many proteins are displayed on figure 2b, COL6A3 is not. This suggests that the rank of COL6A3 is lower.
--

6) The protein with levels associates the most strongly with BMI were LEP, FABP3 and FABP4. The levels of these proteins associating most strongly to BMI when directly associated, where also directly associated strongly to coronary artery disease in the report data from Ferkingstad.

Thank you for the feedback. In fact, C-terminal COL6A3 ranks relatively high in terms of p-value (19th out of 4,907 proteins tested in Step 1 MR). We have modified **Fig. 2b** to clarify this point. We also included PCSK9 as suggested by Reviewer 1. However, we would like to remind the reviewer that our goal is not to find the proteins most strongly regulated by BMI. Rather, our goal is to identify the proteins which are regulated by BMI and influence risk of CAD. COL6A3 fulfills these criteria.

Fig. 2 (b) in page 9

Notably, top proteins such as LEP, FABP3, and FABP4 did not pass Step 2 MR, suggesting there is no shared causal signal between them and CAD outcomes. Indeed, the LocusZoom plots of the *cis*-pQTL for these proteins and the corresponding region of CAD demonstrate the lack of a shared causal variant in the *cis*-region of these proteins in the CAD GWAS.

Reviewer-only Figure. LocusZoom plots for *cis*-pQTL of LEP, FABP3, and FABP4

Comments from Reviewer 2

7) Similar to figure 2c, they should think of looking at association of BMI variants and LEP, FABP3 and FABP4 levels and would notice that they are not different in interpretation.

8) Thus, for majority of tested proteins (at least 55%), BMI has a causal role and talking about BMI driven proteins appears misleading when it is such a massive effect all over the proteome.

Thank you for the comments. We have modified the manuscript to highlight that proteins that are strongly influenced by BMI do not necessarily have causal effects on CAD, as highlighted by top proteins such as LEP and FABP3. Another great example is CRP. It is well-known that obesity leads to increased CRP levels, as captured by Step 1 MR, but plasma CRP levels are not causal for CAD. Therefore, we believe the combination of step 1 and 2 successfully narrowed down the list of proteins that link the effect of obesity to CAD.

Comments from Reviewer 2
9) The association of variants at COL6A3 with CAD has been present in summary statistics for 6 years (Van der Harst P 2017) and also more recently by Koyama (2020) at genome-wide significance according to Open Target Genetics (https://genetics.opentargets.org/Gene/ENSG00000163359/associations?colocTraitFilter=Coronary artery disease&traitFilter=Coronary artery disease)
23) The pQTL and CAD results are published at genome-wide significance at COL6A3 - again I would emphasize that as in the result the author should make it more clear.

We appreciate these valuable comments. We have modified the manuscript to clearly state that previous CAD GWAS identified COL6A3. However, we note that there are many challenges to map GWAS signals, especially non-coding signals like the one for this region, to causal mechanisms. Our study integrated multiple lines of evidence and provide novel insights into the causal pathway in combination: obesity, particularly increased subcutaneous fat, leads to increased C-terminal COL6A3 levels, which is cleaved into endotrophin, which increases CAD risk.

Manuscript page 14

Identification of the causal domain of COL6A3

The COL6A3-encoding gene, COL6A3, has been identified as a putatively causal gene in multiple CAD GWAS studies CAD GWAS^{28,31,32}; however, its mechanism or causal domain has not been fully described.

Comments from Reviewer 2
10) When presenting COL6A3 association to CAD and protein level, more detailed and convincing reasoning and analysis are required. Since the step 1 of the study is not convincing, I would start with trying to perform MR or colocalization of all reported CAD variants with proteomics data.
11) In the chapter starting at line 218, the authors should present a locus plot of the cis association of COL6A3 plasma levels.
12) They should also present a locus plot of the association of variants to CAD at COL6A3.
15) The authors should show a colocalization plot between COL6A3 levels and CAD using variants at COL6A3 locus.

Thank you for your comment. We believe that modification made to Step 1 MR based on your feedback strengthen Step 1 MR (i.e., transition to Bonferroni correction, triangulation with body fat percentage, additional MR analyses to pinpoint the causal effect of subcutaneous fat on increased plasma levels of C-terminal COL6A3). To

address the other comments, we also included the LocusZoom plots (**Fig. 3c** (shown above) and **Extended Fig. 2** (shown below)), which visually illustrate the shared causal signal between C-terminal COL6A3 and CAD, which were further supported by coloc and SuSiE analyses.

Extended Fig. 2. LocusZoom plot for cis-pQTL of COL6A3 and coronary artery disease.

Extended Fig. 2. LocusZoom plots of the region surrounding the lead *cis*-pQTL of COL6A3 (rs11677932).

(a) LocusZoom plot of the pQTL for C-terminal COL6A3 in the 500 kb-region surrounding the lead *cis*-pQTL from the deCODE study, which used SomaScan v4 assay. (b) pQTL for C-terminal COL6A3 in the same region from the UK Biobank, which used Olink Explore 3072 assay²⁶. rs1050785 was the lead *cis*-pQTL in this

pQTL, which was in LD ($R^2 = 0.73$) with rs11677932. (c) Coronary artery disease GWAS from Aragam et al. in the same region³¹.

Additionally, to provide further evidence, we performed a Cox regression analysis that evaluated the association between baseline C-terminal COL6A3 level and the cumulative incidence of CAD. We found that an s.d. increase in plasma C-terminal COL6A3 levels was associated with an increased cumulative incidence of CAD (HR = 1.21, 95% CI: 1.16–1.25, $P < 2.2 \times 10^{-16}$). We added this in the revised Manuscript as below.

Manuscript pages 13–14

Cox regression analysis for cumulative incidence of CAD

As an additional orthogonal evaluation, we performed multivariable Cox proportional-hazards regression analysis to evaluate the association between plasma C-terminal COL6A3 level and up to 10 years of cumulative incidence of CAD in 35,100 individuals (2,969 cases and 32,131 controls) from the UK Biobank (**Supplementary Table 12**). After adjustment for age, sex, recruitment center, and protein sample processing time, we found that an s.d. increase in plasma C-terminal COL6A3 level was associated with an increased cumulative incidence of CAD (HR = 1.40, 95% CI: 1.35–1.45, $P < 2.2 \times 10^{-16}$).

Finally, we plotted the Kaplan-Meier estimates for the cumulative incidence of CAD stratified by baseline COL6A3 level quantiles (25%, 50%, 75%, and 100%; **Fig. 4**). The results showed statistically significant differences, with the top 100% group having the highest incidence and the 25% the lowest (log-rank test $P < 2.2 \times 10^{-16}$).

Kaplan-Meier estimates of cumulative incidence of coronary artery disease by baseline COL6A3 level quantiles in the UK Biobank

Q1 represents the lowest 25% group, Q2 the 26–50% group, Q3 the 51–75% group, and Q4 the highest quantile group (76–100%, covering from the 75th percentile to the maximum value) based on baseline plasma COL6A3 levels. Shaded areas around the lines represent 95% confidence intervals.

Comments from Reviewer 2

13) They should present and discuss trans pQTL COL6A3 association and discuss them (from any study in the literature).

14) The authors should show an effect/effect plot for all variants correlating with COL6A3 in cis and trans and perform and MR not limiting to cis pQTL variants.

Trans-pQTLs are highly susceptible to horizontal pleiotropy, and the use of *cis*-pQTL is recommended for pQTL MR^{33 34 35}. This is supported by the observation that many proteins' pQTLs typically have very strong peaks in the *cis* region and do not have hundreds of peaks across the genome (i.e., the genetic architecture tends to be oligogenic rather than polygenic). Therefore, if we include *trans* variants as IVs, the relatively small number of IVs each protein has will make them susceptible to bias because of unbalanced horizontal pleiotropy. Thus, since *cis*-pQTL MR are less prone to bias, these results are reported in the manuscript.

Comments from Reviewer 2

16) The authors need through the text line 117-121 of introduction to clearly indicate that they perform mendelian randomization on existing pQTL data. Then they need to clearly indicate if it is aptamer (and what normalization of the data has been performed; SMP or no SMP) or antibody-based methods.

17) The authors need to indicate that they are mainly using existing CAD reported data and they should clearly show what are the best signal at COL6A3 locus with CAD.

Data source of pQTL and the assay used

We appreciate your feedback. We have clarified the datasets we used for our analyses. Please kindly note that we used both published data and unpublished data. Specifically, we generated sex-stratified pQTL of COL6A3 using individual-level data from the UK Biobank for sex-stratified analyses, and we also performed observational analysis using individual-level data from the EPIC-Norfolk, in collaboration with colleagues from the University of Cambridge.

Manuscript page 5

Our analysis builds upon data from the deCODE¹¹, Fenland¹³, ARIC³⁶, and EPIC-Norfolk studies using the SomaScan v4 assay, and the UK Biobank²⁶ using the Olink Explore 3072 assay. We also used BMI GWAS from the GIANT and UK Biobank consortia³⁷, CAD GWAS from the CARDIoGRAMplusC4D Consortium³¹, and type 2 diabetes GWAS from the DIAMANTE Consortium³⁰ (**Supplementary Table 1**). In follow-up analyses, we used individual-level data from the UK Biobank, EPIC-Norfolk, and the CellGenBank Cohort³⁸. We also used epigenomic data from RegulomeDB⁹ and ENCODE⁷.

Regarding the assays used in each study, we have clarified that deCODE, Fenland, and ARIC cohorts used aptamer-based SomaScan v4 assay, while the UK Biobank used antibody-based Olink Explore 3072 assay. Additionally, discussions with SomaLogic confirmed that, among the studies using the SomaScan v4 assay, only the deCODE study¹¹ did not perform the sample normalization³⁹ procedure, known as Adaptive Normalization by Maximum Likelihood (ANML), which adjusts for inter-sample technical and biological variability in total signal within and between runs (https://somalogic.com/wp-content/uploads/2022/07/SL00000442_Rev4_2021-07_-_SomaScan-v4.0-and-v4.1-Data-Standardization.pdf). We clarified these in the Manuscript and in **Supplementary Table 1**. Although the deCODE study did perform ANML normalization, it is important to note that we replicated the COL6A3 findings from

Step 1 MR and Step 2 MR using pQTL from Fenland, ARIC, and UK Biobank, demonstrating the robustness of our findings.

Supplementary Table 1.

Plasma protein levels	deCODE (Feringstad et al.)	·GWAS of 4,907 plasma protein levels in 35,559 Icelanders. · SomaScan v4 assay (without ANML*) ·1,881 cis -pQTL were identified.
	UK Biobank (Sun et al.)	·GWAS of plasma COL6A3 level in individuals of European ancestry in 34,557 individuals of European ancestry. ·Olink Explore 3072 assay
	Fenland (Pietzner et al.)	·GWAS of plasma COL6A3 level in individuals of European ancestry in 10,708 individuals of European ancestry. ·SomaScan v4 assay (with ANML*)
	ARIC (Emilsson et al.)	·GWAS of plasma COL6A3 level in individuals of European ancestry in 7,213 individuals of Icelandic ancestry. ·SomaScan v4 assay (with ANML*)
	EPIC-Norfolk	· Plasma COL6A3 level in individuals of European ancestry (207 coronary artery disease cases and 827 controls).

ANML = Adaptive Normalization by Maximum Likelihood. This is a method developed by SomaLogic to adjust for inter-sample technical and biological variability in total signal within and between runs. Further details can be found at https://somalogic.com/wp-content/uploads/2022/07/SL00000442_Rev4_2021-07_-SomaScan-v4.0-and-v4.1-Data-Standardization.pdf

The best signal in the COL6A3 region in the CAD GWAS

We have clarified that we used the CAD GWAS is from Aragam et al. and presented that rs11677932 (the lead *cis*-pQTL of COL6A3 from the deCODE study) is the lead signal in the region (**Fig. 3c** and **Extended Fig. 2**; please see above).

Comments from Reviewer 2
18) The authors restricted instrumental variables to genetic variants that were cis -pQTLs to only one protein. Can they please in main text underline what fraction is removed? I assume it is very limited.

We have added **Extended Fig. 1**, which shows the histogram of the number of proteins that each variant is associated with in a *cis* manner. Although the majority of the variants are associated with one protein only, there were a few variants that have up to 5 *cis*-associated proteins. These variants may violate the MR assumption of no horizontal pleiotropy, and they are likely not valid proxies. Thus, we believe our restriction to *cis*-pQTLs reduces the possibility of this source of bias.

(b) Step 2 MR filtering:

The pQTL variants that are associated with more than one protein in *cis* were removed.

Extended Fig. 1.

(b) A histogram showing the number of the pQTL variants that were associated with more than one protein in *cis* and thus removed before Step 2 MR. Further details can be found in **Methods**.

Comments from Reviewer 2

19) In figure 3b and in abstract and line 237, the authors show a predicted effect on CAD of on s.d. in COL6A3 genetically determined level by cis pQTL. I think this can easily be misleading, since they have a genetic instrument, they should show for the best cis pQTL variant what is the effect on CAD and what its colocalization with CAD signal is.

Thank you for the feedback. We have included the LocusZoom plots in **Fig. 3c** and **Extended Fig. 2** in the revised manuscript (also shown above). Regarding the reporting of the effect of C-terminal COL6A3 on CAD risk, we would like to note that this is a common practice in pQTL MR papers to report the OR per one s.d. increase in the plasma protein levels, as we have done in multiple studies (Zhou et al Nat Med 2021³⁵, Yoshiji et al. Nat Metab 2023³⁴) as have many studies from other groups (Zhao et al. Cell Genom 2022³³, Pietzner et al. Nat Comm 2022⁴⁰).

Comments from Reviewer 2

20) Concerning the example that the authors use for PCSK9, the authors have to be very careful in at least two ways that would not support their claim. First, whereas PCSK9 levels associate with variants in cis, the strongest signal and few other at the locus are represented by coding variants. Coding variants that are cis pQTL are sometimes influenced by epitope binding effects. Thus, it is not clear that the Arg46Leu variant associating with CAD associates with expression level based on the Somascan experiment. Second, whereas obesity influence PCSK9 plasma level, it is not clear to me that there is a large relative contribution of BMI to the CAD risk mediated by PCSK9.

Thank you for the valuable feedback. We modified the manuscript's limitation section to mention that some MR analyses could be influenced by epitope-binding effects. However, we note that MR for the effect of PCSK9 on CAD used 5 *cis*-pQTL as instrumental variables with the IVW, which reduced the risk of bias due to the epitope-binding effects. Additionally, the causal effect of plasma PCSK9 levels on CAD is well established, and multiple MR studies and clinical trials reported the associations, aligning with our findings⁴¹⁻⁴³. We also note that the *cis*-pQTL of C-terminal COL6A3 is in a non-coding region. Further, the same SNP is also an eQTL in aorta for COL6A3 with directionally consistent effects. This supports the validity of the instrument and MR results for C-terminal COL6A3 and that it is unlikely to be influenced by epitope binding effects.

For the relative contribution of BMI to the CAD risk mediated by PCSK9, we agree that obtaining precise estimates is challenging, and we refrained from emphasizing the proportion estimates. Please see **Supplementary Note 4** for the full discussion, which is also attached at the end of this letter. We have clarified this in the limitation section.

We also note that a post-hoc analysis of FOURIER trial⁴¹ showed that people with higher BMI have numerically greater reduction in CAD risk, although the study was not powered to detect the difference. Further trials are needed to evaluate whether people with obesity benefit more from PCSK9 inhibitors in reducing CAD risk.

We also note that these clinical trial results are consistent with our MR results.

We have modified the manuscript as per the below:

Manuscript page 26

Fourth, we refrained from emphasizing the proportion by which C-terminal COL6A3 mediates the causal effect of BMI on CAD risk, given the challenges associated with using cis-MR mediation analysis for this estimation⁴⁴ (see **Supplementary Note**).

Manuscript page 26

Our study also highlighted and replicated other proteins, such as PCSK9. It is well established that increased plasma levels of PCSK9 elevate the risk of CAD, and inhibition of PCSK9 has been demonstrated to reduce CAD risk in multiple large clinical trials⁴¹⁻⁴³. Additionally, a previous MR study from the INTERVAL cohort reported that BMI increases plasma PCSK9 levels, and an observational analysis in the UK Biobank study also showed that BMI is positively associated with plasma PCSK9 levels ($P < 2.2 \times 10^{-16}$)²⁶. Furthermore, a 7-day caloric restriction was shown to result in decreased PCSK9 levels⁴⁵. These reports may align with our finding that PCSK9 at least partially mediates the effect of BMI on CAD risk. However, to date, no clinical trial has specifically tested whether individuals with obesity derive greater benefit from PCSK9 inhibitors. A secondary analysis of the FOURIER randomized clinical trial⁴⁶ found that individuals with metabolic syndrome had greater absolute risk reduction (ARR) in cardiovascular events, though not reaching statistical significance. Specifically, among individuals without diabetes, those with metabolic syndrome had an ARR of 2.5% with evolocumab therapy, compared to an ARR of 0.9% in those without metabolic syndrome. Given that the trial was not powered to detect differences between these subgroups, further research is necessary to assess whether people with metabolic syndrome or obesity might benefit more from PCSK9 inhibitors in reducing CAD risk.

Manuscript page 27

...

Fifth, for some proteins, the influence of epitope-binding effects cannot be ruled out, where variants directly affect the binding of aptamers or antibodies to the epitope. A recent study suggested that some *cis*-pQTL may reflect epitope effects rather than true protein level changes, especially when the *cis*-pQTL are protein-altering variants (PAV) or in LD with PAV and lacks associated eQTL³⁹. However, we note that this was not the case for the lead *cis*-pQTL of C-terminal COL6A3 from the deCODE study¹¹.

Specifically, rs11677932 was not PAV or in LD with PAV, and the variant was a *cis*-eQTL for the aorta in GTEx v8 (see **Methods**), indicating that the variant likely reflects true changes in protein abundance rather than epitope-binding artifacts.

Comments from Reviewer 2

21) When discussing the two SomaScan probes from SomaScan, the authors should clearly name them in the main text.

We have included the SomaScan's SeqID in the manuscript along with their target amino acid sequences.

Manuscript page 8

Notably, some proteins were targeted by more than one protein-targeting aptamer. This was the case for collagen type VI $\alpha 3$ (COL6A3), whose C-terminal and N-terminal regions were targeted by aptamers. For clarity, we will refer to protein-targeting aptamers as "proteins" unless specified otherwise. Specifically for COL6A3, we will refer to the C-terminal COL6A3-targeting aptamer (SeqID: 11196-31) as "C-terminal COL6A3" and the N-terminal COL6A3-targeting aptamer (SeqID: 10511-10) as "N-terminal COL6A3".

Manuscript page 32 (Methods)

Identification of the causal domain of COL6A3

Target region of the SomaScan v4 assay and the Olink Explore 3072 assay

We used the SomaScan Menu 7K (<https://menu.somallogic.com/>) to determine the target amino acid sequence of two aptamers for COL6A3 from on SomaScan v4 assay with additional support from SomaLogic (Boulder, Colorado, USA). We also obtained data on the target region of Olink Explore 3072 assay from Olink (Uppsala, Sweden). In SomaScan v4 assay, two aptamers target COL6A3: one for the C-terminal of COL6A3, also known as Kunitz domain (UniProt ID: P12111, target amino acid sequence: 3108-3165) and another for the N-terminal (UniProt ID: P12111, target

amino acid sequence: 26-1036). The seqID for the C- and N-terminal targeting aptamers are 11196-31 and 10511-10, respectively. In Olink Explore 3072 assay, the assay targets the C-terminal Kunitz domain of COL6A3 with polyclonal antibody (OID20292:v1).

Comments from Reviewer 2

22) Surprisingly, the authors insist in many words on the value of the mediation analysis, but they restrict the numerical result to the supplementary tables. We note that the COL6A3 mediation would have survived a less than 2000 tests. We also note that the authors have 2,714 "BMI proteins" and 4 diseases that they test.

Also in response to:

Comments from Reviewer 2

27) Overall, I believe that the BMI angle is very unclear and not specific at all. I would recommend shortening the story and make it to the point of COL6A3 and CAD.

28) Please remove the first part and all mention to BMI as a prior as well as mediation analysis.

Thank you for the valuable feedback. We believe that modifications we have made to Step 1 MR based on your feedback strengthen Step 1 MR (i.e., transition to Bonferroni correction, triangulation with body fat percentage, additional MR analyses to pinpoint the causal effect of subcutaneous fat on increased plasma levels of C-terminal COL6A3): obesity, specifically increased subcutaneous fat, leads to increased plasma C-terminal COL6A3/endotrophin levels. This aligns with previous studies showing that subcutaneous tissue, which accounts for about 90% of body fat, is enriched with C-terminal COL6A3, which is involved in adipose tissue fibrosis and insulin resistance and metabolic dysfunction. We consider these integrative analyses to be mediation analyses given that they pinpoint the causal mediator.

However, we acknowledge that observational and MR mediation analysis relies on additional assumptions⁴⁴. While MR mediation analysis can address some of these assumptions, the field of *cis*-MR mediation analysis is still evolving, and estimating the proportion of mediated effects is challenging, as discussed in the new **Supplementary Note 4**, which is also attached at the end of this letter. Therefore, we removed the MR mediation analysis from the filtering step, as suggested by your comment below. We still provided the results in **Supplementary Note 4** for reference and for the sake of transparency.

As an additional filtering step, given that BMI increases the risk of cardiometabolic diseases ($\beta_{\text{BMI-to-cardiometabolic}} > 0$; OR for CAD = 1.47, 95% CI: 1.38–1.57, $P = 4.2 \times 10^{-31}$; OR for ischemic stroke = 1.21, 95% CI: 1.14–1.28, $P = 7.6 \times 10^{-11}$; OR for cardioembolic stroke = 1.19, 95% CI: 1.05–1.36, $P = 6.2 \times 10^{-3}$; OR for type 2 diabetes = 2.58, 95% CI: 2.20–3.03, $P = 1.8 \times 10^{-31}$), we restricted the analysis to proteins that meet $\beta_{\text{BMI-to-protein}} \times \beta_{\text{protein-to-cardiometabolic}} > 0$. Among the 9 protein-disease associations, 6 met this condition (Fig. 3b; Supplementary Table 9).

Manuscript page 26 (limitation)

This study has important limitations.

...

Fourth, we refrained from emphasizing the proportion by which C-terminal COL6A3 mediates the causal effect of BMI on CAD risk because mediation analysis relies on additional assumptions⁴⁴, and obtaining precise estimates for the proportion mediated with *cis*-MR mediation analysis is challenging (see **Supplementary Note**).

Comments from Reviewer 2
24) Also, do the authors believe that the sentinel cis pQTL or any linked marker in the credible set can have a functional role?

Thank you for the valuable comment. This prompted us to further investigate the epigenetic evidence regarding the functional role of the lead-*cis*-pQTL (rs11677932) of C-terminal COL6A3 from deCODE, which we used in the primary analysis. We queried ATAC-seq data as well as H3K4me3 and H3K27ac ChIP-seq data from regulomeDB and found that the lead *cis*-pQTL of COL6A3 is located in an open chromatin region and an active enhancer domain in multiple tissues, including adipose tissues and arteries. Additionally, rs11677932 had RegulomeDB score of 1b (eQTL + transcription factor (TF) binding + TF motif + DNase Footprint + DNase peak), which indicates that the variant is likely to be a regulatory variant. We have modified the manuscript accordingly.

Manuscript pages 18–20

Regulatory role of the lead *cis*-pQTL of C-terminal COL6A3

We also evaluated whether rs11677932 is a regulatory variant using ENCODE⁷ and RegulomeDB⁸. RegulomeDB assign heuristic ranking score, representing its potential to

be functional in regulatory elements⁹. rs11677932 had a RegulomeDB score of 1b—which is considered to be a strong evidence for a regulatory variant—based on data from eQTL, transcription factor (TF) binding, TF motif, DNase Footprint, and DNase peak. The variant is located in an open chromatin region and an active enhancer domain in multiple tissues, including the adipose tissue and arteries (**Fig. 6**).

Fig. 6. Epigenetic profile of the lead *cis*-pQTL for C-terminal COL6A3.

(a) LocusZoom plots of the pQTL for C-terminal COL6A3 in the 1Mb-region surrounding the lead *cis*-pQTL from the deCODE study, rs11677932.

(b) ATAC-seq, H3K4me3, and H3K27ac ChIP-seq data for adipose tissue, coronary artery, aorta, thoracic artery, and tibial artery. Data are publicly available at ENCODE and RegulomeDB.

(c) rs11677932 is predicted to affect the binding of a transcription factor MEF2B. ENCODE Accession ID: ENCSR782UOT; Target: BORCS8-MEF2B, MEF2B.

Comments from Reviewer 2

25) Have the authors performed any single-track assay to confirm the Somascan or Olink measures?
--

Our study focused on data using SomaScan and Olink assays and did not perform analyses using data from other modalities. However, we note that there is a previous observational study that used monoclonal antibody targeting C-terminal COL6A3, which showed the positive association between plasma C-terminal COL6A3 levels and mortality and cardiovascular events in patients with atherosclerosis⁴⁷. Another study using immunohistochemistry revealed that COL6A3 is present in adipose tissue extracellular matrix. COL6A3 mRNA is correlated with body mass index ($r = 0.60$, $P < 0.0001$) and fat mass ($r = 0.41$, $P < 0.0001$)². We consider that our study builds upon these previous observational studies and further provides insights into the causal relationship.

Comments from Reviewer 2

26) The authors should in depth assess if any RNA expression (eQTL as well as splice QTL) can be linked or can explain the cis pQTL of COL6A3 in plasma.

Thank you for the valuable suggestion. In the revised manuscript, we clarified that the lead-*cis*-pQTL (rs11677932) of C-terminal COL6A3 from deCODE is identified as COL6A3's *cis*-eQTL in aorta in GTEx v8 with directionally concordant effects (i.e., rs11677932-A is associated with decreased COL6A3 expression and decreased plasma C-terminal COL6A3 abundance).

Comments from Reviewer 2

27) Overall, I believe that the BMI angle is very unclear and not specific at all. I would recommend shortening the story and make it to the point of COL6A3 and CAD.

28) Please remove the first part and all mention to BMI as a prior as well as mediation analysis.

Please see our response to your comment #22.

Comments from Reviewer 2

29) Since the pQTL and CAD use published data, it would have been interesting if the authors could determine in vitro how the variants affect the protein and its level (transcription, post transcription)

To provide insights into the transcription and regulatory roles, we have added findings about eQTL (i.e., the lead *cis*-pQTL is *cis*-eQTL with directionally consistent effects) and epigenetics (i.e., the variant is located in the open chromatin region and an active enhancer domain in multiple tissues, including adipose tissues and arteries; the variant has regulomeDB score of 1b, which strongly indicates that the variant has a regulatory role).

Moreover, we have collaborated with Melina Claussnitzer and her team at the Broad Institute to further evaluated the effect of rs11677932, the lead *cis*-pQTL for C-terminal COL6A3 in subcutaneous adipose tissue (SAT) in humans. We found that rs11677932-G, which is associated with increased C-terminal COL6A3 level, is associated with morphological change of subcutaneous adipose tissue (SAT) and increased COL6A3 expression in SAT (**Supplementary Note**).

However, we acknowledge that we did not explore the molecular mechanism whereby these proteins mediated the effect, which was clarified in the manuscript.

Manuscript page 18

Furthermore, in subcutaneous adipose derived mesenchymal cells (AMSC) from 44 bariatric surgery patients, rs11677932 was significantly associated with a morphological change in adipocytes (**Supplementary Note**).

Manuscript page 26

This study has important limitations.

....

Third, we did not explore the molecular mechanism whereby these proteins mediated the effect.

Supplementary Note

The Effect of rs11677932 on Subcutaneous Adipose-Derived Mesenchymal Cells (AMSC)

Methods

We investigated the effects of the variant rs11677932 on gene expression and cellular programs in primary donor-derived differentiated subcutaneous adipose-derived mesenchymal cells (AMSC) (CellGenBank cohort)³⁸. We applied both cardiometabolic disease oriented high content image-based profiling using LipocyteProfiler⁹³ and transcriptomics profiling which allow linking a variant of interest to gene expression changes and to annotate morphological and cellular functional effects associated with the variant³⁸. In this study, we queried donor-derived subcutaneous AMSC differentiated adipocytes and compared samples with rs11677932 homozygous reference and alternate alleles (number of samples GG=44, AA=4). We investigated the differential expression of *COL6A3* and LipocyteProfiler morphological and cellular features, where we used the non-parametric Wilcoxon comparison of means to calculate the significance of the differences associated with the reference versus alternate genotypes. The Pearson correlation was used to evaluate the correlation between the LipocyteProfiler feature and *COL6A3* expressions. In these processes, *COL6A3* expression was normalized and adjusted for patients' age, sex, BMI, and sampling batch as described previously³⁸. $P < 0.05$ was considered to be nominally significant.

Results

Although the number of samples with AA genotype was low, we observed an increase, albeit non-significant, in the expression of *COL6A3* in homozygous rs11677932-G (**Supplementary Fig. 2a**). This was consistent with the direction of effect observed with *COL6A3* eQTL in aorta from GTEx v8⁴⁸, which showed rs11677932-G increases *COL6A3* expression in aorta ($\beta = 0.18$, $P = 1.6 \times 10^{-6}$).

Additionally, we investigated the effect of rs11677932-G on changes in cardiometabolic disease oriented high content imaging based cellular traits using LipocyteProfiler and found that the G allele was associated with a positive effect on the imaging feature "Nuclei AreaShape Zernike 8 2", which measures the changes in area and shape of

subcutaneous adipocytes nuclei ($P = 6.1 \times 10^{-3}$) (**Supplementary Fig. 2b**). Moreover, we observed a positive trend in the relationship between the imaging feature and increased *COL6A3* expression ($r = 0.2$; $P = 0.05$) (**Supplementary Fig. 2c**).

Supplementary Fig. 2. rs1677932 and LipocyteProfiler.

- (a) the effect of rs1677932 on *COL6A3* expression in the subcutaneous adipose tissue
- (b) the effect of rs1677932 on the LipocyteProfiler feature ““Nuclei AreaShape Zernike 8 2”” in the subcutaneous adipose tissue
- (c) association between the LipocyteProfiler feature ““Nuclei AreaShape Zernike 8 2”” and *COL6A3* expression. R denotes Pearson’s correlation coefficient.

Comments from Reviewer 2
30) Since COL6A3 mutations are known in OMIM to cause Bethlem myopathy 1 Dystonia 27 Ullrich congenital muscular dystrophy 1, it would be appreciated to determine the mechanism of action that is described there. Does it involve the involvement of cardiac disease in the phenotypic spectrum? What do the known mutations do? How could they be put in the context of pQTL?

Given that *COL6A3* belongs to essential collagen families, it is understandable that *COL6A3*, not restricted to C-terminal, is associated with multiple diseases including the ones listed. However, we note that N-terminal *COL6A3* (uncleaved portion) was not causal for CAD, whereas C-terminal *COL6A3*, which is cleaved into endotrophin, was strongly causal for CAD. Building upon previous studies indicating that endotrophin is associated with adipose tissue fibrosis, insulin resistance, and is observationally

associated with mortality and cardiovascular outcomes, our study provides causal insights into the C-terminal COL6A3/endotrophin in the context of subcutaneous fat-C-terminal COL6A3-endotrophin-CAD axis.

Response to Reviewer 3

Comments from Reviewer 3

In this paper, the authors searched for proteome mediators of the relationship between BMI and CAD applying Mendelian Randomization approaches. They detected COL6A3 as an interesting candidate and performed follow-up analyses regarding this protein using different sources of evidence such as epidemiologic data and single-cell analysis. The topic and approach are interesting, but I have a few suggestions for improvement of the analyses and interpretations.

We appreciate your excellent comments and constructive feedback. We performed additional analyses as per your comments and modified the manuscript accordingly. Please find our point-by-point responses below.

Comments from Reviewer 3

1. My biggest concern is with respect to the overall error control of the analyses. Since the authors rely on a sequential testing approach (two Mendelian randomization tests plus mediation test for each of the proteins), the overall error rate is not obvious. In this regard the used FDR cut-offs are not convincingly justified. This could be improved for example by performing permutation testing of the study design or running a small simulation study given the covariance structure of data.
--

Thank you for the important feedback. We discussed this with multiple statisticians and decided to transition from FDR to Bonferroni correction. Additionally, we note that each step tests a different hypothesis, building upon the previous step. For example, the first step assesses which proteins are influenced by BMI, and the second step tests which proteins, among those influenced by BMI, influence CAD. Thus, we believe it is justifiable to apply multiple testing correction given the number of tests in each step. However, to further strengthen the claim, we conducted simulations to examine the false positive control of our two-step analytical framework as outlined below. In summary, these simulations demonstrated that our framework can control for false positives, provided that valid instruments for the candidate mediator are used and that no bi-directional relationship exists between the exposure and the candidate mediator. We would like to note that we performed reverse causation filtering using both the MR-Steiger test and bidirectional MR to remove proteins with potential reverse causation. We have modified the manuscript accordingly and have included a detailed explanation of the simulation study of the two-step design of this study in the **Supplementary Note**.

Supplementary Note:

We assessed whether the association between the exposure, e.g. BMI, and the outcome, e.g. coronary artery disease, could induce a spurious association between

the candidate mediator, i.e. circulating protein level, and the outcome in the absence of a true mediation effect.

We first considered two scenarios:

(1) Scenario 1: The exposure has a causal effect on the candidate mediator, while the association between the exposure and the outcome is fully due to the existence of a confounding factor;

(2) Scenario 2: The exposure has a causal effect on the candidate mediator and an independent causal effect on the outcome.

In practice, a bi-directional relationship between the exposure and the candidate mediator is also possible. Therefore, we considered two additional scenarios:

(3) Scenario 3: The candidate mediator has a causal effect on the exposure, while the association between the exposure and the outcome is fully due to the existence of a confounding factor;

(4) Scenario 4: The candidate mediator has a causal effect on the exposure, and the exposure has a causal effect on the outcome.

We introduced a genetic instrument for the candidate mediator, with a minor allele frequency of 0.1. Based on instrumental variable assumptions, this instrument is independent of the confounding factor and does not have any direct effects on the exposure or the outcome. Detailed model specifications are indicated in

Supplementary Figure 1.

We simulated each scenario 1,000 times with a sample size of 100,000. We performed MR to test whether the candidate mediator has a causal effect on the outcome in each replicate of these four scenarios.

As a result, we found that the distributions of p-values over 1,000 replicates in scenarios 1, 2 and 3 were close to a standard uniform distribution, which suggests that an effective control of false positives can be achieved with appropriate multiple testing correction. In contrast, the distribution of p-values in scenario 4 was severely left skewed. This is not surprising since the candidate mediator does have an indirect causal effect on the outcome. However, this association does not establish the mediating role of the candidate mediator, because it is the exposure that is mediating the causal effect of the candidate mediator on the outcome.

In summary, through these simulations, we have demonstrated that our two-step analytical framework is able to control for false positives, provided that valid instruments for the candidate mediator are used and that there does not exist bi-directional relationship between the exposure and the candidate mediator.

For simplicity, in all scenarios, $U, \epsilon_M, \epsilon_E, \epsilon_Y \sim i.i.d. N(0,1)$, $G \sim Binom(2,0.1)$, $\beta_{GM} = 0.3$, $\beta_{UM} = 0.5$, $\beta_{UE} = 0.4$, $\beta_{UY} = 0.4$; In scenarios 1 and 2, $\beta_{EM} = 0.3$; In scenarios 3 and 4, $\beta_{ME} = 0.3$; In scenarios 2 and 4, $\beta_{EY} = 0.3$.

Comments from Reviewer 3
2. BMI is not a convincing marker of obesity and there are contradicting results regarding its causal relationship to CAD. I would propose also considering WHR for which large GWAS are also available.

Thank you for the valuable suggestion. We agree that BMI does not constitute a single perfect proxy although it is clinically relevant and the most powered proxy of obesity. Thus, we modified step 1 MR to add additional filtering such that proteins must show

directionally concordant estimates across BMI and body fat percentage. Additionally, we evaluated the causal effects of specific body fat compartments on C-terminal COL6A3 levels using GWAS of MRI-derived fat compartment volumes¹⁹ and found that subcutaneous fat is the key compartment that increase plasma COL6A3 levels ($\beta = 0.52$, 95% CI: 0.35–0.69, $P = 9.32 \times 10^{-10}$). Further details can be found below.

Notably, this aligns with previous studies reporting that:

- (i) the collagen matrix surrounding subcutaneous fat is strongly enriched with C-terminal COL6A3, which releases endotrophin,
- (ii) increased COL6A3 expression in subcutaneous adipose tissues is associated with adipose tissue fibrosis, insulin resistance, and metabolic dysfunction.

In combination with these previous findings, our study strongly suggests that obesity results in increased subcutaneous fat, which constitutes around 90% of the total body fat, leading to increased C-terminal COL6A3 and release of endotrophin, thereby contributing to adipose fibrosis, insulin resistance, and an increased risk of coronary artery disease. We modified the manuscript accordingly.

Figure 5. Follow-up analyses for collagen type VI α 3 (COL6A3).

MR for the effect of body fat compartments on COL6A3 stratified by C- and N-terminal COL6A3.

We modified the manuscript accordingly as below.

Manuscript page 29

For assessing directional concordance with body fat percentage, we performed MR to evaluate the effect of body fat percentage on the same 4,907 plasma proteins (**Methods**). We did this because body fat percentage is considered to be a more direct proxy of

obesity, whereas BMI is an easy-to-measure, clinically relevant proxy.¹² However, the sample size available to assess the genetic determinants of BMI is larger than that of body fat percentage, providing more precise estimates. We found that body fat percentage influenced 94.7% of all BMI-driven proteins with the same direction of effect as BMI (**Fig. 2d**), illustrating a high concordance of results between the two different measures of obesity ($r = 0.93$; $P < 2.2 \times 10^{-16}$). **Only proteins influenced by both BMI and body fat percentage in the same direction were retained for downstream analyses.**

Manuscript pages 15–17

Body fat compartments influencing plasma levels of C- and N-terminal COL6A3 levels

We conducted additional MR analyses to evaluate which body fat components specifically influence plasma C- and N-terminal COL6A3 levels, using GWAS of MRI-derived fat compartment volumes for two-sample MR. Among the three fat compartments analyzed—abdominal subcutaneous adipose tissue (ASAT), visceral adipose tissue (VAT), and gluteofemoral adipose tissue (GFAT)—ASAT was found to significantly increase plasma levels of both C- and N-terminal COL6A3, with a more pronounced increase in C-terminal COL6A3 (**Supplementary Table 16**). Notably, this finding aligns with previous studies that (i) the collagen matrix surrounding subcutaneous fat is rich in C-terminal COL6A3¹, which in turn releases endotrophin and (ii) increased expression of COL6A3 in subcutaneous adipose tissues is associated with adipose tissue fibrosis, insulin resistance, and metabolic dysfunction²⁻⁶

Figure 5. Follow-up analyses for collagen type VI α 3 (COL6A3).

(a) Schematic illustration of proposed relationship between obesity, COL6A3 (Collagen type VI α 3 chain), endotrophin, and coronary artery disease. Obesity leads to increased

production of COL6A3, whose C-terminal is cleaved into an active form termed endotrophin, which increases the risk of coronary artery disease.

(b) Schematic diagram of COL6A3 (UniProt ID: P12111). COLA3 consists of a short collagenous region flanked by multiple von Willebrand factor type A (vWF-A) modules (N1–N10 in the N-terminal and C1,2 in the C-terminal). There are three additional C-terminal domains unique to COL6A3 (C3–C5), which are not present in other collagen type VI families. The most C-terminal domain (C5) is cleaved into a soluble protein termed endotrophin.

The two amino acid sequences targeted by the aptamers are as follows: the N-terminal-binding aptamer targets the amino acid sequence 26–1036 (uncleaved section), while the C-terminal aptamer targets the amino acid sequence 3108–3165 (cleaved section). The figure has been modified from ref^{17,18}.

Comments from Reviewer 3

3. I could not find the estimated total causal effect of BMI on CAD. Please add. Likewise, the proportions of causal effect mediated by the proteins should be provided as well as an estimate of the direct BMI effect.
--

Thank you for your comment. We have added the estimated causal effect of BMI on CAD (OR = 1.47, 95% CI: 1.38–1.57, $P = 4.16 \times 10^{-31}$) in the main text and

Supplementary Table 9. Regarding the proportion mediated, we refrain from emphasizing this in this paper, **given the challenges associated with using cis-MR mediation analysis for this estimation**⁴⁴. Please refer to **Supplementary Note 4** for a full discussion, which is attached at the end of this letter. Following Reviewer 2's comment, we have also moved the MR mediation analysis (network MR) to the **Supplementary Note 4**. However, we note that the MR mediation analysis is just one line of evidence, and we have provided multiple lines of evidence supporting COL6A3 as a mediator for the effect of BMI on the CAD risk, including the new body fat compartments MR, which pinpoint the causal effect of subcutaneous fat on increased plasma levels of C-terminal COL6A3.

Manuscript page 11

As an additional filtering step, given that BMI increases the risk of cardiometabolic diseases ($\beta_{\text{BMI-to-cardiometabolic}} > 0$; OR for CAD = 1.47, 95% CI: 1.38–1.57, $P = 4.2 \times 10^{-31}$;

OR for ischemic stroke = 1.21, 95% CI: 1.14–1.28, $P = 7.6 \times 10^{-11}$; OR for cardioembolic stroke = 1.19, 95% CI: 1.05–1.36, $P = 6.2 \times 10^{-3}$; OR for type 2 diabetes = 2.58, 95% CI: 2.20–3.03, $P = 1.8 \times 10^{-31}$), we restricted the analysis to proteins that meet $\beta_{\text{BMI-to-protein}} \times \beta_{\text{protein-to-cardiometabolic}} > 0$. Among the 9 protein-disease associations, 6 met this condition (Fig. 3b; Supplementary Table 9).

Manuscript page 26

This study has important limitations.

....

Fourth, we refrained from emphasizing the proportion by which C-terminal COL6A3 mediates the causal effect of BMI on CAD risk, given the challenges associated with using cis-MR mediation analysis for this estimation⁴⁴ (see Supplementary Note).

Comments from Reviewer 3

4. Since multiple proteins are considered as potential mediators, a multi-trait MR analysis could be appropriate to estimate the total amount of mediated causal BMI effect.
--

Multi-trait MR mediation analyses could be of potential interest and may better quantify the proportion mediated by estimating the independent contribution of the proteins. However, multivariable MR works best when the exposure GWAS has decent number of overlapping instrumental variables (IV), as in the case with body fat mass and fat-free mass. However, *cis*-pQTL of proteins typically do not overlap at all. Therefore, including multiple proteins as exposures in multivariable MR result in great reduction in conditional F-statistics of each protein (i.e., essentially, F-statistics of each protein will be “diluted” because of non-contributing IV derived from other proteins). Although including *trans*-pQTL may results in more overlap of IV, this will greatly increase the risk of horizontal pleiotropy due to the nature of *trans*-pQTL. Thus, we believe multi-trait MR with *cis*-pQTL is an area that requires further method development, which is out of the scope of this paper.

Comments from Reviewer 3

5. It remains unclear how the MR assumptions regarding BMI->Proteome are checked. How did you exclude reverse causality?
--

When testing the causal relationship between BMI and a specific gene, I would strongly suggest removing BMI instruments located at the same chromosome as the protein coding gene.
--

In revised step 1 MR, sensitivity analyses included the following tests:

- heterogeneity ($I^2 < 50\%$), directional horizontal pleiotropy (MR-Egger intercept test: $P > 0.05$)
- outlier-robust estimate (MR-PRESSO outlier-corrected estimate, weighted median estimate, and MR-Egger slope estimate must show directionally consistent effect as the IVW)
- reverse causation (MR-Steiger test indicating “correct_causal_direction” == TRUE as well as reverse MR to estimate the causal effect of protein levels on BMI showing no evidence of causality)
- directional concordance with body fat percentage).

We have included a new **Extended Fig. 1** showing the flowchart for these filtering steps.

We also would like to note that BMI has 304 Instrumental Variables (IVs) across the genome, and the causal effect of BMI on plasma protein levels were driven by a

combination of predominantly *trans*-signals, as illustrated in the representative MR scatter plot for the effect of BMI and COL6A3 below. Specifically regarding COL6A3, we also highlight that there was no IV for BMI in COL6A3's *cis*-region, further supporting that there was no evidence of reverse causation.

Comments from Reviewer 3

6. Regarding the causal relationships Protein->CAD, the authors should comment on the differences in statistical power for these analyses, which strongly depend on the strength of the *cis*-effects found for the proteins. Maybe a table comparing detectable causal effects of the MR-analyses for the different proteins could be helpful.

Thank you for the suggestion. In the **Supplementary Table 2**, we clarified that all proteins had F-statistics of > 10 (Median = 272.4, min = 27.5, and max = 1440.2). Additionally, we also modified the manuscript.

Manuscript page 8

No evidence of weak instrumental variables (suspected when F-statistics < 10) were found (**Supplementary Table 2**).

Comments from Reviewer 3

7. I am a bit disappointed that the authors did not consider sex as a major parameter since both obesity and CAD show strong sex-dimorphisms and there are examples of sex-specific causal effects for both traits. I would suggest adding this analysis because there are sex-stratified GWAS results available for obesity-related traits. This might not be available for CAD, but still, sex-specific causal effects could be approximately estimated by considering the causal relationship Protein (sex-specific) -> CAD (overall). This approach was proposed / applied by others.

We appreciate the excellent suggestion. We performed additional analysis to evaluate the sex-specific effects in step 1 and step 2 MR. We generated sex-stratified COL6A3 pQTL with data from UK Biobank and used it along with sex-stratified GWAS of BMI and CAD. Given that sex-stratified CAD GWAS's sample size was much smaller than that of sex-combined CAD, we also used the sex-combined CAD GWAS. Using these GWAS, we performed Step 1 MR and Step 2 MR in a sex-stratified manner.

Additionally, we performed sex-stratified Cox-regression analysis to evaluate the relationship between baseline plasma COL6A3 level and the cumulative incidence of CAD.

For Step 1 MR, we replicated the causal effect of BMI on C-terminal COL6A3 level in both sexes. For Step 2 MR, we replicated the causal effect of C-terminal COL6A3 level on CAD risk in males, and found a suggestive, directionally consistent effect in females. Additionally, we conducted sex-stratified Cox regression analysis for the 10-year cumulative incidence of CAD in the UK Biobank. We found that an increase in plasma C-terminal COL6A3 level was associated with an increased cumulative incidence of CAD in both females and males. Collectively, these findings indicate the shared effect of C-terminal COL6A3 on CAD risk in both sexes.

Manuscript pages 20–21

Sex-stratified analyses of C-terminal COL6A3

To evaluate whether there is a difference in associations by sex, we performed Step 1 MR (BMI to C-terminal COL6A3) and Step 2 MR (C-terminal COL6A3 to CAD risk) in a sex-stratified manner.

Sex-stratified Step 1 MR:

First, we generated sex-stratified pQTL for C-terminal COL6A3 using data from 19,747 females and 16,876 males from the UK Biobank (**Methods; Extended Fig. 3a**). For BMI, we obtained publicly available sex-stratified GWAS from the GIANT consortium (73,137 females and 60,586 males)⁴⁹. Using these GWAS, we conducted two-sample MR to evaluate the effect of BMI on C-terminal COL6A3 in a sex-stratified manner. We found that an s.d. increase in BMI is associated with increased C-terminal COL6A3 levels in females ($\beta = 0.16$, 95% CI: 0.11–0.20, $P = 1.1 \times 10^{-12}$) and males ($\beta = 0.10$, 95% CI: 0.05–0.14, $P = 1.4 \times 10^{-5}$) (**Extended Fig. 3b; Supplementary Table 17**), replicating the result in sex-combined analysis.

Sex-stratified Step 2 MR:

We identified *cis*-pQTL for C-terminal COL6A3 in females and males from the pQTL we generated in the above step. For CAD, we obtained publicly available sex-stratified GWAS for CAD (female cases: 22,997, female controls: 310,499, male cases: 54,083, male controls: 240,453). We performed two-sample *cis*-MR to evaluate the effect of C-terminal COL6A3 on CAD risk. We found that an s.d. increase in plasma C-terminal COL6A3 levels is associated with increased CAD risk in males (OR = 3.59, 95% CI: 2.14–6.01, $P = 1.3 \times 10^{-6}$); females showed positive trend (OR = 1.51, 95% CI: 0.89–2.60, $P = 0.13$), but did not reach statistical significance with wide 95% CI (**Extended Fig. 3c; Supplementary Table 17**). Colocalization showed that there is a shared causal signal between female-specific and male-specific pQTL ($PP_{\text{shared}} > 0.99$), with rs105785 being the lead pQTL (**Extended Fig. 3d**), which was also the lead pQTL in the sex-combined pQTL (**Extended Fig. 2**).

Sex-stratified Cox regression analysis for cumulative incidents of CAD:

To further evaluate the relationship of C-terminal COL6A3 and CAD risk in a sex-aware manner, we performed sex-stratified Cox regression analysis for cumulative incidents of coronary artery disease in the UK Biobank (see **Methods**). After adjustment for age, sex, recruitment center, and protein sample processing time, we found that an s.d. increase in plasma C-terminal COL6A3 level was associated with increased cumulative incidence of CAD in both females (HR = 1.15, 95% CI: 1.07–1.23, $P = 6.7 \times 10^{-5}$) and males (HR = 1.24, 95% CI: 1.18–1.30, $P = 2.2 \times 10^{-16}$) with overlapping 95% CI between the sexes.

Extended Fig. 3. Sex-stratified analyses of C-terminal COL6A3.

- (a) Sex-stratified GWAS of plasma levels of C-terminal COL6A3 (sex-stratified pQTL).
 (b) Scatter plots showing the sex-stratified two-sample MR results for the effect of BMI

on plasma levels of COL6A3 using the inverse-variance weighted method, weighted median, or MR-Egger slope methods. (c) Sex-stratified Step 1 MR results for the effect of BMI on C-terminal COL6A3 levels (left panel) and Step 2 MR results for the effect of C-terminal COL6A3 on CAD risk (right panel). (d) Sex-stratified LocusZoom plots for *cis*-pQTL of C-terminal COL6A3 from the UK Biobank (left panel) and coronary artery disease GWAS from Aragam et al³¹ (right panel).

Comments from Reviewer 3

8. Page 10: The authors mention a replication MR of the COL6A3 -> CAD relationship in other studies. It remains unclear how this was performed. Did you perform single study MR analyses? Or did you only use study-specific pQTLs as instruments? Please clarify. Please present the meta-effect across studies.

We apologize for the lack of clarity in the previous manuscript. We clarified in the revised manuscript that we performed separate two-sample MR analyses using study-specific pQTLs as instruments, while using the same CAD GWAS. We refrained from doing meta-analysis of MR results given that the same CAD GWAS was used in all of these MR analyses, making these tests not completely independent of each other, which invalidate the meta-analysis.

Manuscript page 30

Replication MR using *cis*-pQTL from different cohorts

We evaluated whether the causal relationship between COL6A3 and CAD could be replicated using different sources of *cis*-pQTLs from other cohorts. For this, we conducted two-sample MR using *cis*-pQTLs from three additional cohorts: UK Biobank⁵⁰ ($n = 34,557$ individuals), Fenland¹³ ($n = 10,708$ individuals), and ARIC³⁶ ($n = 7,213$ individuals). **We used the same CAD GWAS³¹ used in the primary analyses as an outcome.**

Comments from Reviewer 3

9. Searching the supplemental tables is cumbersome. Please consider adding a main table with the primary results.

Thank you for your feedback. We have incorporated a table that summarizes the results related to the C- and N-terminal COL6A3, presented in the style of forest plots, as shown in **Fig 5c–e** (Please see our response above).

Comments from Reviewer 3

10. The possible mediating role of endotrophin could be discussed in more detail. There are for example findings of this hormone in relation to heart failure. Moreover, since sex hormones were described as causally related to obesity and CAD, it could be interesting to analyse possible relationships of endotrophin with sex hormones.
--

We appreciate your suggestion. We have performed further analyses to disentangle effects of the specific body fat compartments on plasma C- and N-terminal COL6A3 levels (please refer to our response “**Body fat compartments influencing plasma levels of C- and N-terminal COL6A3 levels**”). We have put these findings in the context of previous studies and suggested a coherent narrative: obesity results in increased subcutaneous fat, which constitutes around 90% of the total body fat, leading to increased C-terminal COL6A3 and release of endotrophin, thereby contributing to adipose fibrosis, insulin resistance, and an increased risk of coronary artery disease. We modified the manuscript accordingly.

Additionally, we conducted sex-stratified analyses for both Step 1 (BMI to C-terminal COL6A3-endotrophin) and Step 2 (C-terminal COL6A3-endotrophin to CAD) and did not observe evidence of heterogeneity between the sexes. Please refer to our response “**Sex-stratified analyses of C-terminal COL6A3.**”

However, it's important to note that the female-specific analysis in Step 2 MR is constrained by the smaller sample size of females in the CAD GWAS. This limitation has been duly noted in the Discussion section of our manuscript.

Comments from Reviewer 3

Minor:

1. Page 3: It is a bit odd to cite COVID papers here because these are not related to the topic of this research. There are numerous other papers considering causal effects on obesity / CAD and possible interrelationships which would be more appropriate to cite here to my opinion.

Thank you for the feedback. In response, we have updated the manuscript by focusing on examples pertaining to CAD.

Manuscript page 4

Such methods have been successfully leveraged to prioritize therapeutic targets, including IL6R⁵¹ and ANGPTL3 for CAD⁵².

- **Other changes to the manuscript include:**

Addition of new authors who contributed to the revision, along with updates for the current authors' affiliations, which are indicated in red in the manuscript.

- Satoshi Koyama and Tetsushi Nakao contributed to the section “Regulatory role of the lead cis-pQTL of C-terminal COL6A3”.
 - Rama Kaalia, Hesam Dashti, and Melina Claussnitzer contributed to “Supplementary Note 3: The Effect of rs11677932 on Subcutaneous Adipose-Derived Mesenchymal Cells (AMSC).”
 - Jason Flannick contributed to overall discussion and critical review of the manuscript.
- Correction of typos and improvements in clarity, which are indicated in red in the manuscript.

Supplementary Note

Supplementary Note 4:

Challenges regarding the estimation of the proportion mediated in MR mediation analysis with cis-pQTL

Traditional non-instrumental variable (IV) mediation analysis relies on strong assumptions. Some of which are untestable, such as (i) no unmeasured confounding between the exposure, mediator, and outcome; (ii) no exposure-caused confounders of the mediator-outcome relationship; and (iii) no exposure-mediator interaction. MR can be used to overcome some of these challenges^{5,6,7}. Importantly, MR estimates are less likely to be biased due to unmeasured confounding among the exposure, mediator, or outcome. However, it still relies on multiple assumptions: it assumes no interaction between the exposure and mediator and that instrumental variables influence the outcome solely through the exposure and mediator (exclusion restriction), in addition to all the other MR assumptions.

Currently, there are two methods to estimate the proportion mediated in MR mediation analysis: (i) the multivariable MR (MVMR) approach and (ii) the product of coefficients method (network MR approach)⁵. In the MVMR approach, the direct effect of the exposure on the outcome, controlling for the mediator, is estimated by including genetic instruments for both the primary exposure and the mediator. For the product of coefficients method (network MR approach), MVMR is still required when estimating the causal effect of the mediator (C-terminal COL6A3) on the outcome (CAD), adjusting for the primary exposure (BMI) (**Supplementary Fig. 3a**).

However, using *cis*-pQTL of C-terminal COL6A3 in MVMR, with protein level and BMI as exposures, poses unique challenges due to the distinct genetic architectures: the highly polygenic nature of BMI with widespread associations across the genome and much simpler genetic architecture of pQTL with strong associations in the *cis*-region for the protein level.

There are only up to two instrumental variables for C-terminal COL6A3 depending on the cohorts (two from the deCODE cohort; one from other cohorts), while there are 304 instrumental variables for BMI across the genome, reflecting its polygenic nature. Specifically, there is no instrumental variable for BMI in the *cis*-region of the COL6A3 gene. This means that including instrumental variables for BMI and *cis*-pQTL in the same MVMR is likely to introduce weak instrumental bias and horizontal pleiotropy. For

example, adding the instrumental variable of BMI to cis-pQTL of C-terminal COL6A3 will “dilute” the strong association between the cis-pQTL and its protein levels (see Supplementary Figure 3b).

Currently, there is no effective method to overcome the above-mentioned challenges. Therefore, further method development or refinement is required to conduct multivariable MR using cis-pQTL (or any cis molecular trait) and instrumental variables of polygenic traits such as BMI.

Supplementary Fig. 3. Schematic illustration of a product of coefficients method for MR mediation analysis and multivariable MR.

(a) The figure demonstrates the causal relationship between BMI, the protein mediator, and cardiometabolic diseases using directed acyclic graphs. The dark blue arrow represents the total effect of BMI on cardiometabolic diseases ($\beta_{\text{BMI-to-CAD}}$), while the red

arrow represents the effect of BMI on cardiometabolic diseases mediated by the protein mediator ($\beta_{\text{mediated}} = \beta_{\text{BMI-to-protein}} \times \beta_{\text{protein-to-CAD}}$).

(b) The genetic instrumental variables (IVs) for BMI will “dilute” the association between the IV and C-terminal COL6A3 level in MVMR. The red arrow represents the association between IV for BMI and C-terminal COL6A3 level, which may introduce the weak instrumental bias and horizontal pleiotropy.

Network MR without MVMR (without adjustment for BMI)

One way to bypass the issue mentioned above relating to MVMR is to use the product-of-coefficients method without adjusting for BMI when estimating the effect of C-terminal COL6A3 (the mediator) on CAD risk (the outcome) ($\beta_{\text{protein-to-CAD}}$ in **Supplementary Figure 3**). This method has been used in multiple studies⁸⁻¹⁰.

However, it is important to note that this approach still relies on all the aforementioned assumptions and may inflate the estimated proportion mediated due to a larger $\beta_{\text{protein-to-CAD}}$. Nevertheless, as a supplementary analysis, we applied this method to estimate the proportion of the total effect of BMI on CAD risk mediated by the C-terminal COL6A3 level.

Methods:

To estimate the causal mediation effects (β_{mediated}), we estimated the effect of BMI on the plasma protein levels ($\beta_{\text{BMI-to-protein}}$) and the effect of the plasma proteins on cardiometabolic diseases ($\beta_{\text{protein-to-cardiometabolic}}$), and then multiplied these values ($\beta_{\text{mediated}} = \beta_{\text{BMI-to-protein}} \times \beta_{\text{protein-to-CAD}}$). For this, we performed MR using the same instrumental variables as in Steps 1 and 2 of MR. Subsequently, we divided β_{mediated} by β_{total} to estimate the proportion mediated and calculated the *P*-value under the null hypothesis that the protein of interest did not mediate the effect of BMI on the outcome of interest. We considered results with *P* < 0.05 to be significant (denoted with an asterisk *).

Results:

Below are the estimated effect of BMI on CAD ($\beta_{\text{BMI-to-CAD}}$), BMI to C-COL6A3 ($\beta_{\text{BMI-to-protein}}$), and C-COL6A3 to CAD ($\beta_{\text{protein-to-CAD}}$). Estimated proportion mediated was 32.2% (95% CI: 17.2–47.1), *P* = 2.54×10^{-5} .

MR (exposure-to-outcome)	method	SNPs	Estimate	95% CI Lower	95% CI Upper	P
BMI to the CAD ($\beta_{\text{BMI-to-CAD}}$)	IVW	294	0.38 (0.32–0.45)	0.32	0.45	4.16E-31*

BMI to C-COL6A3 ($\beta_{\text{BMI-to-protein}}$)	IVW	285	0.32 (0.26–0.38)	0.26	0.38	3.65E-24*
C-COL6A3 to CAD ($\beta_{\text{protein-to-CAD}}$)	Wald ratio	1	0.38 (0.23–0.53)	0.23	0.53	4.46E-07*
Proportion mediated	–	–	32.2%	17.2%	47.1%	2.54×10^{-5} .

Observational mediation analysis in the EPIC-Norfolk cohort

As additional supplementary analysis, we performed observational mediation analysis using individual-level data from the EPIC-Norfolk cohort (see **Methods** for further details of the cohort).

Methods:

We used the product of coefficients methods to calculate the proportion mediated, as described above, using the R package mediation v4.5.0. We used linear regression analysis to estimate the effect of BMI on plasma C-terminal COL6A3 levels. We used logistic regression to estimate the effect of BMI on the CAD risk and the effect of C-terminal COL6A3 levels on the CAD risk. For covariates, we adjusted for age and sex given the availability of data. Significance of the indirect effect and the proportion mediated was estimated by computing unstandardized effects in 1000 bootstrapped samples and calculating the corresponding 95% confidence intervals.

Results: Proportion mediated was estimated to be 20.54% (95%CI: 3.48–81%), $P = 0.018$.

Mediation analyses in the EPIC-Norfolk cohort				
Effect	Estimate	95% CI Lower	95% CI Upper	p-value
BMI-to-COL6A3 (β , linear regression)	0.060	0.043	0.08	8.54×10^{-12} *
COL6A3-to-CAD (odds ratio, logistic regression)	1.34	1.12	1.59	1.12×10^{-3} *
BMI-to-CAD (odds ratio, logistic regression)	0.0035	0.0018	0	0.006 *
Proportion mediated	20.54%	3.48%	81%	0.018 *

Observational mediation analysis in the UK Biobank

We repeated the observational mediation analysis in 35,100 individuals from the UK Biobank. We included participants from the UK Biobank for whom we have protein measurements with the Olink Explore 3072 assay (UK Biobank data field: 30900) and ICD10-based diagnosis (data field: 41270). Further details of the cohort can be found in the **Methods** (see **Observational regression analysis for baseline BMI and plasma**

C-terminal COL6A3 level in the UK Biobank). We adjusted for age and sex and used the R package mediation v4.5.0, as described above.

Estimated proportion mediated was 35.34% (95% CI: 28.6%–44%, $P < 2.2 \times 10^{-16}$)

Effect	Estimate	95% CI Lower	95% CI Upper	p-value
BMI-to-COL6A3 (β , linear regression)	0.060	0.043	0.08	8.54×10^{-12} *
COL6A3-to-CAD (odds ratio, logistic regression)	1.34	1.12	1.59	1.12×10^{-3} *
BMI-to-CAD (odds ratio, logistic regression)	0.0035	0.0018	0	0.006 *
Proportion mediated	35.34%	28.6%	44%	$< 2.2 \times 10^{-16}$

We note that these estimates of three methods/datasets align with each other: The estimated proportion mediated from the MR mediation analysis = 32.2% (95% CI: 17.2–47.1), $P = 2.54 \times 10^{-5}$; that from the observational analysis in the EPIC-Norfolk = 20.54% (95% CI: 3.48–81%), $P = 0.018$; that from the observational analysis in the UK Biobank = 35.34% (95%CI: 28.6–44%), $P < 2.2 \times 10^{-16}$.

However, as mentioned previously, traditional observational mediation analysis relies on strong assumptions. While MR mediation analysis is less susceptible to bias, it still relies on multiple additional assumptions that are difficult to verify. Importantly, it was not possible to adjust for BMI when estimating the effect of C-COL6A3 on CAD ($\beta_{\text{protein-to-CAD}}$) in order to avoid the weak instrumental bias and horizontal pleiotropy in MVMR. To overcome these limitations, further method development or refinement is required.

References

- 1 Heumuller, S. E. *et al.* C-terminal proteolysis of the collagen VI alpha3 chain by BMP-1 and proprotein convertase(s) releases endotrophin in fragments of different sizes. *J. Biol. Chem.* **294**, 13769-13780 (2019).
<https://doi.org/10.1074/jbc.RA119.008641>
- 2 Pasarica, M. *et al.* Adipose tissue collagen VI in obesity. *J Clin Endocrinol Metab* **94**, 5155-5162 (2009). <https://doi.org/10.1210/jc.2009-0947>
- 3 Jo, W. *et al.* MicroRNA-29 Ameliorates Fibro-Inflammation and Insulin Resistance in HIF1alpha-Deficient Obese Adipose Tissue by Inhibiting Endotrophin Generation. *Diabetes* **71**, 1746-1762 (2022).
<https://doi.org/10.2337/db21-0801>
- 4 Sun, K., Park, J., Kim, M. & Scherer, P. E. Endotrophin, a multifaceted player in metabolic dysregulation and cancer progression, is a predictive biomarker for the response to PPARgamma agonist treatment. *Diabetologia* **60**, 24-29 (2017).
<https://doi.org/10.1007/s00125-016-4130-1>
- 5 Dankel, S. N. *et al.* COL6A3 expression in adipocytes associates with insulin resistance and depends on PPARgamma and adipocyte size. *Obesity (Silver Spring)* **22**, 1807-1813 (2014). <https://doi.org/10.1002/oby.20758>
- 6 Sun, K. *et al.* Endotrophin triggers adipose tissue fibrosis and metabolic dysfunction. *Nat. Commun.* **5**, 3485 (2014).
<https://doi.org/10.1038/ncomms4485>
- 7 Consortium, E. P. *et al.* Expanded encyclopaedias of DNA elements in the human and mouse genomes. *Nature* **583**, 699-710 (2020).
<https://doi.org/10.1038/s41586-020-2493-4>
- 8 Boyle, A. P. *et al.* Annotation of functional variation in personal genomes using RegulomeDB. *Genome Res.* **22**, 1790-1797 (2012).
<https://doi.org/10.1101/gr.137323.112>
- 9 Dong, S. *et al.* Annotating and prioritizing human non-coding variants with RegulomeDB v.2. *Nat Genet* **55**, 724-726 (2023).
<https://doi.org/10.1038/s41588-023-01365-3>
- 10 Burgess, S., Davies, N. M. & Thompson, S. G. Bias due to participant overlap in two-sample Mendelian randomization. *Genet. Epidemiol.* **40**, 597-608 (2016).
<https://doi.org/10.1002/gepi.21998>
- 11 Ferkingstad, E. *et al.* Large-scale integration of the plasma proteome with genetics and disease. *Nat. Genet.* **53**, 1712-1721 (2021).
<https://doi.org/10.1038/s41588-021-00978-w>

- 12 Peltz, G., Aguirre, M. T., Sanderson, M. & Fadden, M. K. The role of fat mass index in determining obesity. *Am. J. Hum. Biol.* **22**, 639-647 (2010).
<https://doi.org/10.1002/ajhb.21056>
- 13 Pietzner, M. *et al.* Mapping the proteo-genomic convergence of human diseases. *Science* **374**, eabj1541 (2021).
<https://doi.org/10.1126/science.abj1541>
- 14 Umans, B. D., Battle, A. & Gilad, Y. Where Are the Disease-Associated eQTLs? *Trends Genet.* **37**, 109-124 (2021). <https://doi.org/10.1016/j.tig.2020.08.009>
- 15 Sun, B. B. *et al.* Genomic atlas of the human plasma proteome. *Nature* **558**, 73-79 (2018). <https://doi.org/10.1038/s41586-018-0175-2>
- 16 Andersson, R. *et al.* An atlas of active enhancers across human cell types and tissues. *Nature* **507**, 455-461 (2014). <https://doi.org/10.1038/nature12787>
- 17 Wang, J. & Pan, W. The Biological Role of the Collagen Alpha-3 (VI) Chain and Its Cleaved C5 Domain Fragment Endotrophin in Cancer. *Onco Targets Ther.* **13**, 5779-5793 (2020). <https://doi.org/10.2147/OTT.S256654>
- 18 Chen, P., Cescon, M. & Bonaldo, P. Collagen VI in cancer and its biological mechanisms. *Trends Mol. Med.* **19**, 410-417 (2013).
<https://doi.org/10.1016/j.molmed.2013.04.001>
- 19 Agrawal, S. *et al.* BMI-adjusted adipose tissue volumes exhibit depot-specific and divergent associations with cardiometabolic diseases. *Nat Commun* **14**, 266 (2023). <https://doi.org/10.1038/s41467-022-35704-5>
- 20 Maffei, M. *et al.* Leptin levels in human and rodent: Measurement of plasma leptin and ob RNA in obese and weight-reduced subjects. *Nature Medicine* **1**, 1155-1161 (1995). <https://doi.org/10.1038/nm1195-1155>
- 21 Haider, D. G. *et al.* Plasma adipocyte and epidermal fatty acid binding protein is reduced after weight loss in obesity. *Diabetes Obes Metab* **9**, 761-763 (2007).
<https://doi.org/10.1111/j.1463-1326.2007.00717.x>
- 22 Visser, M., Bouter, L. M., McQuillan, G. M., Wener, M. H. & Harris, T. B. Elevated C-Reactive Protein Levels in Overweight and Obese Adults. *JAMA* **282**, 2131-2135 (1999). <https://doi.org/10.1001/jama.282.22.2131>
- 23 Park, H. S., Park, J. Y. & Yu, R. Relationship of obesity and visceral adiposity with serum concentrations of CRP, TNF- α and IL-6. *Diabetes Research and Clinical Practice* **69**, 29-35 (2005).
<https://doi.org/https://doi.org/10.1016/j.diabres.2004.11.007>

- 24 Zaghlool, S. B. *et al.* Revealing the role of the human blood plasma proteome in obesity using genetic drivers. *Nat. Commun.* **12**, 1279 (2021). <https://doi.org/10.1038/s41467-021-21542-4>
- 25 Goudswaard, L. J. *et al.* Effects of adiposity on the human plasma proteome: observational and Mendelian randomisation estimates. *Int. J. Obes. (Lond.)* **45**, 2221-2229 (2021). <https://doi.org/10.1038/s41366-021-00896-1>
- 26 Sun, B. B. *et al.* Plasma proteomic associations with genetics and health in the UK Biobank. *Nature* **622**, 329-338 (2023). <https://doi.org/10.1038/s41586-023-06592-6>
- 27 Watanabe, K. *et al.* Multiomic signatures of body mass index identify heterogeneous health phenotypes and responses to a lifestyle intervention. *Nat Med* **29**, 996-1008 (2023). <https://doi.org/10.1038/s41591-023-02248-0>
- 28 van der Harst, P. & Verweij, N. Identification of 64 Novel Genetic Loci Provides an Expanded View on the Genetic Architecture of Coronary Artery Disease. *Circ. Res.* **122**, 433-443 (2018). <https://doi.org/10.1161/CIRCRESAHA.117.312086>
- 29 Mishra, A. *et al.* Stroke genetics informs drug discovery and risk prediction across ancestries. *Nature* **611**, 115-123 (2022). <https://doi.org/10.1038/s41586-022-05165-3>
- 30 Mahajan, A. *et al.* Multi-ancestry genetic study of type 2 diabetes highlights the power of diverse populations for discovery and translation. *Nat. Genet.* **54**, 560-572 (2022). <https://doi.org/10.1038/s41588-022-01058-3>
- 31 Aragam, K. G. *et al.* Discovery and systematic characterization of risk variants and genes for coronary artery disease in over a million participants. *Nat. Genet.* (2022). <https://doi.org/10.1038/s41588-022-01233-6>
- 32 Koyama, S. *et al.* Population-specific and trans-ancestry genome-wide analyses identify distinct and shared genetic risk loci for coronary artery disease. *Nature Genetics* (2020). <https://doi.org/10.1038/s41588-020-0705-3>
- 33 Zhao, H. *et al.* Proteome-wide Mendelian randomization in global biobank meta-analysis reveals multi-ancestry drug targets for common diseases. *Cell Genom* **2**, None (2022). <https://doi.org/10.1016/j.xgen.2022.100195>
- 34 Yoshiji, S. *et al.* Proteome-wide Mendelian randomization implicates nephronectin as an actionable mediator of the effect of obesity on COVID-19 severity. *Nat. Metab.* **5**, 248-264 (2023). <https://doi.org/10.1038/s42255-023-00742-w>

- 35 Zhou, S. *et al.* A Neanderthal OAS1 isoform protects individuals of European ancestry against COVID-19 susceptibility and severity. *Nat. Med.* **27**, 659-667 (2021). <https://doi.org/10.1038/s41591-021-01281-1>
- 36 Zhang, J. *et al.* Plasma proteome analyses in individuals of European and African ancestry identify cis-pQTLs and models for proteome-wide association studies. *Nat. Genet.* (2022). <https://doi.org/10.1038/s41588-022-01051-w>
- 37 Yengo, L. *et al.* Meta-analysis of genome-wide association studies for height and body mass index in ~700000 individuals of European ancestry. *Human Mol. Genet.* **27**, 3641-3649 (2018). <https://doi.org/10.1093/hmg/ddy271>
- 38 Laber, S. *et al.* Discovering cellular programs of intrinsic and extrinsic drivers of metabolic traits using LipocyteProfiler. *Cell Genom* **3**, 100346 (2023). <https://doi.org/10.1016/j.xgen.2023.100346>
- 39 Eldjarn, G. H. *et al.* Large-scale plasma proteomics comparisons through genetics and disease associations. *Nature* **622**, 348-358 (2023). <https://doi.org/10.1038/s41586-023-06563-x>
- 40 Pietzner, M. *et al.* ELF5 is a potential respiratory epithelial cell-specific risk gene for severe COVID-19. *Nat Commun* **13**, 4484 (2022). <https://doi.org/10.1038/s41467-022-31999-6>
- 41 Sabatine, M. S. *et al.* Evolocumab and Clinical Outcomes in Patients with Cardiovascular Disease. *N. Engl. J. Med.* **376**, 1713-1722 (2017). <https://doi.org/10.1056/NEJMoa1615664>
- 42 Schwartz, G. G. *et al.* Alirocumab and Cardiovascular Outcomes after Acute Coronary Syndrome. *N. Engl. J. Med.* **379**, 2097-2107 (2018). <https://doi.org/10.1056/NEJMoa1801174>
- 43 Ray, K. K. *et al.* Two Phase 3 Trials of Inclisiran in Patients with Elevated LDL Cholesterol. *N. Engl. J. Med.* **382**, 1507-1519 (2020). <https://doi.org/10.1056/NEJMoa1912387>
- 44 Carter, A. R. *et al.* Mendelian randomisation for mediation analysis: current methods and challenges for implementation. *Eur. J. Epidemiol.* **36**, 465-478 (2021). <https://doi.org/10.1007/s10654-021-00757-1>
- 45 Pietzner, M. *et al.* Systemic proteome adaptations to 7-day complete caloric restriction in humans. *Nature Metabolism* (2024). <https://doi.org/10.1038/s42255-024-01008-9>
- 46 Deedwania, P. *et al.* Efficacy and Safety of PCSK9 Inhibition With Evolocumab in Reducing Cardiovascular Events in Patients With Metabolic Syndrome Receiving Statin Therapy: Secondary Analysis From the FOURIER

- Randomized Clinical Trial. *JAMA Cardiol* **6**, 139-147 (2021).
<https://doi.org/10.1001/jamacardio.2020.3151>
- 47 Holm Nielsen, S. *et al.* The novel collagen matrikine, endotrophin, is associated with mortality and cardiovascular events in patients with atherosclerosis. *J. Intern. Med.* **290**, 179-189 (2021). <https://doi.org/10.1111/joim.13253>
- 48 GTExConsortium. The GTEx Consortium atlas of genetic regulatory effects across human tissues. *Science* **369**, 1318-1330 (2020).
- 49 Randall, J. C. *et al.* Sex-stratified Genome-wide Association Studies Including 270,000 Individuals Show Sexual Dimorphism in Genetic Loci for Anthropometric Traits. *PLOS Genetics* **9**, e1003500 (2013).
<https://doi.org/10.1371/journal.pgen.1003500>
- 50 Sun, B. B. *et al.* Genetic regulation of the human plasma proteome in 54,306 UK Biobank participants. *bioRxiv*, 2022.2006.2017.496443 (2022).
<https://doi.org/10.1101/2022.06.17.496443>
- 51 Zheng, J. *et al.* Phenome-wide Mendelian randomization mapping the influence of the plasma proteome on complex diseases. *Nat. Genet.* **52**, 1122-1131 (2020). <https://doi.org/10.1038/s41588-020-0682-6>
- 52 Dewey, F. E. *et al.* Genetic and Pharmacologic Inactivation of ANGPTL3 and Cardiovascular Disease. *N. Engl. J. Med.* **377**, 211-221 (2017).
<https://doi.org/10.1056/NEJMoa1612790>